

# The Plankton Lifeform Extraction Tool: A digital tool to increase the discoverability and usability of plankton time-series data

Clare Ostle[1*], Kevin Paxman[1], Carolyn A. Graves[2], Mathew Arnold[1], Felipe Artigas[3], Angus Atkinson[4], Anaïs Aubert[5], Malcolm Baptie[6], Beth Bear[7], Jacob Bedford[8], Michael Best[9], Eileen Bresnan[10], Rachel Brittain[1], Derek Broughton[1], Alexandre Budria[5,11], Kathryn Cook[12], Michelle Devlin[7], George Graham[1], Nick Halliday[1], Pierre Hélaouët[1], Marie Johansen[13], David G. Johns[1], Dan Lear[1], Margarita Machairopoulou[10], April McKinney[14], Adam Mellor[14], Alex Milligan[7], Sophie Pitois[7], Isabelle Rombouts[5], Cordula Scherer[15], Paul Tett[16], Claire Widdicombe[4], and Abigail McQuatters-Gollop[8]

[1]The Marine Biological Association (MBA), The Laboratory, Citadel Hill, Plymouth, PL1 2PB, UK.

[2]Centre for Environment Fisheries and Aquacu∑lture Science (Cefas), Weymouth, UK.

[3]Université du Littoral Côte d'Opale, Université de Lille, CNRS UMR 8187 LOG, Laboratoire d'Océanologie et de Géosciences, Wimereux, France.

[4]Plymouth Marine Laboratory, Prospect Place, Plymouth, PL1 3DH, UK.

[5]Muséum National d'Histoire Naturelle (MNHN), CRESCO, 38 UMS Patrinat, Dinard, France.

[6]Scottish Environment Protection Agency, Angus Smith Building, Maxim 6, Parklands Avenue, Eurocentral, Holytown, North Lanarkshire ML1 4WQ, UK.

[7]Centre for Environment Fisheries and Aquaculture Science (Cefas), Lowestoft, UK.

[8]Marine Conservation Research Group, University of Plymouth, Drake Circus, Plymouth, PL4 8AA, UK.

[9]The Environment Agency, Kingfisher House, Goldhay Way, Peterborough, PE4 6HL, UK.

[10]Marine Scotland Science, Marine Laboratory, 375 Victoria Road, Aberdeen, AB11 9DB, UK.

[11]Department of Coastal Systems, NIOZ Royal Netherlands Institute for Sea Research, Den Burg, Texel, The Netherlands.

[12]National Oceanography Centre, European Way, Southampton, S014 3ZH, UK.

[13]Swedish meteorological and hydrological institute, Sven Kallfelts gata 15, 426 71 Vastra Frolunda. Sweden.

[14]Fisheries and Aquatic Ecosystems Branch, Agri-Food and Biosciences Institute, 18a Newforge Lane, Belfast BT9 5PX.

[15]Trinity Centre for Environmental Humanities, Department of History, School of Histories and Humanities, Trinity College, University of Dublin, Ireland.

[16]Scottish Association for Marine Science, Scottish Marine Institute, Oban, PA37 1QA, UK.

Correspondence: *Clare Ostle (claost@mba.ac.uk)



## 1   Abstract

Plankton form the base of the marine food web and are sensitive indicators of environmental change. Plankton time-series are therefore an essential part of monitoring progress towards global biodiversity goals, such as the Convention on Biological Diversity Aichi Targets, and for informing ecosystem-based policy, such as the EU Marine Strategy Framework Directive. Multiple plankton monitoring programmes exist in Europe, but differences in sampling and analysis methods prevent the integration of their data, constraining their utility over large spatio-temporal scales. The Plankton Lifeform Extraction Tool brings together disparate European plankton datasets into a central database from which it extracts abundance time-series of plankton functional groups, called 'lifeforms', according to shared biological traits. This tool has been designed to make complex plankton datasets accessible and meaningful for policy, public interest, and scientific discovery. It allows examination of large-scale shifts in lifeform abundance or distribution (for example, holoplankton being partially replaced by meroplankton), providing clues to how the marine environment is changing. The lifeform method enables datasets with different plankton sampling and taxonomic analysis methodologies to be used together to provide insights into the response to multiple stressors and robust policy evidence for decision making. Lifeform time-series generated with the Plankton Lifeform Extraction Tool currently inform plankton and food web indicators for the UK's Marine Strategy, the EU's Marine Strategy Framework Directive, and for the Convention for the Protection of the Marine Environment of the North- East Atlantic (OSPAR) biodiversity assessments. The Plankton Lifeform Extraction Tool currently integrates 155,000 samples, containing over 44 million plankton records, from 9 different plankton datasets within UK and European Seas, collected between 1924 and 2017. Additional datasets can be added, and time-series updated. The Plankton Lifeform Extraction Tool is hosted by The Archive for Marine Species and Habitats Data (DASSH) at https://www.dassh.ac.uk/lifeforms/. The lifeform outputs are linked to specific, doi-ed, versions of the Plankton Lifeform Traits Master List and each underlying dataset.



## 2   Introduction

Plankton form the foundation of the marine food web, help to regulate ocean chemistry, and provide
approximately half of the world's oxygen (Capuzzo et al., 2018; Falkowski, 2012). Globally, plankton
communities are undergoing significant changes in distribution (Reid et al., 2016), community
composition (Beaugrand et al., 2002), phenology (Edwards and Richardson, 2004), and productivity
(Kulk et al., 2020). These changes vary in space and time, reflecting both direct and locally acute
anthropogenic pressure on the marine environment, such as nutrient loading, and wider-scale climate-
driven changes in ocean chemistry and temperature (Beaugrand et al., 2010; Bedford et al., 2020a).

Plankton have short life cycles, drift freely in the ocean and have wide distributions. For these reasons
they are considered to be particularly sensitive indicators to climate change (Richardson, 2008).
Changes in the composition and abundance of plankton can have negative impacts on industries such as
fisheries and aquaculture (Richardson et al., 2009; Schmidt et al., 2020). As the base of the food web,
they are a key element of the ecosystem approach to marine management (Morishita, 2008). Monitoring
plankton communities over wide spatial and long temporal scales can help tease apart the prevailing
footprint of climate change on marine ecosystems from other, more localised pressures, for example,
pollution, nutrient loading and fishing (Bedford et al., 2020b). Consequently, plankton time-series play
an increasingly important role in decision-making and provision of advice. Plankton indicators
contribute to the delivery of global, regional and national policy drivers such as the Convention on
Biological Diversity's Aichi targets (Chiba et al., 2018), the regional Convention for the Protection of
the Marine Environment of the North-East Atlantic (OSPAR) (OSPAR, 2017), and biodiversity state in
the European Union's Marine Strategy Framework Directive (MSFD) and the UK Marine Strategy
(McQuatters-Gollop et al., 2019).

Although there are a number of programmes that monitor plankton in Northwest European waters, they
operate at different spatial scales, from fixed-point sampling stations to long-distance continuously
sampled ship transects (O'Brien et al., 2017). Furthermore, European plankton surveys employ different
sampling methods, enumerate specimens at a variety of taxonomic levels and employ different counting
regimes (Raybaud et al., 2011). These methodological differences and the lack of direct comparability



between datasets has meant that the tools to use all available datasets together to produce a comprehensive assessment have only recently been developed (Bedford et al., 2020a; McQuatters-Gollop et al., 2019). While most datasets are regularly submitted to appropriate data repositories (e.g.: the Ocean Biodiversity Information System: OBIS; the British Oceanographic Data Centre: BODC, or the PANGAEA data publisher for earth and environmental science) and some are available through

institutional websites or data centres, the aggregation of plankton data into functional groups (or 'lifeforms' e.g.: diatoms, dinoflagellates, holoplankton, meroplankton) has not yet been linked to traceable dataset versions or been possible to apply in an accessible, transparent and centralised way. Accordingly, understanding of plankton change across multiple spatial and temporal scales has been limited. The International Group of Marine Ecological Time Series IGMETS (https://igmets.net;

O'Brien et al., 2017), represents valuable progress towards this goal: it provides a global-scale compilation of pelagic time series, with a tool to summarise visualisations of trends across a variety of temporal and spatial scales. However, this initiative summarises time trends of highly aggregated variables (e.g.: total zooplankton) for multiple sites. Trajectories of the key component plankton functional groups are not described, and the underlying data products are not made available to users for

further analysis. Aggregating these disparate plankton datasets increases the spatial-temporal scope of analysis, increases their robustness and provides decision makers with more scientifically robust evidence.

Building on previous work (Gowen et al., 2011; Scherer et al., 2014; Tett et al., 2008, 2013) an indicator of shifts in plankton structure based on time-series of broad plankton functional groups, called

'lifeforms', has been developed for use in policy assessments (McQuatters-Gollop *et al*., 2019). The term 'lifeform' is derived from work carried out by Margalef (1978), to distinguish between diatoms and dinoflagellates based on traits related to survival in specific hydrodynamic conditions. Lifeforms differ slightly from the term 'Plankton Functional Type' (PFT), in that PFTs are often used to describe plankton based on their ecosystem function and not on their traits. This indicator enables plankton

datasets with different sample collection and analysis routines to be used congruently to investigate changes in pelagic habitat functioning. By using these pre-defined lifeforms to group plankton taxa, the new Plankton Lifeform Extraction Tool (PLET), hosted by the Archive for Marine Species and Habitats



Data (DASSH, https://www.dassh.ac.uk), brings together disparate plankton datasets, increasing their accessibility and promoting compliance with the FAIR data principles (Wilkinson et al., 2016). The

110 PLET enables the user to investigate multiple datasets to assess changes in plankton ecology at multiple spatio-temporal scales using a consistent plankton indicator for the first time. As time-series grow in length and/or spatial distribution, and new plankton time-series are established, additional plankton taxa and datasets can be added to the PLET in order to improve future biodiversity assessments. The tool is a key step towards transparent and standardised assessment allowing the integration of information from

115 multiple datasets at multiple spatial and temporal scales.

## 3    Plankton datasets

In its current form, the PLET integrates 155 thousand samples containing over 44 million plankton records from 9 different data providers around the UK and European Seas, collected between 1924 and 2017 (**Table 1**). Flexibility of the PLET design allows existing time-series to be updated and new time-

series to be added, continuing the expansion of integrated data sets beyond the UK, where policy reporting motivated its initial development. Plankton time-series have been collected both along transects and at fixed-point stations (**Figure 1**). These datasets, which underpin the PLET lifeform outputs, enumerate plankton in taxon groupings (see **Table A1**).

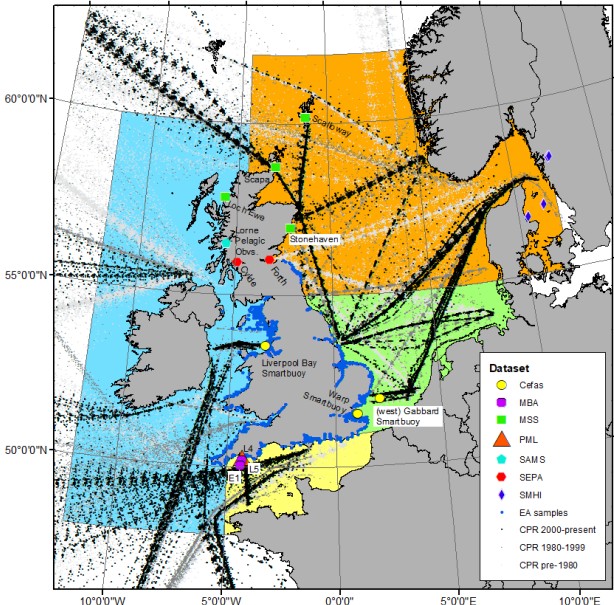

**Figure 1. Spatial coverage of plankton measurements** currently integrated with the PLET tool for lifeform extraction. See data provider and station information in **Table 1**, individual station names are given next to the symbols, while symbols designate data providers as shown in the legend. The sampling transects for the CPR are coloured by date sampled, with pre-1980 in light grey, 1980-1999 in darker grey, and the most recent 2000-onwards in black. Coloured regions indicate how data are summarised for presentation of Lifeform outputs in **Section 6**: blue: Celtic Seas; red: northern North Sea; green: southern North Sea; light yellow: Channel.





**Table 1. Plankton data currently held in PLET** and used to produce the aggregated lifeform outputs. For un-aggregated plankton data, contact information and institute-specific data holdings (where available) are given for each data institute. Most of these time series are ongoing and many sample at higher temporal resolution than the monthly average data held in PLET. Prospective users for these higher resolution versions of the respective time series are encouraged to consult with the contact people listed below.

| **Institute,** dataset name, *primary contact*; data web address, [PLET doi] | **Region or Station Name** | **Sampling Period** | |
| --- | --- | --- | --- |
| | | **Phytoplankton** | **Zooplankton** |
| **The Marine Biological Association** (MBA), Continuous Plankton Recorder Survey, *David Johns (djoh@mba.ac.uk);* https://data.cprsurvey.org/datacatalog/ https://doi.org/10.17031/1629 (CPR and Johns, 2019) | UK and European Seas | 1958-2017 | 1958-2017 |
| **The Marine Biological Association** (MBA), Station sampling, *Rachel Brittain (racbri@mba.ac.uk)* https://doi.org/10.17031/1636 (MBA, 2019) | L5 | not determined | 1924 - 1940, 1945 - 1987, 2001 - 2013 |
| | E1 | | |
| **Plymouth Marine Laboratory** (PML) *Angus Atkinson (aat@pml.ac.uk);* http://www.westernchannelobservatory.org.uk/ https://doi.org/10.17031/1632 (PML, 2019) | L4 | 1992-2015 | 1988-2017 |
| **Centre for Environment Fisheries and Aquaculture Science** (Cefas), Smartbuoys, *Michelle Devlin (michelle.devlin@cefas.ac.uk);* https://www.cefas.co.uk/cefas-data-hub/smartbuoys/ https://doi.org/10.17031/1634 (CEFAS, 2019) | Dowsing | 2000-2017 | not determined |
| | Gabbard and West Gabbard | 2001-2017 | |
| | Liverpool Bay | 2002-2017 | |
| | Warp | 2001-2012 | |
| **Environment Agency** (EA) *Mike Best (mike.best@environment-agency.gov.uk)* https://doi.org/10.17031/1635 (EA, 2019) | UK coastal and transitional waters | 2010-2017 | not determined |
| **Marine Scotland Science** (MSS) *Eileen Bresnan & Margarita Machairopoulou (Eileen.Bresnan@gov.scot; Margarita.Machairopoulou@gov.scot)* https://data.marine.gov.scot/search/type/dataset https://doi.org/10.17031/1637 (MSS, 2019) | Stonehaven | 1997-2017 | 1999-2017 |
| | Loch Ewe | 2002-2017 | 2002-2017 |
| | Scapa (Orkney Islands) | 2001-2017 | not determined |
| | Scalloway (Shetland Islands) | 2001-2017 | |
| **Swedish Meteorological and Hydrological Institute** (SMHI) *Marie Johansen (marie.johansen@smhi.se)* https://sharkweb.smhi.se/ https://doi.org/10.17031/1633 (SMHI, 2019) | Swedish west coast | 1986-2015 | 1998-2015 |
| **Scottish Environment Protection Agency** (SEPA) *Malcolm Baptie (Malcolm.Baptie@sepa.org.uk)* https://doi.org/10.17031/b84a-7951 (SEPA, 2020) | Forth | 2007-2017 | 2014 - 2017 |
| | Clyde | | |
| **Scottish Association for Marine Science** (SAMS) *Paul Tett (Paul.Tett@sams.ac.uk)* https://doi.org/10.17031/nz24-br35 (SAMS, 2020) | Lorne Pelagic Observatory | 1970-2015 | not determined |



### 3.1 Plankton sampling and analysis methodology

All individual datasets that have been added to the PLET have been pre-processed to ensure suitability for extraction of monthly-aggregated lifeform data products. Pre-processing was the responsibility of the individual data providers. Examples of pre-processing required are (i) the exclusion of instances of 140 'double counting' where, for example, a taxon is included in both higher and lower taxonomic groupings within the same dataset and (ii) the removal of taxa that have not been looked for (recorded) over the entire time-period to avoid apparent changes in lifeform abundance due to methodological changes.

Existing datasets were gathered through a data call issued by OSPAR in 2016. The purpose of the data 145 call was to gather plankton datasets to use for assessment and reporting for the European Union's and individual Member States' Marine Strategy Framework Directive initial biodiversity assessment in 2017 (https://oap.ospar.org/en/ospar-assessments/intermediate-assessment-2017/biodiversity-status/habitats/changes-phytoplankton-and-zooplankton-communities/). A simple data submission template was developed as part of this process and is now available on the PLET website for wider use. 150 To make data submission as simple and easy as possible, the template allows data-holders to submit the datasets in either list (long) or matrix (wide) formats. A data archiving and access permission agreement form is also available from the PLET website, and allows data-holders to specify their preferred level of data access, such as full access to raw data or access to lifeform data products only.

All plankton records currently included have been identified using light microscopy. For simplicity we 155 use the term "phytoplankton" to mean protist cells, mindful that these include a spectrum of auto-, mixo- and heterotrophic forms (Flynn et al., 2013). This terminology is used to differentiate from "zooplankton" which are the metazoans usually counted from net haul samples. For quality assurance, analysts participate in NMBAQC (the Northeast Atlantic Marine Biological Analytical Quality Control Scheme) and the International Phytoplankton Intercomparison external identification ring trials, 160 although these do not cover the full length of some of the historical data sets. Field abundance, in individuals per unit volume, is calculated as sample abundance multiplied by subsample factor, divided by the sampled water volume. Concentrations of phytoplankton identified by light microscopy are





typically expressed as numbers (cells) per mL, and those of zooplankton are typically expressed as numbers (individuals) per m$^{-3}$.

### 165    3.1.1    *Continuous Plankton Recorder Survey (Marine Biological Association)*

The Continuous Plankton Recorder (CPR) is a marine sampler that is towed behind volunteer ships of opportunity at speeds of up to ~20 knots and samples at a depth of ~7 m below the surface. Plankton have been sampled on routes crossing the North Atlantic and NW European shelf seas using a consistent methodology since 1958.

The CPR unit is a metal casing in the shape of a ~1 m torpedo that houses a roll of silk which automatically rotates using a geared propeller system. The seawater enters the front aperture where plankton and small particles are captured onto the rotating silk, which has a mesh size of 270 μm. This silk is stored in 4 % buffered formalin to preserve the sample until microscopic analysis at the laboratory in Plymouth. The silk is cut into pre-defined sections that represent one sample and equate to

10 nautical miles of tow. Phytoplankton and zooplankton are identified and counted at different stages of the microscopic analysis: semi-quantitative count of phytoplankton across 20 fields of view per sample, quantitative count of all zooplankton >= 2 mm (these are picked off the silk for identification), and semi-quantitative traverse count of all zooplankton < 2 mm.

For a more in-depth description of the sampling methodology please refer to Richardson et al. (2006).

CPR monthly abundance counts from 1958 to 2017 are available from the following open access data portal: https://data.cprsurvey.org/datacatalog/.

### 3.1.2    *Western Channel Observatory (Marine Biological Association and Plymouth Marine Laboratory)*

The Marine Biological Association (MBA) and Plymouth Marine Laboratory (PML) jointly sample at

three offshore stations in the western English Channel as part of their Western Channel Observatory (https://westernchannelobservatory.org.uk). These stations are termed: L4 (50.25˚N, 4.3˚W; approx. 55 m water depth) 13 km south-west of Plymouth, which can be regarded as a coastal station, albeit in transitionally stratified water; L5 (50.18˚N, 4.3˚W; approx. 58 m depth) is positioned between coastal and offshore waters, and E1 (50.03˚N, 4.37˚W; approx. 70 m depth) is 40 km offshore in seasonally





stratified water. Sampling at these historical sites began in 1924 with interruptions between 1940-45 and 1987-2001. Sampling frequency has varied between weekly and fortnightly; current sampling is weekly at station L4 and, weather-permitting, fortnightly at L5 and E1.

The phytoplankton and zooplankton time-series at L4 are provided by PML. Sampling for phytoplankton began in 1992 and for mesozooplankton in 1988. Detailed phytoplankton taxonomic
microscope counts are from water samples collected at 10 m depth. These samples are preserved in 2% acid Lugol's iodine solution and enumerated for all taxa larger than approximately 2 μm using the Utermöhl (1958) technique, usually settling 50 ml (Widdicombe et al., 2010). Mesozooplankton are collected each week in two replicate 0-50 m vertical hauls with a WP2 net (0.57 m diameter, 200 μm mesh-size). Each of these are analysed in two aliquots, the first being a stempel pipette – derived small
subsample for enumeration of the more numerous taxa and the second larger fraction, often one-half to one-eighth, analysed for the larger or rarer taxa.

Weekly densities are calculated as the average of the two separate net hauls. Environmental conditions and the mesozooplankton sampling and analysis methods are described in detail in Atkinson et al. (2015).

Macroplankton and larval fish sampling at the WCO sites is carried out by the MBA. Although net design and methods of deployment have changed on several occasions, care has been taken to ensure that sampling characteristics have not altered appreciably. The 1 m$^2$ Young Fish Trawl (YFT), fitted with a 700 μm knitted mesh is hauled for 20 minutes in an oblique profile to an ideal depth ~5 m above the seabed. Depth and temperature profiles are occasionally recorded and the volume of water filtered
calculated using flow data recorded by a flowmeter fitted across the net mouth. The samples are preserved in 4 % buffered formalin and analysed as soon as possible after collection using a WILD M5 binocular microscope. Results are standardized to number of individuals per 4000 m$^3$ in order to mitigate historical changes in sampling gear and deployment.

A comprehensive summary of these macroplankton sampling methods and analysis is given in
Southward et al. (2004) and references therein.



### 3.1.3 Smartbuoys (Centre for Environment Fisheries and Aquaculture Science)

Water samples for phytoplankton analysis are collected from several of the Centre of Environment Fisheries and Aquaculture Science (Cefas) 'Smartbuoy' moorings using automated water samplers mounted at 1 m below the surface. Time-series at approximately monthly resolution from four buoy

stations are available: Dowsing off the Humber estuary (51.53°N, 1.05°E, sampled 2000-present), Gabbard/West Gabbard off the Thames estuary (51.95°N, 2.11°E, sampled 2001-present), Warp in the outer Thames estuary (51.52°N, 1.028°E, sampled 2001-2012), and Liverpool Bay (53.53 °N, 3.35°W, sampled 2002-present).

Water samplers are pre-programmed to collect 150 mL samples on an approximately weekly cycle into

225 sample bags pre-spiked with acidified Lugol's iodine solution. Phytoplankton samples are returned for analysis at Cefas every 1–3 months, where they are decanted into 175 mL glass jars and topped up with acidified Lugol's iodine. A minimum of one sample per month is selected for analysis from each deployment location where sample availability allows. Samples are analysed at Cefas using the Utermöhl (1958) technique under inverted Olympus microscopes within 1 year of collection. Species

are identified and enumerated to the lowest possible taxonomic level and counts recorded in cells per litre.

More detailed methodology is available in Weston et al. (2008) and Greenwood (2019). Plankton and environmental parameters from the Smartbuoy monitoring programme are available from the Cefas Data Hub: https://www.cefas.co.uk/data-and-publications/smartbuoys/.

### 3.1.4 England's Estuarine and Coastal Waters (Environment Agency)

The Environment Agency (EA) and its predecessors have been collecting phytoplankton on targeted campaigns since the 1990's, however from the inception of EU Water Framework Directive (WFD; EU, 2000) monitoring in 2006, Environment Agency routine phytoplankton samples have been collected from sites in near-shore WFD waterbodies from boats, or occasionally, jetties or bridges in estuaries

Devlin et al. (2012).

Sampling in WFD transitional and coastal waters typically consists of one sample per calendar month from 3 to 5 sites per water body. Ideally, samples should be 28-31 days apart throughout the year. There

must be at least a 14-day interval between sampling occasions at each site. Phytoplankton samples are taken in the mixed surface layer usually between 1-2 m below the water surface using a standard
NIO/Niskin-style water sampler, avoiding the surface film and without disturbing bottom sediments. In coastal or non-turbid waters >5m depth, the diurnal vertical migration of phytoplankton with light availability is accommodated by collection during daylight hours. However, for some samples, the use of integrated depth sampling using a Lund-type tube system negated the need to constrain the sampling window to daylight hours. Samples are collected in 250ml clear PET bottles filled to approximately
90%, leaving sufficient headspace to allow for preservation and homogenisation. Samples are preserved with acidified Lugol's iodine, and stored in the dark, ideally at a temperature of 3°C ±2°C for no longer than 6 months. Samples are analysed using the Utermöhl (1958)  method under inverted microscopes. Analysis was conducted at Cefas until 2013, then at both Cefas and an external laboratory from 2013 onwards. Some samples are analysed by multiple analysts to check for comparability of results.

*3.1.5   The Scottish Coastal Observatory (Marine Scotland Science)*
Marine Scotland Science (MSS) routinely samples the plankton in Scottish waters as part of the Scottish Coastal Observatory. Weekly phytoplankton samples have been collected from Stonehaven (56.96˚N, 2.11 ˚W) since 1997, Scapa (Orkney Islands; 58.74 ˚N, 2.97 ˚W) since 2001, Loch Ewe (57.84 ˚N, 5.65 ˚W) since 2023 and Scalloway (Shetlands, 60.18 ˚N, 1.28 ˚W) since 2001. Meso-zooplankton have been
sampled, also weekly at Stonehaven since 1999 and Loch Ewe since 2002.

Phytoplankton samples are collected using a 10 m integrated tube sampler. A 1 L subsample is preserved with 0.5% acidic Lugol's iodine and returned to the MSS Marine Laboratory. Phytoplankton samples are analysed using a modified Utermöhl (1958) technique. Phytoplankton samples are analysed using an inverted Zeiss Axiovert microscope. The presence/absence of all cells in the chamber are
265 recorded and fields of view across a transect are counted at X200 magnification. Phytoplankton are identified to the lowest taxonomic level possible, however due to the limitations of light microscopy and the Lugol's fixative in some instances a genus level identification or 'unidentified' category is assigned.



Zooplankton samples are collected using 40 cm diameter bongo nets fitted with 200 μm mesh and
270 filtering cod ends. The nets are hauled vertically from near bottom (45 m at Stonehaven and 35 m at
Loch Ewe) to surface at a speed of 1 m s$^{-1}$. The samples are immediately fixed in 4 % borax buffered
formaldehyde for later analysis in the laboratory. Zooplankton samples are analysed in the laboratory
using a Zeiss Stemi SV-11 stereomicroscope. Larger zooplankton categories (such as *Calanus* spp.,
chaetognaths, jellyfish, euphausiids etc.) are identified and enumerated from the whole sample. The
275 remaining zooplankton categories are identified and enumerated from a series of subsamples (of
variable volumes depending on concentration of animals but a minimum 2.5 % of the whole sample) so
that at least 100 animals of the most common taxa are recorded. Most taxa are identified to the lowest
taxonomic level possible, whilst other animals are recorded at Class or Phylum level.

More detailed methodology is available in Bresnan et al. (2016). Phytoplankton monthly densities from
280 Stonehaven, Loch Ewe, Scapa and Scalloway, and zooplankton weekly densities from Stonehaven and
Loch Ewe are available from: https://data.marine.gov.scot/search/type/dataset.

### 3.1.6   Scotland Coastal Stations (Scottish Environment Protection Agency)

The Scottish Environment Protection Agency (SEPA) collects plankton samples at two near-shore
stations (Forth: 56.03˚N, 3.18˚W; Clyde: 55.95˚N, 4.89˚W). Monthly samples for phytoplankton have
285 been collected since 2007 and zooplankton since 2014.

Phytoplankton samples are collected using an integrated tube column water sampler with foot valve and
closure tap, which is lowered open to 10 m depth. The closure tap is then moved to the closed position
and the sampler retrieved. The foot valve is opened and the contents of the sampler are emptied into a
rinsed bucket. A 250 ml sample bottle prefilled with 2.5 ml of 5% w/v Lugol's iodine solution is gently
submerged to fill with water from the bucket. The sample bottle is gently inverted to mix preservative
and stored in the dark in a refrigerator at 4 °C. Samples of phytoplankton are removed from cold storage
and left to acclimatise at room temperature for 24 hours after which they are gently inverted 100 times
to re-suspend settled cells and a volume of sample, typically 50 ml, poured into a sample tube and left to
settle for 24 hours. After this time, 40ml of supernatant is drawn off slowly and discarded. The
295 remaining 10 ml of sample is then gently inverted 100 times before being carefully poured into a 10 ml



Utermöhl (1958) counting chamber. This is then left to settle for a further 24 hours before being analysed on an inverted microscope (Leica DM IRB or Leica DMI4000B – Wetzlar, Germany or Zeiss Axiovert S100 – Jena, Germany). The chamber plate is scanned to assess rough composition of the sample and to determine if settled cells are randomly distributed. Depending on the cell type, size and
density, cell counts are made of the whole counting chamber, a number of transects of the widest point, or a number of random fields of view. At least 400 cells are counted when employing transect or field of view counting strategies. Field abundance in cells $L^{-1}$ is calculated by multiplying sample count by microscope subsample factor and 1000 divided by settled volume.

Zooplankton samples are taken with a 27 cm diameter net fitted with a 200 μm mesh with a non-
filtering 1 litre cod end. A Hydro-bios (Kiel-Altenholz, Germany) digital flow meter with a back run stop is fitted to the mouth of the net in order to determine the volume hauled, and therefore abundance in individuals $m^3$. The net is deployed vertically from near-bottom to the surface at approximately 0.5 m $s^{-1}$. Upon recovery, the net is rinsed with seawater and the contents of the cod ends are transferred into a sample bottle and preserved in 4 % borax buffered formaldehyde. These samples are gently rinsed
through a 63 μm wire mesh sieve for microscopic analysis using Leica (Wetzlar, Germany) M165C microscopes. Abundance is determined by counting in the full sample any zooplankton larger than stage IV *Calanus* (including and from copepodite stage V) and enumerating all other zooplankton in a subsample taken using a Folsom or Motoda splitter or a plunge sampler as appropriate to achieve an acceptable density of zooplankton, being no less than 100 of the most abundant taxa.

*3.1.7   Lorne Pelagic Observatory (Scottish Association of Marine Science)*
Phytoplankton samples have been collected weekly at the Lorne Pelagic Observatory (56.48 ˚N, 5.5˚W) since 1970 by the Scottish Association of Marine Science (SAMS). Water samples for microplankton (i.e.: phytoplankton and pelagic micro-heterotrophs) are taken with water bottles and in some cases with a 10 m integrating hose. They are preserved with 0.5% acidic Lugol's iodine and volumes of 10 to 50
320  mL sedimented for counting using the Utermöhl (1958) technique and Wild and Zeiss inverted microscopes equipped with phase contrast. Depending on abundance and organism size, a variety of counting patterns are used, ranging from examination of the whole base of the sedimentation chamber at low power to narrow transects or a few fields at high power.



### 3.1.8 Swedish West Coast (Swedish Meteorological and Hydrological Institute)

The Swedish Meteorological and Hydrological Institute (SMHI) samples both phytoplankton and zooplankton at four stations on the Swedish west coast. Phytoplankton are sampled as an integrated sample using a hose (0-10 m) and preserved in acidic Lugol's; alkaline Lugol's is used for counts of coccolithophores. Twenty-five ml of each sample is analysed using the Utermöhl (1958) method. The samples are stored in the dark and at room temperature prior to analyses. Zooplankton are sampled with a WP2 net (100 μm mesh size), and an integrated sample is taken from 0-25 m. Samples are preserved in formalin and stored in the dark prior to analyses. The subsample volume used when counting depends on the concentration of copepods in the sample to enable statistically sound data.



### 3.2 *Spatio-temporal data distribution*

The plankton datasets currently available for lifeform extraction by the PLET have variable spatial and
335 temporal extents, summarised herein into the four regions shown in **Figure 1**. Within each region, the
availability of plankton data over time differs between datasets (**Figure 2**). Due to their high spatial
coverage the CPR and EA datasets contain the largest numbers of samples available for any month,
within each region. The number of samples at fixed-point sampling stations shows variations in
sampling frequency; in some cases this has changed over the course of the time-series (for example, in
the Celtic Seas both Cefas Smartbuoy and the SAMS dataset).

While sampling is typically carried out weekly or monthly in order to capture the seasonal cycle of the
plankton community structure and rapid changes associated with plankton bloom events, several
datasets include sampling gaps and changes in sampling intensity for a variety of reasons (**Figure 3**).
For example, the EA dataset sampling frequency (and spatial distribution of samples) increased
alongside the implementation of the Water Framework Directive (EU, 2000) in 2007, while the SAMS
time-series stopped between 1982 and 1999. Missing months in the Cefas Smartbuoy time-series largely
indicate failures of the automated sampling system, or sample loss related to logistical delays in buoy
servicing. Ongoing sampling of all time-series are at risk of additional reductions in sampling frequency
and quality related to funding (McQuatters-Gollop et al., 2017; Zingone et al., 2015).



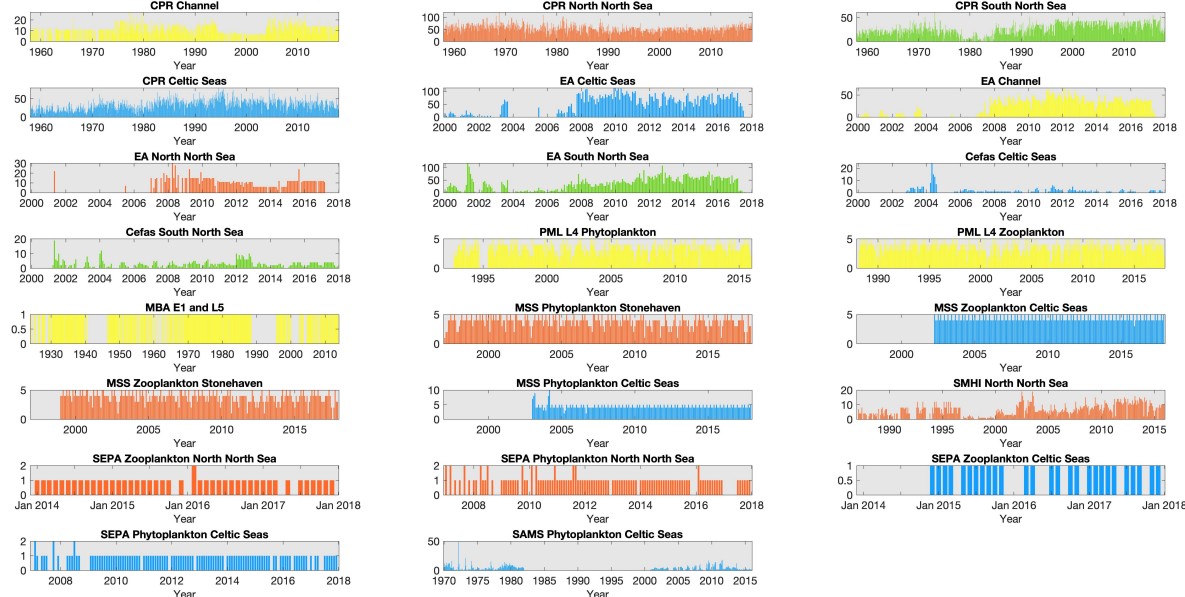

**Figure 2. Sampling effort of each dataset within each region**: number of sampling timepoints collected per month for each dataset within each of the regions defined in **Figure 1**, except for MSS stations in the northern North Sea where only Stonehaven is shown as an example of the three MSS stations in that region. Note that axis limits are not fixed between panels. Bar colour indicates spatial region (see **Figure 1**); blue: Celtic Seas; red: northern North Sea; green: southern North Sea; light yellow: Channel.





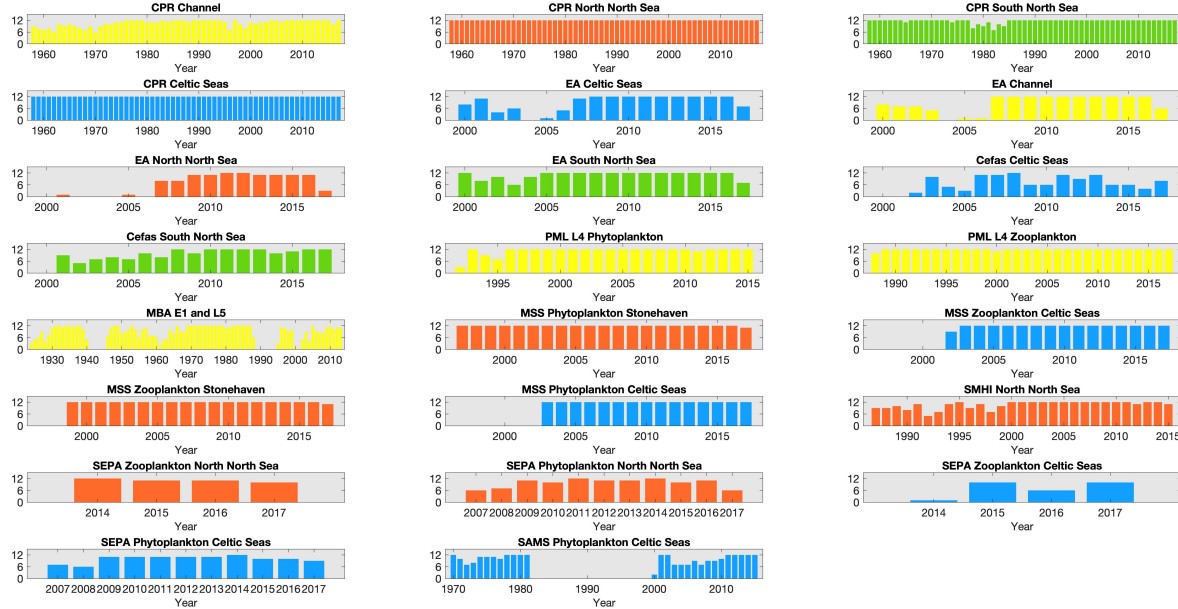

**Figure 3**. Number of months sampled per year for each dataset within each region: number of months sampled within each year, for each dataset within each of the regions defined in **Figure 1**, except for MSS stations in the northern North Sea where only

55 Stonehaven is shown as an example of the three MSS stations in that region. Widths of the bars indicate the total time-series length. Bar colour indicates spatial region (see **Figure 1**); blue: Celtic Seas; red: northern North Sea; green: southern North Sea; light yellow: Channel.



## 4 Plankton lifeforms

The PLET uses a trait look-up table to aggregate plankton taxa into lifeforms. The lifeforms have been
pre-defined using biological traits to represent groups of plankton which perform similar ecological
functional roles (McQuatters-Gollop et al., 2019; Scherer et al., 2014). Details of each lifeform, and
lifeform pairings with ecological relevance for assessment, are given in **Table 2**. It should be noted that
these traits have been developed for marine taxa only (see list of included taxon groups in **Table A1**,
and traits in **Table B1**), with the goal of simplifying plankton datasets for use in assessments; they are
375 not intended as a fully comprehensive list of plankton traits.

The trait look-up table (Plankton Lifeform Traits Master List (UK Pelagic Habitats Expert Group, 2020)
was developed by using a combination of extensive literature synthesis and expert opinion. The World
Register of Marine Species (WoRMS Editorial Board, 2020) Aphia IDs are used to link the taxa to their
associated traits. Confidence in the lifeforms extracted is assigned based on a combination of the ability
to identify each of the taxa it comprises reliably by light microscopy and the ability to assign traits to
each taxon (**Table 3**). The confidence associated with each lifeform is described in **Table 4**. Only
lifeforms with a 'high' confidence rating are provided in the PLET outputs. In some cases, confidence
assignment reflects the limitations of identification by light microscopy by nature of the datasets around
which the table was developed. Similarly, the size-based lifeforms (as currently defined and used)
reflect the size limitations of sampling and identification of the currently included datasets. For
example, the lifeform 'small phytoplankton', defined as phytoplankton with size < 20 μm, is more
appropriately termed 'small micro-phytoplankton' because while the lower size limit of identification
by routine light microscopy will include some large nanophytoplankton, it excludes pico-plankton and
the smaller nano- phytoplankton.





**Table 2. List of lifeforms and their ecological importance**. The definitions of the lifeforms (see also McQuatters-Gollop et al. (2019) based on the trait lookup table are given in **Table C1**.

| Lifeform | Definition[1] | Ecological Importance |
|---|---|---|
| (micro)Phytoplankton (size range determined by possible enumeration by light microscopy) | Protista taxa that contribute to primary production | Encompasses key primary producers, with notable exclusion of pico-, small nano- and microphytoplankton Important for food web support, dynamics and biogeochemical cycling. |
| Large microphytoplankton | ≥ 20 μm individual cell diameter | Changes in relative abundance provides a size-based indicator of the efficiency of energy flow to higher trophic levels. (Schmidt *et al.*, 2020). |
| Small microphytoplankton | < 20 μm individual cell diameter, with lower size limit determined by current enumeration by light microscopy. | |
| Diatoms | Taxa of the class Bacillariophyceae | Key groups of primary producers. Changes in abundance, and relative abundance in particular, are used to monitor changes in ecosystem functioning (Hinder et al., 2012; Wasmund et al., 2017). Dominance of dinoflagellates over diatoms may be an indicator of eutrophication or of change in water column stability, indicating changes in nutrient concentration or stratification (Devlin et al., 2009; Wasmund et al., 2017). When dinoflagellates are mainly heterotrophic, then they can account for a significant part of diatom grazing (as in the Eastern English Channel; Grattepanche et al. (2011)). |
| Dinoflagellates | Taxa of the phylum Dinoflagellata | |
| Autotrophic and mixotrophic dinoflagellates | Autotrophic: nutrition by photosynthesis; Mixotrophic: capable of obtaining nourishment via photo(auto)trophy and phago(hetero)trophy, as well as via osmo(hetero)trophy (see Flynn *et al.*, 2019) | Shift in primary producers may indicate eutrophication (Gowen et al., 2012). |
| Pelagic diatoms | Diatoms living in the water column. | Changes in relative abundance provides an indicator of benthic disturbance and frequency of resuspension events (Cibic et al., 2012). |
| Tychopelagic diatoms | Benthic diatoms which can become mixed into the water column. | |
| Potentially toxic or nuisance diatoms | Diatoms and dinoflagellates which are either 'toxic', defined as *capable of producing toxins which can cause illness or death in humans, animals and/or fish*, or 'nuisance' defined as taxa producing effects which are detrimental to aquaculture and benthos via physical harm or causing anoxia or produce water discolorations, scums or foams that can be aesthetically, socially, or economically negative. | These groups include species which have the potential for negative impacts on human health and provision of ecosystem services for people as well as other higher trophic levels of the system (Hallegraeff et al., 2021; Wells et al., 2020). |
| Potentially toxic or nuisance dinoflagellates | | |
| Ciliates | Protozoans characterized by the presence of cilia. | Increases in abundance could indicate a shift from primarily autotrophic to a more heterotrophic system (Scherer, 2012). |
| Holoplankton | Zooplankton taxa which spend their entire lifecycle in the plankton. | Changes in relative abundance provides an indicator of change in pelagic-benthic food web structure (Bedford et al., 2020a; Kirby et al., 2008). |
| Meroplankton | Taxa which spend part of their lifecycle as zooplankton. | |



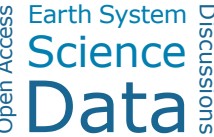

| Lifeform | Definition[1] | Ecological Importance |
|---|---|---|
| Gelatinous zooplankton | Taxa of the phyla Cnidaria and Ctenophora only. | Changes in relative abundance indicate potential alternative energy flows through the food web of varying importance to fisheries, aquaculture, tourism etc. (Richardson *et al.*, 2009). |
| Fish larvae and fish eggs | Includes fish eggs as well as larvae. | |
| Carnivorous zooplankton | Taxa which prey mainly on other zooplankton. | Non-carnivorous functionally refers to zooplankton that could be grazers on phytoplankton, at some point in their lifecycle. Changes in relative abundance of carnivorous and non-carnivorous zooplankton indicates a shift in energy flow and balance between primary consumers and secondary consumers. |
| Non carnivorous zooplankton | Zooplankton with less-carnivorous feeding mechanisms, i.e.: predominately suspension or filter feeders, omnivores which can use both carnivorous and herbivorous feeding, or ambiguous (diet uncertain). | |
| Crustaceans | Taxa of the Subphylum Crustacea | Crustaceans are important for commercial fisheries, either directly or in food chains that fuel them. Changes in crustacean zooplankton abundance can reflect both bottom-up and top-down controls and may indicate changes in food availability for exploited fish stocks (Capuzzo et al., 2018). |
| Large copepod species (≥ 2 mm) | ≥ 2 mm adult total body length | Changes in relative abundance provide a size-based indicator of food web structure and energy flow (Daufresne et al., 2009). |
| Small copepod species (< 2 mm) | < 2 mm adult total body length | |

[1] Modified from McQuatters-Gollop et al. (2019)

05





**Table 3. Lifeform confidence assignment matrix**, where 'High', 'Medium', or 'Low' are based on the ability to identify and assign traits for the constituent taxa groups of a lifeform.

|  | Can assign traits to constituent taxa | Cannot assign traits to constituent taxa |
|---|---|---|
| **Can identify constituent taxa** | high | medium |
| **Cannot identify constituent taxa** | medium | low |

**Table 4. Lifeform confidences** based on ability to identify and assign traits, applying rationale in **Table 3**. Only lifeforms with a 'high' confidence rating are provided in the PLET outputs.

| Lifeform | Confidence | Reason for confidence (where not 'high') |
|---|---|---|
| (micro)Phytoplankton | High | |
| Large microphytoplankton | Medium | Can reliably identify individual plankton species size class but cannot always reliably assign the size trait if the group counted spans taxa that are both larger and smaller than 20 microns. |
| Small microphytoplankton | Medium | |
| Diatoms | High | |
| Dinoflagellates | High | |
| Autotrophic and mixotrophic dinoflagellates | Medium | Can identify taxa, but assigning feeding mechanism trait is not always clear (see discussion in Flynn et al. (2019) |
| Pelagic diatoms | High | |
| Tychopelagic diatoms | High | |
| Potentially toxic and nuisance diatoms | Low | Designation of some algal blooms as "harmful" (i.e.: Harmful Algal Blooms, 'HABS'), relates more to societal assessment than plankton traits, these 'lifeforms' are therefore not currently recommended for use though they are defined in the traits list and will be the focus of future development work. Specific issues include:<br>• The toxin producing diatom genus *Pseudo-nitzschia* contains both amnesic shellfish toxin-producers which can render shellfish unfit for human consumption and potentially negatively impact the health of marine mammals, and non toxin-producing species/individuals. It is not possible to identify these cells to species level using routine light microscopy; some toxin and non-toxin producing species are morphologically identical. |
| Potentially toxic and nuisance dinoflagellates | Low | • The genus *Alexandrium* contains both paralytic shellfish toxin- and non-toxin-producing species/strains and it is not possible to distinguish these using routine light microscopy; some toxin and non-toxin producing species are morphologically identical.<br>• Currently it is unknown if the negative impact from *Karenia mikimotoi* in European waters is via toxin production or anoxia arising from high biomass blooms.<br>• Not all datasets included in PLET reliably record key species (e.g.: CPR does not record *Alexandrium*) |
| Ciliates | Low | • Ecological function can be duplicated by heterotrophic and mixotrophic dinoflagellates.<br>• Ciliates do not preserve well in the standard 0.5% Lugol's iodine preservative used to preserve phytoplankton samples and some (but not all) are too small and too fragile to be well sampled by many of the datasets currently in PLET. |
| Holoplankton | Medium | • May not identify taxa specifically enough to determine traits. |
| Meroplankton | Medium | • Some of the rarer species are resuspended from the seabed and definition of their holo- or meroplanktonic status is difficult |
| Gelatinous zooplankton | High | |
| Fish larvae | High | |
| Carnivorous zooplankton | Medium | Can identify taxa, but assigning diet trait is unclear, especially at different life stages. |
| Non-carnivorous zooplankton | Medium | |
| Crustaceans | High | |
| Small copepods | High | |
| Large copepods | High | |



## 5    Plankton lifeform extraction tool functionality

The PLET is accessed through a web-based user-interface (see **Figure 4**). To generate custom plankton lifeform outputs, users select: time-series start and end dates, a spatial area, and a data set. Because of methodological differences in sampling and analysis it is not appropriate to produce average lifeforms

across the multiple datasets, as such all sample locations within the selected spatial area for any data set are aggregated into a single lifeform data product but stations from different datasets are never aggregated. The resulting data product, monthly averaged aggregated lifeform abundance, is generated either within the web-browser, or for download in *.csv* or *.json* format. The output data includes the number of individual samples from which each monthly average was derived, as well as a list of

component taxon groupings. Blank output component taxon groupings indicate that the originally submitted sample data did not include information in the (optional) 'Taxon Name' field.

The PLET also has a simple API (Application Programming Interface), which provides the option of bypassing the webpage interface and sending queries to the tool using the URL only. The base URL is ' 'https://www.dassh.ac.uk/lifeforms/cgi-bin/get_form.py '. The parameters are: *startdate* (YYYY-MM-

DD), *enddate* (YYYY-MM-DD), *north* (northern edge of bounding box, in decimal degrees), *south* (southern edge of bounding box, in decimal degrees), *east* (eastern edge of bounding box, in decimal degrees), *west* (western edge of bounding box, in decimal degrees), *dataset* (currently: CPR, L4_phyto, L4_zoo, SMHI, CEFAS_SmartBuoy, EA_PHYTO_2000-2017, MBA_E1_L5, MSS_phyto, MSS_zoo, SEPA_Zooplankton, SEPA_Phytoplankton or SAMS-LPO), *format* (csv, json or pretty). For example,

to retrieve results from the CPR dataset for May 1975 between 50 and 60 degrees of Latitude and -5 and 5 degrees of Longitude, and return in CSV format, the URL request is: https://www.dassh.ac.uk/lifeforms/cgi-bin/get_form.py?startdate=1975-05-01&enddate=1975-05-30&north=60&south=50&east=5&west=-5&dataset=CPR&format=CSV. Sending such URL commands via Curl or similar tools allows the PLET to be used programmatically if desired.

There are a number of options for defining the spatial domain of the lifeform data product. A rectangular extent can be manually defined by the northern, southern, western, and eastern edges of a



rectangular bounding box, by simply drawing a rectangle on the interactive map display which shows
sample locations for each data set. Similarly, a polygon shaped extent can be manually defined in 'well-
known text' (WKT) format or through the interactive map. A query specifying spatial extent by polygon

instead of bounding box can be constructed for the API by designating the parameter *wkt* instead of
*north*, *south*, *east* and *west*. A more complex area, for example a formal assessment region, can be used
by uploading a shapefile to the tool.

All integrated datasets are listed within the web interface, with full metadata. The trait look-up table can
also be accessed and downloaded. To facilitate submission of new and updated plankton data, templates

for data submission are also provided.



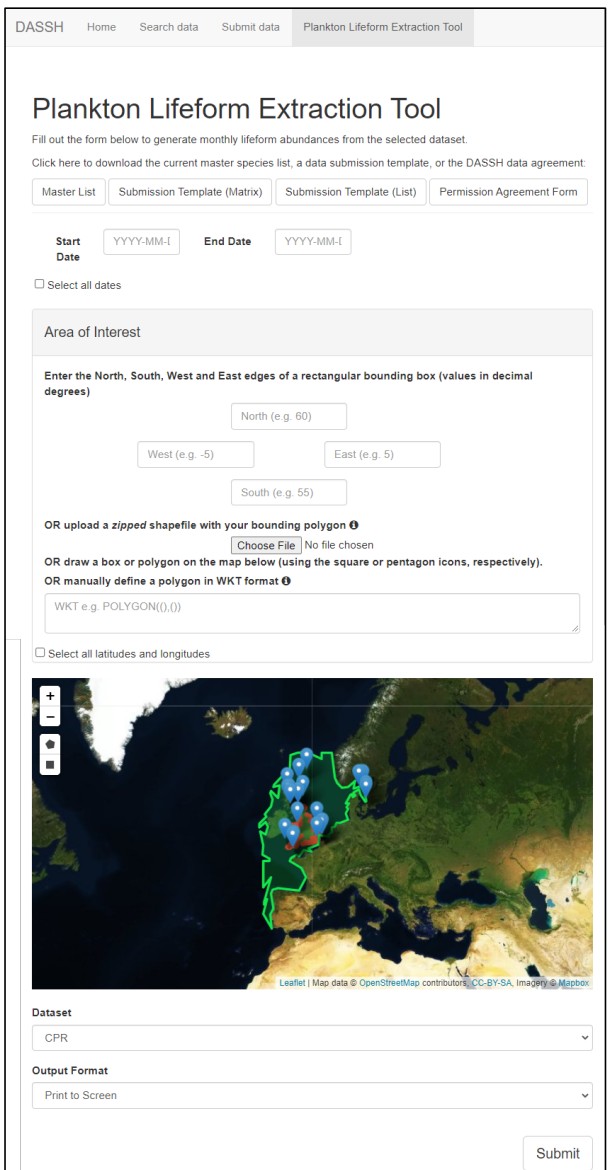

**Figure 4. The Plankton Lifeform Extraction Tool**. Screenshot of the Plankton Lifeform Extraction Tool (v1).

## 6    Lifeform outputs

The spatial and temporal patterns of plankton lifeforms, based on the data currently held in PLET, are

summarised to highlight seasonal patterns in **Figure 5** (phytoplankton) and **Figure 6** (zooplankton), and inter-annual patterns in **Figure 7** and **Figure 8**. In order to facilitate visualisation across the different lifeforms and datasets, where absolute lifeform abundances are extremely variable, within-lifeform and dataset changes are shown as standardised z-scores that indicate the difference from the overall time-series mean values (Glover et al., 2005).

Plankton abundance peaks in spring and summer are associated with nearly all plankton lifeforms, across all datasets. However, the timing, duration, and intensity of these peaks differ between lifeforms and datasets (see **Figure 5** and **Figure 6)**, in some cases partly because of spatial aggregation of the data. In the CPR dataset, which samples furthest offshore, seasonal zooplankton lifeform abundance peaks last longer than those of the phytoplankton lifeforms. The EA datasets, which represent estuarine

and coastal waters, include much shorter seasonal phytoplankton lifeform abundance peaks than the CPR for the corresponding regions, and differences in bloom timing are also evident, highlighting the small-scale spatial variability in plankton abundance. This is further evidenced in the comparison between seasonal patterns in PML's L4 station in the Channel and both the EA and CPR data aggregation for the same larger region. This heterogeneity demonstrates the added value of integrating

datasets to achieve a representative description of plankton community seasonal succession within even a relatively localised sea area, particularly where different programmes sample different subsets of the plankton community (in this case the fragile dinoflagellates being less well preserved by CPR compared to PML and EA sampling). Comparing across larger spatial scales, differences in seasonal patterns between English and Scottish waters are likely influenced by the latitudinal gradients (Fanjul et al.,

2017, 2019; Uriarte et al., 2021) as well as local hydrographic conditions (e.g.: Atlantic inflow). The Swedish stations, located in the Kattegat, show the most divergent lifeform seasonality compared to the other datasets, notably in the timing of abundance peaks (e.g.: the absence of April plankton blooms) which likely reflect their distinct oceanographic setting and the influence of Baltic Sea outflow waters.



Interannual trends in lifeform abundance can be related to changes in pressures within the marine
system (Bedford et al. (2020a) and McQuatters-Gollop et al. (2019)). Given the strong seasonal
variability, summarising plankton abundance to compare across years is non-trivial. Representative data
coverage, typically at least monthly is needed to ensure that inter-annual differences are not due to
missing samples. For example, the WFD eutrophication assessment procedure (Greenwood (2019))
requires phytoplankton data for at least 9 months of every year assessed. The Plankton Lifeform Index
(Tett et al., 2007, 2008), by looking at changes from a reference envelope defined by 3-5 years of
adequate data (i.e.: at least monthly sampling) is robust against missing samples (months) so long as
these are not biased to particular times of the year. The Pelagic Habitat Expert Group has recommended
at least monthly sampling to adequately take account of seasonal changes in the balance of plankton
lifeforms, while noting that higher temporal resolution would provide greater confidence that all
475 transient bloom events (which may last less than a month) were observed. Given the tool's robustness
against data loss, annual assessments can be made reliably when one to three months have been lost, so
long as there is no persistent bias in lost months over several years.

Despite missing months being an important consideration for annual aggregation (in some datasets in
particular, see (**Figure 4)** the interannual trends in phytoplankton (**Figure 7**) and zooplankton (**Figure
8**) lifeforms show considerable changes in lifeform abundance among years in all datasets and regions.
The longest time-series (MBA L5&E1 since 1924, CPR since 1960 and SAMS phytoplankton since
1970) capture decreases over several years followed by subsequent increases, which caution against
over-interpretation of the shorter time-series. For example, there have been decreases in all zooplankton
lifeforms at the MSS Celtic Seas station since 2013 that cannot be seen in the nearby SEPA Celtic Seas
station which only has observations from 2014 onwards; while both in the Celtic Seas area, these two
sites are characterised by very different hydrographical settings. The importance of considering both
short- and long-term changes in plankton lifeforms is discussed in detail in (Bedford et al., 2020b).

Bringing diverse datasets together to extend both spatial and temporal coverage is a key tool for
distinguishing small-scale, short-term fluctuations from larger-scale longer-term changes. For example,
Bedford *et al.*, 2020a identified regional-scale trends in lifeform changes (increasing diatom abundance
in the northern North Sea and increasing mesozooplankton abundance across almost the whole North-



West European shelf) using timeseries data from 5 different UK plankton surveys and linked some of
these changes to changing sea surface temperature. Assessment of the status of the marine pelagic
habitat (McQuatters-Gollop et al., 2019) requires linking changes to pressures (Scherer et al., 2016),

which relies on high temporal and spatial resolution good quality observations, such as climate (Bedford
*et al.*, 2020a) and eutrophication (Gowen et al., 2015; Greenwood, 2019).

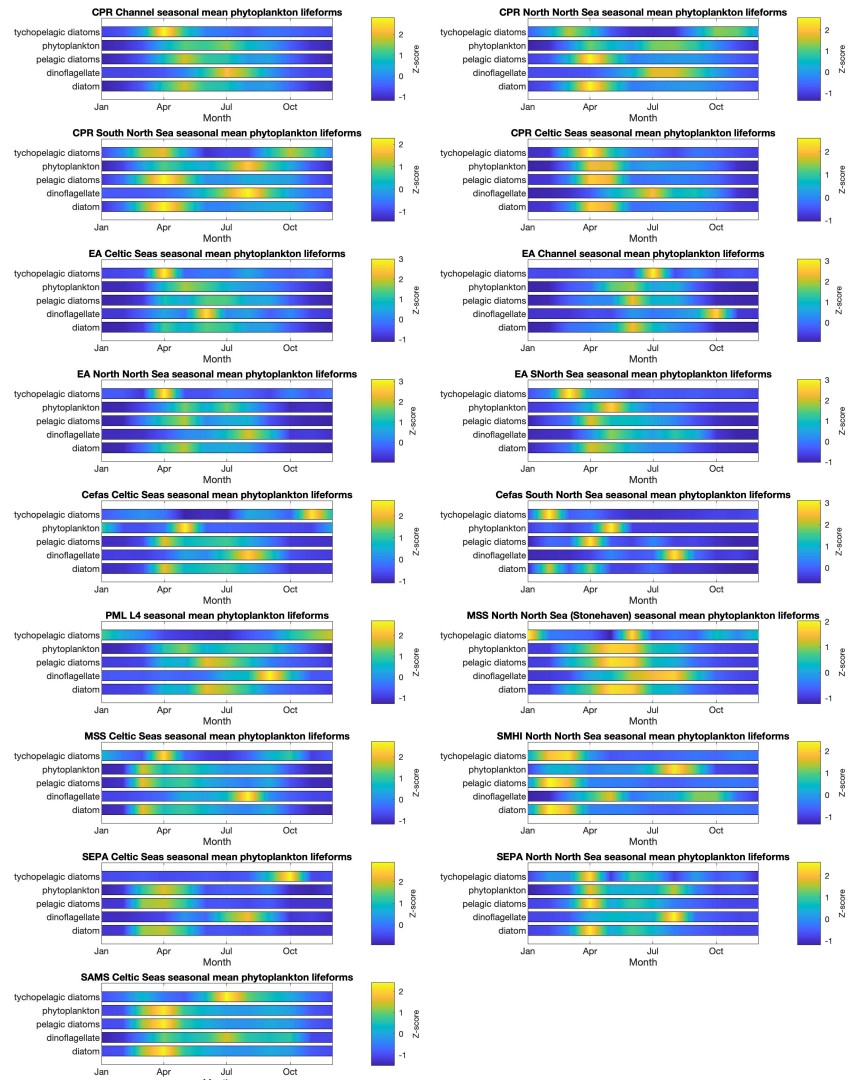

**Figure 5. Phytoplankton lifeform monthly means, by data provider and region.** Colour indicates lifeform abundance relative to the long-term mean of each lifeform within each region and dataset as standardised z-score (Glover et al., 2005): Scores of zero are equal to the long-term mean, positive scores (in green/yellow) signify values above the long-term mean and negative scores values below the long-term mean (in blue). Only those lifeforms that have been assigned a confidence level of 'high' are shown (see **Table 2** and **Table 3**). Regions are defined in **Figure 1**.

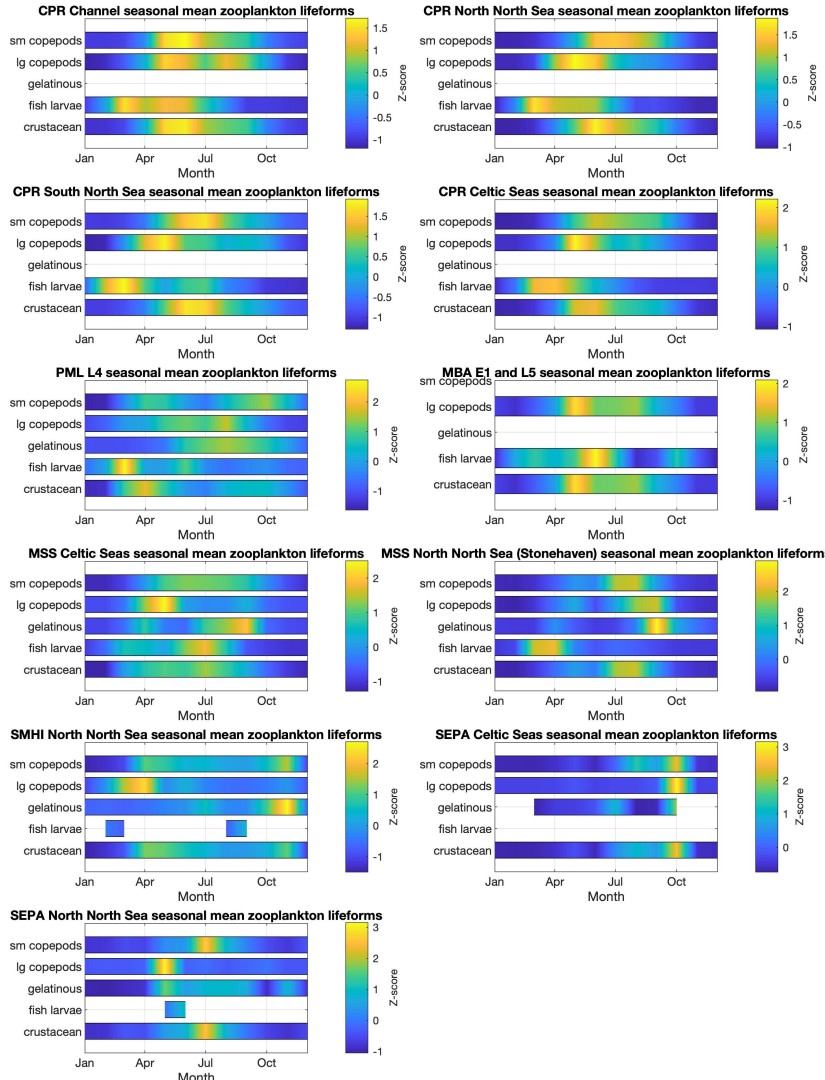

**Figure 6. Zooplankton lifeform monthly means, by data provider and region.** Colour indicates lifeform abundance relative to the long-term mean of each lifeform within each region and dataset as standardised z-score (Glover et al., 2005): Scores of zero are equal to the long-term mean, positive scores (in green/yellow) signify values above the long-term mean and negative scores values below the long-term mean (in blue). Only those lifeforms that have been assigned a confidence level of 'high' are shown (see **Table 2** and **Table 3**). Regions are defined in **Figure 1**.



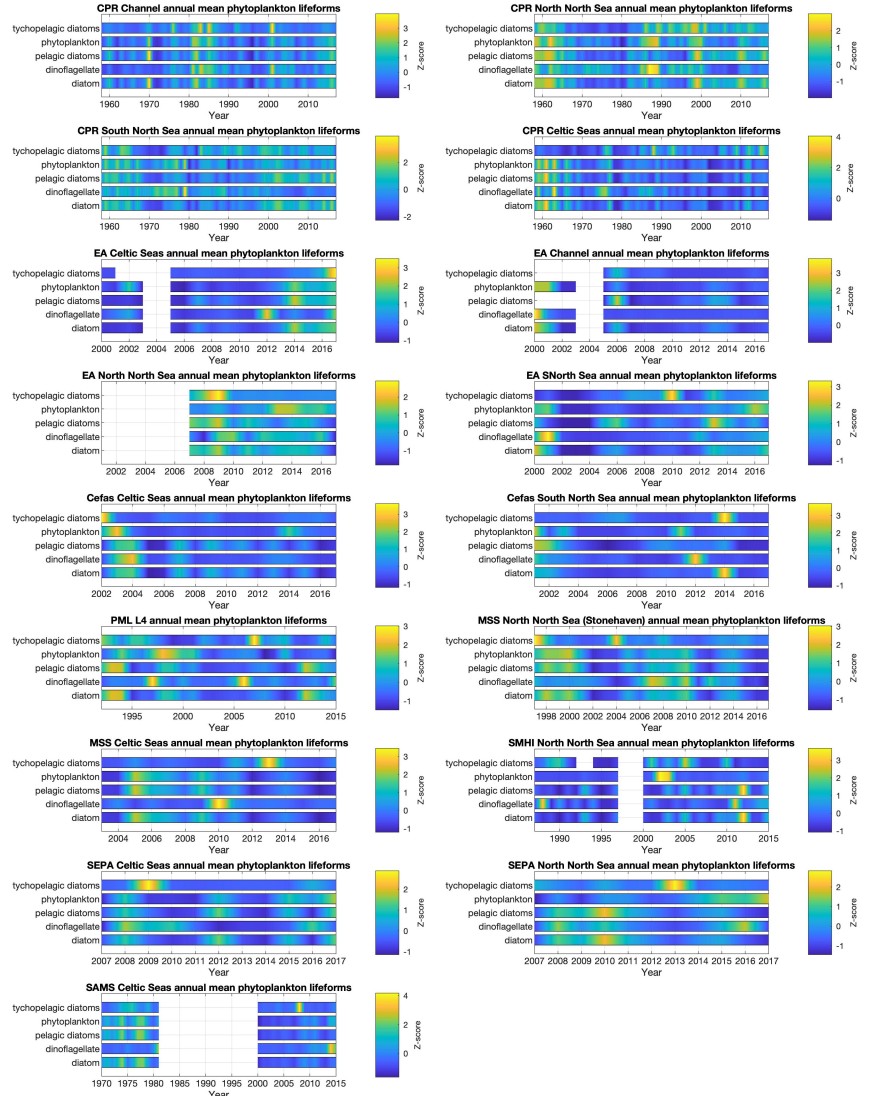

**Figure 7. Phytoplankton lifeform annual means, by data provider and region.** Colour indicates lifeform abundance relative to the long-term mean of each lifeform within each region and dataset as standardised z-score (Glover et al., 2005): Scores of zero are equal to the long-term mean, positive scores (in green/yellow) signify values above the long-term mean and negative scores values below the long-term mean (in blue). Only those lifeforms that have been assigned a confidence 515 level of 'high' are shown (see **Table 2** and **Table 3**). Regions are defined in **Figure 1**.



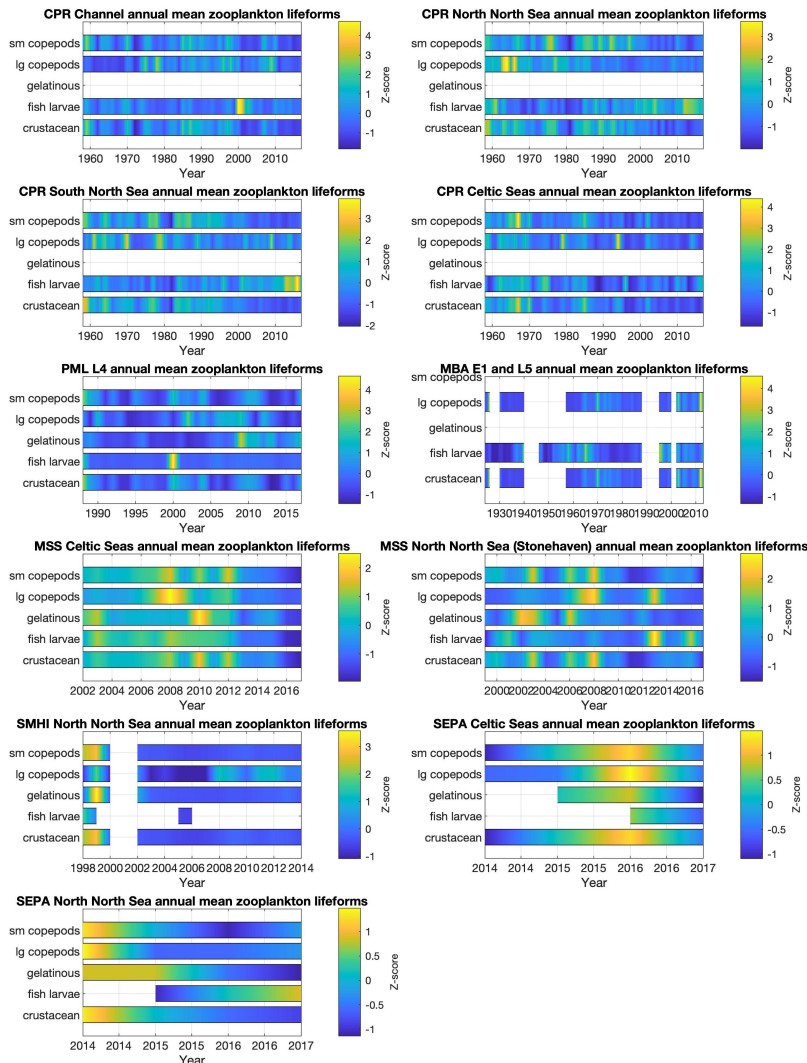

**Figure 8. Zooplankton lifeform annual means, by data provider and region.** Colour indicates lifeform abundance relative to the long-term mean of each lifeform within each region and dataset as standardised z-score (Glover et al., 2005): Scores of zero are equal to the long-term mean, positive scores (in green/yellow) signify values above the long-term mean and negative scores values below the long-term mean (in blue). Only those lifeforms that have been assigned a confidence level of 'high' are shown (see **Table 2** and **Table 3**). Regions are defined in **Figure 1**.



## 7    Discussion

Large-scale trends in the abundance of individual species are challenging to compare across multiple
time-series due to difficulties in sampling and in counting at the species level, particularly at the limits
of the geographic range of a species. Conversely, trends in bulk indices such as total zooplankton
biomass and abundance or total chlorophyll *a* concentration can miss important underlying details. Our
method to aggregate at the level of functional groups (lifeforms) provides a tractable approach to reveal
meaningful information at an intermediate level of organisation that is still ecologically relevant.

While the main available UK plankton time series are included here as well as Swedish data from the
south eastern North Sea, further extending the geographical and temporal extent of plankton time-series
held in the PLET will improve robustness of evidence underpinning pelagic habitat assessment. As a
transparent and accessible source of diverse plankton datasets, the PLET also facilitates exploration of
associated research questions in an integrated way. Observations of changes in size-based plankton
lifeform abundances (Greenwood *et al.*, 2019; McQuatters-Gollop *et al*., 2019, Bedford *et al*., 2020),
alongside methodological developments in measuring the complete plankton size spectrum (Atkinson et
al., 2021), provide an improved ability to understand what is driving changes in plankton size and
species composition across the full spatial-temporal scales of the component datasets (Schmidt *et al*.,
2020). Data from different data sources are not aggregated within PLET in order to maintain the
scientific robustness of the outputs, and incorporate a range of methodologies (units, scale, fixed-point,
transect etc.). The data is outputted in a unified way from PLET to encourage comparison and
interpretation of the changes in the plankton, if the user wishes to combine the outputs within a
specified area and time period a normalisation technique can be applied, however care needs to be taken
to ensure compatibility and coverage does not bias the combined results. There is also the flexibility to
improve confidence in low confidence groupings, and to potentially incorporate new types of plankton
data into the tool in the future, such as the use of flow cytometry data.



Time-series datasets are critical for identifying and assessing changes in the marine environment. Given the expense and effort which goes into producing and maintaining these invaluable datasets, tools which make them more widely available, transparent, and accessible to the broader user community are
needed. The PLET provides a centralised, easily accessible, source for version-controlled time-series data and metadata and is an essential component of a robust assessment process as well as a tool to support the research which is needed to underpin assessment. This includes exploring new ecologically relevant lifeform groupings and improving the understanding of lifeforms currently designated as medium and low confidence, and which will in turn feed back into the process of assessing the health of
the marine environment.

The PLET is not a static resource; it is designed to readily accept additional datasets and be updated to support future assessments as the assessment procedure continues to evolve. This is a critical step towards using multiple datasets collected with diverse methods to populate and assess a common indicator, allowing the assessment of pelagic biodiversity at a regional scale. As the tool is expanded
with additional datasets, its ability to detect change in plankton communities will increase, and the policy evidence it provides will continue to become more robust, providing decision makers with critical information to inform management measures. As pelagic habitat assessments continue to improve and adapt to the changing policy landscape (Boyes and Elliott, 2016) and to evolving plankton data availability, the PLET's flexibility will allow it to continue to underpin assessment.

**8     Data Availability and Citation**

The Plankton Lifeform Extraction Tool is hosted by the Archive for Marine Species and Habitats Data (DASSH), which is accredited as the UK Node of the Ocean Biodiversity Information System (OBIS) and through the Marine Environmental Data and Information Network (MEDIN), the UK partnership of organisations committed to improving access to UK marine data, and core-funded by the Department
for the Environment, Food and Rural Affairs (Defra) and the Scottish Government. Lifeform data products can be generated at: https://www.dassh.ac.uk/lifeforms/.



The PLET's lifeform data products are generated by applying the Plankton Lifeform Traits Master List trait look-up table (UK Pelagic Habitats Expert Group, 2020; https://www.dassh.ac.uk/doitool/data/1709) to standardised-format versions of the integrated plankton datasets (see **Table 1** for details (CPR and Johns, 2019; MBA, 2019; PML, 2019; CEFAS, 2019; EA, 2019; MSS, 2019; SMHI, 2019; SEPA, 2020; SAMS, 2020)**)**. These time-series may be updated in the future to include ongoing plankton monitoring, and more datasets may be added. Versions of several of these datasets are also available through other data repositories (e.g.: institute-specific websites provided in **Table 1**, or the Ocean Biodiversity Information System: OBIS). However, PLET provides the first centralised database for all time-series feeding into UK Marine Strategy and OSPAR biodiversity lifeform-based assessments, and importantly: a format and structure compatible with extraction of lifeform time-series. By attaching doi's to the underlying dataset versions held within PLET and the traits list, the tool provides full transparency and reproducibility for the generated lifeform outputs. Users of these datasets are encouraged to appropriately cite the data sources by means of their doi's as well as this data paper, so that usage can be more easily traced. Doing so provides evidence of data uptake, enhancing the possibility of continued funding for valuable time series and for plankton indicator development.



## 9 Acknowledgements

We are greatly indebted to the crews and scientists for collecting the component samples over the last 100 years, to the large number of taxonomic analysts for processing the samples, and to those individuals who strived to keep plankton time-series going under difficult circumstances. Funding that supports this work and the data collected has come from the European Union (EU) Grant No. 11.0661/2015/712630/SUB/ENVC.2 OSPAR, UK Natural Environment Research Council,

Grant/Award Number: NE/R002738/1 and NE/M007855/1; EMFF; Climate Linked Atlantic Sector Science, Grant/Award Number: NE/ R015953/1, DEFRA UK ME-5308 and ME-414135, NSF USA OCE-1657887, DFO CA F5955- 150026/001/HAL, NERC UK NC-R8/H12/100, Horizon 2020: 862428 Atlantic Mission, IMR Norway, DTU Aqua Denmark and the French Ministry of Environment, Energy, and the Sea (MEEM). The MSS Scottish Coastal Observatory data and analyses are funded and

maintained by the Scottish Government Schedules of Service ST05a and ST02H, MSS Stonehaven Samplers, North Atlantic Fisheries College, Shetland, Orkney Islands Harbour Council, and Isle Ewe Shellfish.

## 10 Author contribution

AMG co-ordinated the project. All authors developed the Plankton lifeform traits master list and
605 provided input on the lifeform traits and groupings. CO and GG designed the Plankton Lifeform Extraction Tool (PLET) and wrote the initial code. KP developed the PLET and performed the maintenance of the site, with support from DL and DB. DGJ, CO, AA, CW, RB, MB, MD, CG, BB, EB, KC, MM, MJ, and PT provided datasets. CO, CG and AMG prepared the manuscript with contributions from all co-authors.

## 11 Competing interests

The authors declare that they have no conflict of interest.



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



## 13 Appendices

**Table A1:** List of taxa groupings included in traits list, and datasets in which they appear. Institutes: AFBI: Agri-food and
Biosciences Institute; CEFAS: Centre for Environment, Fisheries and Aquaculture Science; EA: Environment Agency;
MBA: Marine Biological Association; MSS: Marine Scotland Science; PML: Plymouth Marine Laboratory; SEPA; Scottish
Environment Protection Agency; SMHI: Swedish Meteorological and Hydrological Institute; SAMS: Scottish Association
for Marine Science.

| AphiaID | Taxon | Institutes |
|---|---|---|
| 235747 | *Acantharea* | CEFAS |
| 235802 | *Acanthoica quattrospina* | PML; SMHI |
| 345919 | *Acartia bifilosa* | MSS; SEPA; SMHI |
| 149755 | *Acartia clausi* | MSS; PML; SEPA; SMHI |
| 346026 | *Acartia danae* | MBA |
| 234125 | *Acartia discaudata* | MSS; SEPA; SMHI |
| 346037 | *Acartia longiremis* | MBA; MSS; SEPA; SMHI |
| 346030 | *Acartia negligens* | MBA |
| 104108 | *Acartia* spp. | MBA; SEPA; SMHI |
| 345943 | *Acartia tonsa* | SEPA; SMHI |
| 149191 | *Achnanthes* | AFBI; CEFAS; EA |
| 149387 | *Achnanthes brevipes* | AFBI |
| 156533 | *Achnanthes longipes* | CEFAS; EA; PML; SEPA |
| 160542 | *Actinastrum* | CEFAS; EA |
| 160543 | *Actinastrum hantzschii* | Unallocated |
| 109717 | *Actiniscus pentasterias* | AFBI; MBA |
| 148944 | *Actinocyclus* | AFBI; EA; PML |
| 148945 | *Actinocyclus normanii* | Unallocated |
| 162770 | *Actinocyclus octonarius var. octonarius* | SMHI |
| 149654 | *Actinocyclus octonarius var. ralfsii* | MBA |
| 10194 | *Actinopterygii* | MBA; SMHI |
| 148947 | *Actinoptychus* | AFBI; CEFAS; EA; MBA; MSS; SEPA |
| 149163 | *Actinoptychus octonarius* | AFBI |
| 148948 | *Actinoptychus senarius* | AFBI; CEFAS; PML; SMHI |
| 148949 | *Actinoptychus splendens* | AFBI |
| 623576 | *Acutodesmus acuminatus* | Unallocated |
| 110069 | *Adenoides eludens* | MSS |
| 109525 | *Adenoides* spp. | MSS |
| 992758 | *Aduncodinium glandula* | Unallocated |
| 116998 | *Aequorea* spp. | MSS; SEPA |
| 104075 | *Aetideidae* | SMHI |
| 356886 | *Aetideopsis armatus* | SEPA |
| 104275 | *Aetideus armatus* | MBA; MSS; SEPA |
| 104276 | *Aetideus giesbrechti* | MBA |
| 104112 | *Aetideus* spp | MBA; SEPA |
| 135484 | *Agalma elegans* | MSS; SEPA |
| 117849 | *Aglantha digitale* | MSS; SEPA |
| 232546 | *Akashiwo sanguinea* | AFBI; CEFAS; EA; MSS; SEPA; SMHI |
| 109470 | *Alexandrium* | AFBI; CEFAS; EA; MSS; SEPA; SMHI |
| 109707 | *Alexandrium affine* | Unallocated |
| 109711 | *Alexandrium minutum* | AFBI; CEFAS; SMHI |
| 109712 | *Alexandrium ostenfeldii* | AFBI; CEFAS; EA; SMHI |
| 109713 | *Alexandrium pseudogonyaulax* | SMHI |
| 109714 | *Alexandrium tamarense* | AFBI; CEFAS; PML; SMHI |



| AphiaID | Taxon | Institutes |
|---|---|---|
| 125516 | *Ammodytidae larva* | MSS; SEPA |
| 109556 | *Amphidiniopsis* | MSS |
| 109473 | *Amphidinium* | AFBI; CEFAS; EA; MSS; SEPA; SMHI |
| 109723 | *Amphidinium carterae* | AFBI; CEFAS; EA; MSS; SEPA; SMHI |
| 109726 | *Amphidinium crassum* | AFBI; MSS; PML; SMHI |
| 109741 | *Amphidinium longum* | AFBI; CEFAS; SMHI |
| 109754 | *Amphidinium sphenoides* | AFBI; CEFAS; EA; MSS; PML; SMHI |
| 109517 | *Amphidoma* | AFBI; EA; MSS |
| 110005 | *Amphidoma caudata* | AFBI; EA; MBA; MSS; PML |
| 117178 | *Amphinema* spp. | MSS; SEPA |
| 1135 | *Amphipoda* | MBA; MSS; SEPA; SMHI |
| 149140 | *Amphiprora* | AFBI; CEFAS; EA |
| 179174 | *Amphiprora hyperborea* | MBA |
| 163647 | *Amphiprora paludosa var. paludosa* | SMHI |
| 160671 | *Amphiprora surirelloides* | PML |
| 109459 | *Amphisolenia* | AFBI; MBA |
| 149200 | *Amphora* | AFBI; CEFAS |
| 109518 | *Amylax* | AFBI; CEFAS; EA; MSS |
| 233480 | *Amylax buxus* | AFBI; CEFAS; MSS; SEPA |
| 110007 | *Amylax triacantha* | AFBI; CEFAS; EA; MSS; PML; SEPA; SMHI |
| 146585 | *Anabaena* | CEFAS; EA; SMHI |
| 177635 | *Anabaena baltica* | SMHI |
| 163489 | *Ankistrodesmus* | CEFAS; EA |
| 163490 | *Ankistrodesmus falcatus* | Unallocated |
| 608578 | *Ankistrodesmus fusiformis* | SMHI |
| 104722 | *Anomalocera patersoni* | MBA; MSS; PML; SEPA |
| 106671 | *Anomura* | SEPA; SMHI |
| 13551 | *Anthoathecata* | MSS; SEPA |
| 1292 | *Anthozoa* | PML; SEPA; SMHI |
| 248096 | *Apedinella* | AFBI; SMHI |
| 248097 | *Apedinella radians* | AFBI; CEFAS; SMHI |
| 160567 | *Aphanizomenon* | SMHI |
| 248099 | *Aphanizomenon flos-aquae* | SMHI |
| 146562 | *Aphanocapsa* | SMHI |
| 162668 | *Aphanocapsa incerta* | Unallocated |
| 146715 | *Aphanothece* | SMHI |
| 162689 | *Aphanothece minutissima* | SMHI |
| 146421 | *Appendicularia* | MSS; PML; SEPA; SMHI |
| 624607 | *Archaeperidinium minutum* | Unallocated |
| 127126 | *Arnoglossus laterna* | SMHI |
| 415082 | *Ascampbelliella* | AFBI |
| 1839 | *Ascidiacea larva* | MSS; PML; SEPA |
| 292898 | *Askenasia stellaris* | Unallocated |
| 148953 | *Asterionella* | AFBI; CEFAS; EA; MSS |
| 148954 | *Asterionella formosa* | AFBI; CEFAS; EA; MSS; SEPA; SMHI |
| 148955 | *Asterionella glacialis* | MSS |
| 149374 | *Asterionella gracillima* | Unallocated |
| 149138 | *Asterionellopsis* | AFBI; EA |
| 149139 | *Asterionellopsis glacialis* | AFBI; CEFAS; EA; MBA; PML; SEPA; SMHI |
| 149618 | *Asterionellopsis kariana* | MBA; SEPA |
| 123080 | *Asteroidea* | SEPA; SMHI |
| 162254 | *Asteromphalus* | AFBI; CEFAS; EA; MBA; MSS; SEPA |
| 162255 | *Asteromphalus flabellatus* | PML |
| 394598 | *Asteromphalus sarcophagus* | PML |
| 251745 | *Asteroplanus karianus* | AFBI; CEFAS; EA |



| AphiaID | Taxon | Institutes |
|---|---|---|
| 251744 | *Asteroplanus species* | AFBI |
| 394840 | *Attheya armata* | AFBI |
| 160520 | *Attheya decora* | AFBI; CEFAS; SMHI |
| 464444 | *Attheya longicornis* | MSS; SMHI |
| 162823 | *Attheya septentrionalis* | AFBI; CEFAS; PML; SMHI |
| 160519 | *Attheya* spp. | AFBI; CEFAS; EA; MSS; SEPA |
| 104137 | *Augaptilus* spp. | MBA |
| 149280 | *Aulacodiscus argus* | AFBI; MBA |
| 148959 | *Aulacoseira* | AFBI; SMHI |
| 148960 | *Aulacoseira ambigua* | Unallocated |
| 149096 | *Aulacoseira distans* | Unallocated |
| 148961 | *Aulacoseira granulata* | Unallocated |
| 162877 | *Aulacoseira italica* | SMHI |
| 611550 | *Aulacoseira subborealis* | Unallocated |
| 135306 | *Aurelia aurita* | MSS; SEPA |
| 135263 | *Aurelia* spp. *ephyra* | MSS; SEPA |
| 391508 | *Azadinium* | AFBI; CEFAS; EA; MSS; SMHI |
| 391509 | *Azadinium spinosum* | AFBI; SMHI |
| 149651 | *Azpeitia nodulifera* | AFBI |
| 558242 | *Bacillaria paradoxa* | Unallocated |
| 149149 | *Bacillaria paxillifer* | SEPA |
| 558243 | *Bacillaria paxillifera* | AFBI; CEFAS; EA; MBA; MSS; PML |
| 149148 | *Bacillaria* spp. | AFBI; CEFAS; EA; MSS |
| 149001 | *Bacillariales* | PML |
| 148899 | *Bacillariophyceae* | AFBI; CEFAS; EA; MSS; SEPA; SMHI |
| 149118 | *Bacteriastrum* | AFBI; CEFAS; EA; MBA; MSS; SEPA |
| 248066 | *Bacteriastrum comosum* | Unallocated |
| 164108 | *Bacteriastrum delicatulum* | AFBI; EA |
| 164110 | *Bacteriastrum furcatum* | AFBI; PML |
| 149119 | *Bacteriastrum hyalinum* | AFBI; CEFAS; EA; SMHI |
| 162927 | *Bacterosira bathyomphala* | AFBI; CEFAS |
| 292899 | *Balanion* | Unallocated |
| 427290 | *Balanion comatum* | SMHI |
| 149305 | *Bellerochea* | AFBI; CEFAS; EA; MSS; SEPA |
| 447730 | *Bellerochea horologicalis* | AFBI; CEFAS |
| 149306 | *Bellerochea malleus* | MBA |
| 106331 | *Beroe* | SEPA |
| 106358 | *Beroe cucumis* | MSS; SEPA |
| 708294 | *Bicosoeca campanulata* | Unallocated |
| 104002 | *Bicosoeca petiolata* | Unallocated |
| 149655 | *Biddulphia alternans* | EA; MBA |
| 162952 | *Biddulphia biddulphiana* | AFBI; MBA |
| 149324 | *Biddulphia rhombus* | AFBI; CEFAS |
| 148967 | *Biddulphia* spp. | AFBI; EA; MSS |
| 148965 | *Biddulphiales* | AFBI; CEFAS |
| 105 | *Bivalvia* | MBA; MSS; PML; SEPA; SMHI |
| 163736 | *Bleakeleya notata* | AFBI; MBA |
| 110178 | *Blepharocysta paulsenii* | AFBI; MBA |
| 22556 | *Bodonidae* | PML |
| 110067 | *Boreadinium pisiforme* | MSS |
| 106265 | *Bosmina* | MSS; SEPA; SMHI |
| 106271 | *Bosmina coregoni* | SMHI |
| 248104 | *Botryococcus braunii* | Unallocated |
| 117328 | *Bougainvillia muscus* | SEPA |
| 117015 | *Bougainvillia* spp. | MSS; SEPA |





| AphiaID | Taxon | Institutes |
|---|---|---|
| 235922 | *Braarudosphaera bigelowii* | PML |
| 106673 | *Brachyura* | SEPA; SMHI |
| 104115 | *Bradyidius* | SEPA |
| 104902 | *Branchiostoma* | PML; SEPA; SMHI |
| 104906 | *Branchiostoma lanceolatum* | MBA; SEPA; SMHI |
| 149136 | *Brockmanniella* | AFBI; EA |
| 149137 | *Brockmanniella brockmannii* | AFBI; CEFAS; PML |
| 231802 | *Bysmatrum* | AFBI |
| 1100 | *Calanoida C1-6* | MBA; MSS; SEPA; SMHI |
| 104462 | *Calanoides carinatus* | MBA; PML; SEPA |
| 104464 | *Calanus finmarchicus* | MBA; MSS; SEPA; SMHI |
| 104465 | *Calanus glacialis* | MBA; SMHI |
| 104466 | *Calanus helgolandicus* | MBA; MSS; PML; SEPA |
| 104467 | *Calanus hyperboreus* | MBA; SEPA; SMHI |
| 104152 | *Calanus I-IV* | MBA; MSS; SEPA; SMHI |
| 555889 | *Calciosolenia brasiliensis* | PML |
| 135513 | *Caligidae C1-6* | MSS; SEPA |
| 135566 | *Caligus* spp. *C1-6* | MSS; SEPA |
| 105559 | *Calliacantha longicaudata* | SMHI |
| 105561 | *Calliacantha natans* | SMHI |
| 125522 | *Callionymidae* | SMHI |
| 104193 | *Calocalanus* spp. | MBA; MSS; PML; SEPA |
| 149488 | *Caloneis* | AFBI; MSS |
| 235828 | *Calyptrosphaera* | PML |
| 149616 | *Campylodiscus* | AFBI |
| 178059 | *Campyloneis* | AFBI |
| 149357 | *Campylosira cymbelliformis* | AFBI; MBA |
| 104474 | *Candacia armata* | MBA; MSS; PML; SEPA |
| 104475 | *Candacia bipinnata* | MBA |
| 220915 | *Candacia bispinosa* | MBA |
| 104476 | *Candacia curta* | MBA |
| 104478 | *Candacia ethiopica* | MBA |
| 104479 | *Candacia giesbrechti* | MBA |
| 104481 | *Candacia longimana* | MBA |
| 104483 | *Candacia pachydactyla* | MBA |
| 220914 | *Candacia simplex* | MBA |
| 104157 | *Candacia* spp. | MBA; SEPA; SMHI |
| 104485 | *Candacia tenuimana* | MBA |
| 104486 | *Candacia varicans* | MBA |
| 101361 | *Caprellidae* | SMHI |
| 196121 | *Caprellidea* | MBA |
| 106674 | *Caridea* | SMHI |
| 341285 | *Centrales* | AFBI; CEFAS |
| 109526 | *Centrodinium* | AFBI; MBA |
| 104491 | *Centropages bradyi* | MBA |
| 104494 | *Centropages chierchiae* | MBA; SEPA |
| 104495 | *Centropages furcatus* | MBA |
| 104496 | *Centropages hamatus* | MBA; MSS; SEPA; SMHI |
| 104159 | *Centropages* spp. | MBA; SEPA; SMHI |
| 104499 | *Centropages typicus* | MBA; MSS; SEPA; SMHI |
| 104500 | *Centropages violaceus* | MBA |
| 416184 | *Cephalobrachia* spp. | MBA |
| 1824 | *Cephalochordata* | MSS |
| 11707 | *Cephalopoda larvae* | MBA; PML; SEPA |
| 149619 | *Cerataulina pelagica* | AFBI; CEFAS; EA; MBA; MSS; PML; SEPA; SMHI |



| AphiaID | Taxon | Institutes |
|---|---|---|
| 149236 | *Cerataulina* spp. | AFBI; CEFAS; EA; MSS |
| 109506 | *Ceratium* | AFBI; CEFAS; EA; MSS; SMHI |
| 156509 | *Ceratium arcticum* | MBA; MSS |
| 109929 | *Ceratium arietinum* | EA; MBA; MSS; SEPA |
| 844439 | *Ceratium articum* | Unallocated |
| 109930 | *Ceratium azoricum* | CEFAS; MBA; MSS; SEPA |
| 109931 | *Ceratium belone* | AFBI; MBA |
| 109932 | *Ceratium breve* | AFBI; MBA |
| 196820 | *Ceratium bucephalum* | MBA; SMHI |
| 109934 | *Ceratium buceros* | MBA |
| 109935 | *Ceratium candelabrum* | AFBI; MBA |
| 109936 | *Ceratium carriense* | MBA |
| 109939 | *Ceratium compressum* | CEFAS; EA; MBA; MSS; SEPA |
| 109940 | *Ceratium concilians* | Unallocated |
| 109941 | *Ceratium contortum* | MBA |
| 109943 | *Ceratium declinatum* | MBA |
| 109947 | *Ceratium extensum* | AFBI; MBA |
| 109948 | *Ceratium falcatiforme* | MBA |
| 109949 | *Ceratium falcatum* | MBA |
| 109950 | *Ceratium furca* | EA; MBA; MSS; SEPA; SMHI |
| 109951 | *Ceratium fusus* | AFBI; EA; MBA; MSS; SEPA; SMHI |
| 109952 | *Ceratium geniculatum* | MBA |
| 109953 | *Ceratium gibberum* | AFBI; MBA |
| 109955 | *Ceratium hexacanthum* | AFBI; CEFAS; EA; MBA; MSS; PML; SEPA |
| 109956 | *Ceratium horridum* | AFBI; EA; MBA; MSS; SEPA; SMHI |
| 422708 | *Ceratium horridum var. buceros* | Unallocated |
| 109958 | *Ceratium inflatum* | MBA |
| 109960 | *Ceratium karstenii* | MBA |
| 109961 | *Ceratium kofoidii* | MBA |
| 109962 | *Ceratium limulus* | MBA |
| 109963 | *Ceratium lineatum* | EA; MBA; MSS; SEPA; SMHI |
| 109964 | *Ceratium longipes* | EA; MBA; MSS; PML; SEPA; SMHI |
| 109965 | *Ceratium longirostrum* | AFBI; MBA |
| 196822 | *Ceratium lunula* | MBA |
| 109967 | *Ceratium macroceros* | EA; MBA; MSS; SEPA; SMHI |
| 109968 | *Ceratium massiliense* | AFBI; MBA |
| 109969 | *Ceratium minutum* | CEFAS; EA; MBA; MSS; SEPA |
| 109971 | *Ceratium pavillardii* | MBA |
| 109972 | *Ceratium pentagonum* | MBA |
| 196824 | *Ceratium platycorne* | AFBI; CEFAS; MBA; MSS; SEPA |
| 109974 | *Ceratium pulchellum* | MBA |
| 109975 | *Ceratium ranipes* | MBA |
| 109977 | *Ceratium setaceum* | AFBI; CEFAS; EA; MBA; MSS; SEPA |
| 109979 | *Ceratium symmetricum* | AFBI; EA |
| 109980 | *Ceratium teres* | AFBI; MBA |
| 109981 | *Ceratium trichoceros* | AFBI; MBA |
| 109982 | *Ceratium tripos* | AFBI; EA; MBA; MSS; SEPA; SMHI |
| 109983 | *Ceratium vultur* | MBA |
| 109507 | *Ceratocorys* spp. | AFBI; MBA |
| 163932 | *Ceratoneis closterium* | Unallocated |
| 1361 | *Ceriantharia* | SEPA |
| 100782 | *Cerianthus* spp. | MSS; SEPA |
| 148985 | *Chaetoceros* | AFBI; CEFAS; EA; MBA; MSS; PML; SEPA; SMHI |
| 370366 | *Chaetoceros (Hyalochaete)* | AFBI; MBA; SEPA |
| 370367 | *Chaetoceros (Phaeoceros)* | AFBI; EA; SEPA |




| AphiaID | Taxon | Institutes |
|---|---|---|
| 149241 | *Chaetoceros affinis* | AFBI; CEFAS; PML; SMHI |
| 149292 | *Chaetoceros anastomosans* | AFBI; PML; SMHI |
| 162998 | *Chaetoceros anastomosans var. externa* | SMHI |
| 149288 | *Chaetoceros atlanticus* | AFBI |
| 149124 | *Chaetoceros borealis* | AFBI; CEFAS; PML; SMHI |
| 149291 | *Chaetoceros brevis* | AFBI; CEFAS; PML |
| 163013 | *Chaetoceros calcitrans* | AFBI; SMHI |
| 149297 | *Chaetoceros ceratosporus* | AFBI; CEFAS |
| 163016 | *Chaetoceros ceratosporus var. ceratosporus* | SMHI |
| 163019 | *Chaetoceros circinalis* | AFBI; SMHI |
| 149129 | *Chaetoceros compressus* | AFBI; CEFAS; PML; SMHI |
| 156607 | *Chaetoceros concavicornis* | AFBI; CEFAS; PML; SMHI |
| 156609 | *Chaetoceros constrictus* | AFBI; CEFAS; SMHI |
| 149623 | *Chaetoceros contortus* | AFBI; SMHI |
| 156611 | *Chaetoceros convolutus* | AFBI; PML; SMHI |
| 163026 | *Chaetoceros coronatus* | AFBI; SMHI |
| 149289 | *Chaetoceros costatus* | AFBI; PML; SMHI |
| 149171 | *Chaetoceros crinitus* | AFBI; CEFAS; SMHI |
| 465389 | *Chaetoceros criophilus* | AFBI |
| 149221 | *Chaetoceros curvisetus* | AFBI; CEFAS; PML; SMHI |
| 149120 | *Chaetoceros danicus* | AFBI; CEFAS; EA; MSS; PML; SMHI |
| 149219 | *Chaetoceros debilis* | AFBI; CEFAS; PML; SMHI |
| 149126 | *Chaetoceros decipiens* | AFBI; CEFAS; PML; SMHI |
| 149121 | *Chaetoceros densus* | AFBI; CEFAS; PML; SMHI |
| 149128 | *Chaetoceros diadema* | AFBI; CEFAS; SMHI |
| 149122 | *Chaetoceros didymus* | AFBI; CEFAS; PML; SMHI |
| 163048 | *Chaetoceros didymus var. didymus* | SMHI |
| 163054 | *Chaetoceros didymus var. protuberans* | SMHI |
| 163056 | *Chaetoceros difficilis* | Unallocated |
| 160521 | *Chaetoceros eibenii* | AFBI; CEFAS; PML |
| 160522 | *Chaetoceros externus* | PML |
| 178194 | *Chaetoceros filiformis* | AFBI; PML |
| 149173 | *Chaetoceros fragilis* | AFBI; PML |
| 465524 | *Chaetoceros gracilis* | SMHI |
| 163063 | *Chaetoceros holsaticus* | Unallocated |
| 163080 | *Chaetoceros impressus* | SMHI |
| 156613 | *Chaetoceros ingolfianus* | SMHI |
| 149228 | *Chaetoceros laciniosus* | AFBI; CEFAS; PML; SMHI |
| 160523 | *Chaetoceros lauderi* | AFBI; PML |
| 156617 | *Chaetoceros lorenzianus* | AFBI; CEFAS; EA; SMHI |
| 163089 | *Chaetoceros minimus* | SMHI |
| 148986 | *Chaetoceros mitra* | Unallocated |
| 163098 | *Chaetoceros muelleri* | SMHI |
| 418510 | *Chaetoceros neogracile* | AFBI |
| 178185 | *Chaetoceros peruvianus* | AFBI; CEFAS; MSS; PML |
| 163055 | *Chaetoceros protuberans* | AFBI; PML |
| 163109 | *Chaetoceros pseudobrevis* | AFBI; SMHI |
| 149222 | *Chaetoceros pseudocrinitus* | AFBI; SMHI |
| 178229 | *Chaetoceros pseudocurvisetus* | AFBI |
| 163112 | *Chaetoceros radicans* | AFBI; PML |
| 621740 | *Chaetoceros salsugineum* | Unallocated |
| 163118 | *Chaetoceros seiracanthus* | AFBI; CEFAS; SMHI |
| 149127 | *Chaetoceros similis* | AFBI; CEFAS; PML; SMHI |
| 149294 | *Chaetoceros simplex* | AFBI; PML; SMHI |
| 149123 | *Chaetoceros socialis* | AFBI; CEFAS; EA; MSS; PML; SEPA; SMHI |





| AphiaID | Taxon | Institutes |
|---------|-------|------------|
| 163126 | *Chaetoceros socialis f. radians* | SMHI |
| 163124 | *Chaetoceros socialis f. socialis* | SMHI |
| 156621 | *Chaetoceros subtilis* | AFBI; CEFAS; SMHI |
| 163131 | *Chaetoceros subtilis var. subtilis* | SMHI |
| 156623 | *Chaetoceros tenuissimus* | AFBI; CEFAS; SMHI |
| 149125 | *Chaetoceros teres* | AFBI; CEFAS; PML; SMHI |
| 163137 | *Chaetoceros throndsenii* | AFBI; CEFAS; SMHI |
| 163150 | *Chaetoceros throndsenii var. throndsenii* | SMHI |
| 163161 | *Chaetoceros tortissimus* | AFBI; PML |
| 160524 | *Chaetoceros wighamii* | AFBI; PML; SMHI |
| 156625 | *Chaetoceros willei* | PML |
| 2081 | *Chaetognatha* | MBA; PML; SEPA; SMHI |
| 233776 | *Chattonella* | AFBI; CEFAS; EA; SMHI |
| 547444 | *Chattonella marina var. antiqua* | Unallocated |
| 178587 | *Chlamydomonas coccoides* | Unallocated |
| 573809 | *Chlamydomonas quadrilobata* | Unallocated |
| 160576 | *Chlorella* | Unallocated |
| 532029 | *Chlorella vulgaris* | EA |
| 17666 | *Chlorodendrales* | SMHI |
| 178734 | *Chlorogonium* | MSS |
| 600769 | *Chloromonas* | Unallocated |
| 802 | *Chlorophyceae* | AFBI; EA |
| 580116 | *Choanoflagellatea* | AFBI; CEFAS; MSS; PML |
| 25 | *Choanoflagellida* | SMHI |
| 341353 | *Choreotrichida* | SEPA |
| 249689 | *Chromulina* | Unallocated |
| 106281 | *Chroomonas* | SMHI |
| 115090 | *Chrysochromulina* | CEFAS; MSS; SMHI |
| 115116 | *Chrysochromulina ericina* | SMHI |
| 115119 | *Chrysochromulina hirta* | SMHI |
| 115126 | *Chrysochromulina parkeae* | Unallocated |
| 571998 | *Chrysochromulina parva* | Unallocated |
| 115127 | *Chrysochromulina polylepis* | SMHI |
| 146230 | *Chrysophyceae* | AFBI; EA |
| 125741 | *Ciliata* | Unallocated |
| 11 | *Ciliophora* | AFBI; CEFAS; EA; MSS; PML; SEPA; SMHI |
| 1082 | *Cirripedia* | MBA; MSS; PML; SEPA; SMHI |
| 1082 | *Cirripede cyprid* | MBA; MSS; PML; SEPA; SMHI |
| 1082 | *Cirripede nauplii* | MBA; MSS; PML; SEPA; SMHI |
| 1076 | *Cladocera* | SEPA; SMHI |
| 109509 | *Cladopyxis* | AFBI; MBA |
| 233351 | *Cladopyxis claytonii* | SMHI |
| 233352 | *Cladopyxis setifera* | SMHI |
| 128567 | *Clausidiidae* | SEPA |
| 104082 | *Clausocalanidae* | SMHI |
| 104502 | *Clausocalanus arcuicornis* | SMHI |
| 104161 | *Clausocalanus* spp. | MBA; MSS; PML; SEPA; SMHI |
| 196804 | *Climacodium frauenfeldianum* | MBA |
| 248081 | *Climacosphenia* | CEFAS |
| 137751 | *Clio* spp. | MBA; SEPA |
| 137793 | *Clione* | MBA; PML; SEPA |
| 139178 | *Clione limacina* | MBA; MSS; SEPA |
| 162725 | *Closterium* | EA |
| 577666 | *Closterium moniliferum* | Unallocated |
| 125464 | *Clupeidae larva* | MSS; SEPA |



| AphiaID | Taxon | Institutes |
|---|---|---|
| 115273 | *Clytemnestra* | SEPA |
| 587514 | *Clytemnestrinae* | SEPA |
| 117368 | *Clytia hemisphaerica* | MSS; SEPA |
| 1267 | *Cnidaria* | SEPA; SMHI |
| 178597 | *Coccolithaceae* | Unallocated |
| 235993 | *Coccolithophorid* | AFBI; EA |
| 178600 | *Coccolithus pelagicus* | PML |
| 555900 | *Coccolithus pelagicus f. hyalinus* | PML |
| 148990 | *Cocconeis placentula* | Unallocated |
| 179573 | *Cocconeis pseudomarginata* | Unallocated |
| 149376 | *Cocconeis scutellum* | CEFAS; SMHI |
| 163880 | *Cocconeis scutellum var. scutellum* | CEFAS; SMHI |
| 148989 | *Cocconeis* sp | AFBI; CEFAS; SMHI |
| 626348 | *Cochlearisigma falcatum* | Unallocated |
| 109474 | *Cochlodinium* | AFBI; CEFAS; EA; MSS; PML; SEPA; SMHI |
| 232650 | *Cochlodinium polykrikoides* | Unallocated |
| 109773 | *Cochlodinium pupa* | SMHI |
| 565154 | *Codonium proliferum* | SEPA |
| 160550 | *Coelastrum* | EA |
| 163975 | *Coelastrum microporum* | Unallocated |
| 152230 | *Coelenterata* | MSS |
| 567048 | *Commation* | SMHI |
| 623754 | *Conticribra weissflogii* | AFBI; CEFAS |
| 109534 | *Coolia* | AFBI; CEFAS; SEPA |
| 1080 | *Copepoda* C1-6 | MSS; SEPA; SMHI |
| 128721 | *Copilia* spp. | MBA |
| 149109 | *Corethron* | AFBI; CEFAS; EA; MSS |
| 957579 | *Corethron criophilum* | AFBI |
| 179596 | *Corethron hystrix* | AFBI; CEFAS; EA; MBA; MSS |
| 341496 | *Corethron pennatum* | AFBI; CEFAS; EA; PML; SEPA |
| 235934 | *Coronosphaera* | PML |
| 128569 | *Corycaeidae* | MSS; SEPA; SMHI |
| 128800 | *Corycaeus speciosus* | MBA |
| 128634 | *Corycaeus* spp. | MBA; SEPA; SMHI |
| 162519 | *Corymbellus aureus* | CEFAS; PML |
| 117452 | *Corymorpha nutans* | MSS; SEPA |
| 151860 | *Coryne eximia* | SEPA |
| 107277 | *Corystes cassivelaunus* | SEPA |
| 109527 | *Corythodinium* | AFBI; MBA; MSS |
| 110073 | *Corythrodinium diploconus* | Unallocated |
| 149110 | *Corythron criophilum* | Unallocated |
| 148900 | *Coscinodiscophyceae* | MSS; SMHI |
| 148917 | *Coscinodiscus* | AFBI; CEFAS; EA; MBA; MSS; SEPA; SMHI |
| 149274 | *Coscinodiscus asteromphalus* | PML |
| 149159 | *Coscinodiscus centralis* | AFBI; CEFAS; PML; SMHI |
| 148991 | *Coscinodiscus commutatus* | AFBI; CEFAS; SMHI |
| 148992 | *Coscinodiscus concinnus* | AFBI; CEFAS; EA; MBA; PML; SMHI |
| 149271 | *Coscinodiscus granii* | AFBI; CEFAS; EA; PML; SMHI |
| 149307 | *Coscinodiscus pavillardii* | AFBI |
| 149158 | *Coscinodiscus radiatus* | AFBI; CEFAS; EA; PML; SMHI |
| 156632 | *Coscinodiscus wailesii* | AFBI; CEFAS; EA; MBA; PML; SMHI |
| 478557 | *Cosmarium* | Unallocated |
| 117747 | *Cosmetira pilosella* | SEPA |
| 178617 | *Crucigenia species* | CEFAS; EA; MSS; SEPA |
| 178619 | *Crucigenia tetrapedia* | EA |



| AphiaID | Taxon | Institutes |
|---|---|---|
| 1066 | *Crustacea* | SEPA |
| 118011 | *Cryothecomonas* | SMHI |
| 118047 | *Cryothecomonas scybalophora* | SMHI |
| 109998 | *Crypthecodinium cohnii* | Unallocated |
| 17644 | *Cryptomonadaceae* | PML |
| 17640 | *Cryptomonadales* | CEFAS; SMHI |
| 106282 | *Cryptomonas* | CEFAS; EA; SMHI |
| 238840 | *Cryptomonas curvata* | Unallocated |
| 155555 | *Cryptomonas erosa* | Unallocated |
| 248112 | *Cryptomonas marssonii* | Unallocated |
| 155554 | *Cryptomonas ovata* | Unallocated |
| 17639 | *Cryptophyceae* | AFBI; CEFAS; EA; MSS; SMHI |
| 17638 | *Cryptophyta* | Unallocated |
| 104162 | *Ctenocalanus* spp. | MBA; SEPA |
| 104510 | *Ctenocalanus vanus* | MBA; MSS; PML; SEPA |
| 1248 | *Ctenophora* | MSS; SEPA |
| 1137 | *Cumacea* | MBA; MSS; PML; SEPA |
| 135301 | *Cyanea capillata* | SEPA |
| 135302 | *Cyanea lamarckii* | MSS; SEPA |
| 135259 | *Cyanea* spp. | MSS; SEPA |
| 146537 | *Cyanobacteria* | AFBI; CEFAS; EA; MSS; SEPA; SMHI |
| 177108 | *Cyanodictyon* | SMHI |
| 357973 | *Cyclopinoides longicornis* | MBA |
| 1101 | *Cyclopoida* | MSS; SEPA; SMHI |
| 148996 | *Cyclostephanos dubius* | Unallocated |
| 148905 | *Cyclotella* | AFBI; CEFAS; EA; SMHI |
| 148907 | *Cyclotella atomus* | EA |
| 148908 | *Cyclotella choctawhatcheeana* | Unallocated |
| 148998 | *Cyclotella cryptica* | Unallocated |
| 163197 | *Cyclotella hakanssoniae* | SMHI |
| 148909 | *Cyclotella meneghiniana* | Unallocated |
| 149000 | *Cyclotella radiosa* | Unallocated |
| 148906 | *Cyclotella scaldensis* | Unallocated |
| 148911 | *Cyclotella striata* | Unallocated |
| 292893 | *Cyclotrichiida* | SEPA |
| 149004 | *Cylindrotheca closterium* | AFBI; CEFAS; EA; MBA; MSS; PML; SEPA; SMHI |
| 149570 | *Cylindrotheca gracilis* | AFBI; CEFAS; MSS; SEPA |
| 149003 | *Cylindrotheca* spp. | AFBI; CEFAS; EA; MSS |
| 149008 | *Cymatopleura solea* | Unallocated |
| 149012 | *Cymatosira belgica* | AFBI; CEFAS |
| 179688 | *Cymatosira lorenziana* | CEFAS; MSS |
| 134527 | *Cymbomonas* | SMHI |
| 134545 | *Cymbomonas tetramitiformis* | SMHI |
| 146142 | *Cyphonautes* | MBA; MSS; PML; SEPA; SMHI |
| 196830 | *Cystodinium* | AFBI; MBA |
| 149309 | *Dactyliosolen* | AFBI; CEFAS; EA; SEPA |
| 179785 | *Dactyliosolen antarcticus* | AFBI; CEFAS; EA; MBA; MSS; SEPA |
| 179786 | *Dactyliosolen blavyanus* | AFBI; CEFAS; EA; PML; SMHI |
| 149310 | *Dactyliosolen fragilissimus* | AFBI; CEFAS; EA; MBA; MSS; PML; SEPA; SMHI |
| 248064 | *Dactyliosolen phuketensis* | CEFAS; SMHI |
| 771485 | *Daphnia* | SMHI |
| 341300 | *Daturella* | SMHI |
| 1130 | *Decapoda larvae* | MBA; MSS; PML; SEPA; SMHI |
| 149179 | *Delphineis* | AFBI; CEFAS; EA; MSS; PML |
| 621926 | *Delphineis minutissima* | Unallocated |





| AphiaID | Taxon | Institutes |
|---|---|---|
| 149180 | *Delphineis surirella* | AFBI |
| 162724 | *Desmidiaceae* | SEPA |
| 886288 | *Desmidiales* | CEFAS; EA |
| 249711 | *Desmodesmus* | CEFAS; EA; MSS; SEPA |
| 612499 | *Desmodesmus armatus* | Unallocated |
| 576237 | *Desmodesmus communis* | Unallocated |
| 612502 | *Desmodesmus dispar* | Unallocated |
| 624737 | *Desmodesmus granulatus* | Unallocated |
| 149286 | *Detonula confervacea* | AFBI; CEFAS; MBA; MSS; SEPA; SMHI |
| 149647 | *Detonula pumila* | AFBI; CEFAS; EA; MBA; MSS; PML; SEPA; SMHI |
| 149285 | *Detonula* spp. | AFBI; CEFAS; EA; MSS |
| 104167 | *Diaixis* | SEPA |
| 104521 | *Diaixis hibernica* | MBA; MSS; PML; SEPA |
| 104522 | *Diaixis pygmaea* | MBA; SEPA |
| 247924 | *Diaptomus* spp. | MSS |
| 149013 | *Diatoma* | Unallocated |
| 160092 | *Diatoma tenue* | CEFAS; EA |
| 149014 | *Diatoma tenuis* | AFBI; SMHI |
| 149347 | *Diatoma vulgare* | Unallocated |
| 232110 | *Dicroerisma psilonereiella* | SMHI |
| 231762 | *Dicroerisma* spp | MSS |
| 157258 | *Dictyocha* | AFBI; CEFAS; EA; SMHI |
| 375788 | *Dictyocha crux* | AFBI |
| 157463 | *Dictyocha fibula* | AFBI; CEFAS; EA; MSS; PML; SEPA; SMHI |
| 157260 | *Dictyocha speculum* | AFBI; CEFAS; EA; MSS; PML; SEPA; SMHI |
| 157256 | *Dictyochales* | CEFAS |
| 146232 | *Dictyochophyceae* | AFBI; EA; MSS; SEPA |
| 178623 | *Dictyosphaerium* | EA |
| 178625 | *Dictyosphaerium ehrenbergianum* | Unallocated |
| 341301 | *Didinium* | SMHI |
| 163233 | *Dimeregramma* | Unallocated |
| 157240 | *Dinobryon* | AFBI; CEFAS; EA; MSS; PML; SEPA; SMHI |
| 160552 | *Dinobryon balticum* | SMHI |
| 157248 | *Dinobryon divergens* | CEFAS; SMHI |
| 160553 | *Dinobryon faculiferum* | CEFAS; SMHI |
| 157252 | *Dinobryon sertularia* | EA |
| 19542 | *Dinophyceae* | AFBI; CEFAS; EA; MSS; SEPA; SMHI |
| 109406 | *Dinophysiaceae* | EA |
| 109462 | *Dinophysis* | AFBI; CEFAS; EA; MBA; MSS; PML; SEPA; SMHI |
| 109603 | *Dinophysis acuminata* | AFBI; CEFAS; EA; MSS; PML; SEPA; SMHI |
| 109604 | *Dinophysis acuta* | AFBI; CEFAS; EA; MSS; PML; SEPA; SMHI |
| 232155 | *Dinophysis borealis* | MSS |
| 109612 | *Dinophysis caudata* | AFBI; CEFAS; EA; MSS; SEPA |
| 109616 | *Dinophysis dens* | AFBI; CEFAS; EA; MSS; SEPA; SMHI |
| 109624 | *Dinophysis fortii* | AFBI; CEFAS; MSS; SEPA |
| 109627 | *Dinophysis hastata* | AFBI; CEFAS; EA; MSS; SEPA; SMHI |
| 232496 | *Dinophysis nasuta* | AFBI; CEFAS; MSS; PML; SEPA |
| 109637 | *Dinophysis norvegica* | AFBI; CEFAS; EA; MSS; SEPA; SMHI |
| 109638 | *Dinophysis odiosa* | AFBI; CEFAS; EA; MSS; SEPA; SMHI |
| 646201 | *Dinophysis ovum* | AFBI; EA; MSS; SEPA |
| 109649 | *Dinophysis pulchella* | AFBI; CEFAS; EA; MSS; SEPA |
| 109651 | *Dinophysis punctata* | AFBI; CEFAS; EA; MSS; SEPA |
| 162793 | *Dinophysis rotundata* | AFBI; EA; MSS |
| 232261 | *Dinophysis sacculus* | AFBI; CEFAS; EA; MSS; PML; SEPA |
| 109659 | *Dinophysis skagii* | AFBI; MSS; SEPA |





| AphiaID | Taxon | Institutes |
|---|---|---|
| 109662 | *Dinophysis tripos* | AFBI; CEFAS; EA; MSS; PML; SEPA; SMHI |
| 135338 | *Diphyidae* | MSS; SEPA |
| 149018 | *Diploneis* | AFBI; CEFAS; EA; MBA; SMHI |
| 149194 | *Diploneis bombus* | Unallocated |
| 149396 | *Diploneis crabro* | PML |
| 149195 | *Diploneis didyma* | Unallocated |
| 149019 | *Diploneis interrupta* | Unallocated |
| 149404 | *Diploneis littoralis* | Unallocated |
| 164085 | *Diploneis stroemii* | Unallocated |
| 109515 | *Diplopsalis* | AFBI; CEFAS; EA; PML; SEPA; SMHI |
| 110001 | *Diplopsalis lenticula* | AFBI; SMHI |
| 155560 | *Diplopsalopsis bomba* | AFBI; SMHI |
| 146221 | *Discomitochondria* | AFBI |
| 465546 | *Discostella pseudostelligera* | Unallocated |
| 109569 | *Dissodinium* | AFBI; EA; SEPA |
| 110325 | *Dissodinium pseudolunula* | AFBI; SMHI |
| 110326 | *Dissodium asymmetricum* | AFBI; EA |
| 128805 | *Ditrichocorycaeus anglicus* | PML; SEPA |
| 149023 | *Ditylum brightwellii* | AFBI; CEFAS; EA; MBA; MSS; PML; SEPA; SMHI |
| 134530 | *Dolichomastix* | Unallocated |
| 625374 | *Dolichospermum lemmermannii* | SMHI |
| 137212 | *Doliolida* | PML; SEPA |
| 137215 | *Doliolidae* | MBA; MSS; SMHI |
| 149510 | *Donkinia* | Unallocated |
| 178589 | *Dunaliella* | Unallocated |
| 106889 | *Ebalia* | SEPA |
| 118051 | *Ebria tripartita* | CEFAS; SEPA; SMHI |
| 1806 | *Echinodermata* | MBA; MSS; PML; SEPA; SMHI |
| 123082 | *Echinoidea* | SEPA |
| 558776 | *Echinospira larvae* | MBA |
| 117512 | *Eirene viridula* | SEPA |
| 115104 | *Emiliania huxleyi* | AFBI; PML; SMHI |
| 1820 | *Enteropneusta* | SEPA |
| 156598 | *Entomoneis* | AFBI; SMHI |
| 341375 | *Entomoneis paludosa var. hyperborea* | Unallocated |
| 1271 | *Entoprocta* | SEPA |
| 291403 | *Ephemera* | AFBI |
| 341542 | *Ephemera planamembranacea* | MBA; PML |
| 341668 | *Epiplocylis undella* | Unallocated |
| 149027 | *Epithemia turgida* | Unallocated |
| 231793 | *Erythropsidinium* | PML |
| 104085 | *Eucalanidae* | SEPA |
| 104171 | *Eucalanus* | SEPA |
| 149718 | *Eucalanus elongatus* | Unallocated |
| 847452 | *Eucalanus elongatus elongatus* | MSS |
| 345781 | *Eucalanus hyalinus* | MBA |
| 149130 | *Eucampia* | AFBI; EA; MBA; MSS |
| 248058 | *Eucampia cornuta* | AFBI; CEFAS; EA; MSS; SEPA |
| 157430 | *Eucampia groenlandica* | AFBI; CEFAS; EA; MBA; MSS; SEPA; SMHI |
| 149131 | *Eucampia zodiacus* | AFBI; CEFAS; EA; MBA; PML; SEPA; SMHI |
| 616809 | *Eucampia zoodiacus* | MSS |
| 104550 | *Euchaeta acuta* | MBA |
| 104552 | *Euchaeta marina* | MBA |
| 104553 | *Euchaeta media* | MBA |
| 104554 | *Euchaeta pubera* | MBA |



| AphiaID | Taxon | Institutes |
|---|---|---|
| 104555 | *Euchaeta spinosa* | MBA |
| 104174 | *Euchaeta* spp. | SEPA |
| 104086 | *Euchaetidae* | MBA; MSS; SEPA; SMHI |
| 104296 | *Euchirella amoena* | MBA |
| 104299 | *Euchirella curticauda* | MBA; SEPA |
| 104300 | *Euchirella maxima* | MBA |
| 104301 | *Euchirella messinensis* | MBA |
| 104302 | *Euchirella pulchra* | MBA |
| 104303 | *Euchirella rostrata* | MBA; SEPA |
| 104120 | *Euchirella* spp. | MBA; SEPA |
| 390664 | *Eudorina* | EA |
| 8012 | *Euglena* | EA; MSS; SMHI |
| 163466 | *Euglena proxima* | Unallocated |
| 21000 | *Euglenales* | SMHI |
| 582177 | *Euglenoidea* | AFBI; CEFAS |
| 19539 | *Euglenophyceae* | EA; SMHI |
| 582161 | *Euglenozoa* | SEPA |
| 105416 | *Eukrohnia hamata* | SEPA |
| 1128 | *Euphausiacea Total* | MBA; SMHI |
| 110671 | *Euphausiid* | MSS; PML; SEPA |
| 117095 | *Euphysa* spp. | MSS; SEPA |
| 416226 | *Euryarchaeota* | SMHI |
| 104240 | *Eurytemora* | SMHI |
| 104872 | *Eurytemora affinis* | SEPA |
| 157670 | *Eurytemora herdmani* | SEPA |
| 115348 | *Euterpina* | SMHI |
| 116162 | *Euterpina acutifrons* | PML; SEPA; SMHI |
| 117515 | *Eutima gracilis* | MSS; SEPA |
| 183543 | *Eutintinnus* | SMHI |
| 183557 | *Eutintinnus elongatus* | SMHI |
| 178582 | *Eutreptia* | SMHI |
| 17657 | *Eutreptiella* | AFBI; CEFAS; MSS; PML; SMHI |
| 248121 | *Eutreptiella braarudii* | SMHI |
| 573868 | *Eutreptiella cornubiense* | Unallocated |
| 573871 | *Eutreptiella eupharyngea* | Unallocated |
| 110652 | *Eutreptiella gymnastica* | SMHI |
| 172264 | *Eutreptiella hirudoidea* | Unallocated |
| 160556 | *Eutreptiella marina* | Unallocated |
| 106273 | *Evadne nordmanni* | MSS; SEPA; SMHI |
| 106267 | *Evadne* spp. | MBA; PML; SEPA; SMHI |
| 172431 | *Favella* | SMHI |
| 235761 | *Favella ehrenbergii* | SMHI |
| 292923 | *Favella helgolandica* | Unallocated |
| 233761 | *Fibrocapsa japonica* | AFBI |
| 11676 | *Fish larvae* | MSS; PML; SEPA; SMHI |
| 146222 | *Flagellates* | CEFAS; EA; PML; SEPA; SMHI |
| 1410 | *Foraminifera* | SMHI |
| 149028 | *Fragilaria* | AFBI; CEFAS; EA; MBA; MSS; PML; SEPA |
| 157458 | *Fragilaria striatula* | EA; SMHI |
| 148952 | *Fragilariaceae* | AFBI |
| 149313 | *Fragilariopsis* | AFBI; CEFAS; EA; MBA; MSS; PML; SEPA |
| 232601 | *Fragilidinium* | AFBI; MSS |
| 109468 | *Fragilidium* | AFBI; SMHI |
| 109705 | *Fragilidium subglobosum* | EA; SMHI |
| 103358 | *Fritillaria* | SMHI |



| AphiaID | Taxon | Institutes |
|---|---|---|
| 103375 | *Fritillaria borealis* | SEPA |
| 156946 | *Frustulia* | Unallocated |
| 178847 | *Fusopsis incertae sedis* | MBA |
| 10313 | *Gadiformes larva* | MSS; SEPA |
| 104312 | *Gaetanus minor* | MBA |
| 104121 | *Gaetanus* spp. | MBA |
| 237965 | *Gaetanus tenuispinus* | MBA |
| 231798 | *Gambierdiscus* spp | AFBI; MSS |
| 101383 | *Gammaridae* | SMHI |
| 1207 | *Gammaridea* | MBA |
| 236816 | *Gammariida* | PML |
| 101 | *Gastropoda larva* | MSS; PML; SEPA; SMHI |
| 235823 | *Gephyrocapsa* | PML |
| 115070 | *Gephyrocapsaceae* | AFBI; EA |
| 128775 | *Giardella callianassae* | MBA |
| 128776 | *Giardella thompsoni* | MBA |
| 109538 | *Glenodinium* | AFBI; MBA; SMHI |
| 233386 | *Goniodoma polyedricum* | AFBI; MBA |
| 346509 | *Goniopsyllus clausi* | PML |
| 577670 | *Gonium* | EA |
| 109519 | *Gonyaulax* | AFBI; CEFAS; EA; MBA; MSS; PML; SEPA; SMHI |
| 109519 | *Gonyaulax* | AFBI; CEFAS; EA; MBA; MSS; PML; SEPA; SMHI |
| 109519 | *Gonyaulax* | AFBI; CEFAS; EA; MBA; MSS; PML; SEPA; SMHI |
| 110009 | *Gonyaulax alaskenses* | AFBI; MSS |
| 110015 | *Gonyaulax digitale* | AFBI; CEFAS; EA; MSS; PML; SMHI |
| 110023 | *Gonyaulax grindleyi* | MSS; PML; SEPA |
| 110035 | *Gonyaulax polygramma* | AFBI; MSS; SMHI |
| 110039 | *Gonyaulax scrippsae* | AFBI; MSS; SMHI |
| 110041 | *Gonyaulax spinifera* | AFBI; CEFAS; EA; MSS; PML; SEPA |
| 110043 | *Gonyaulax turbynei* | AFBI |
| 110045 | *Gonyaulax verior* | AFBI; CEFAS; EA; MSS; PML; SEPA; SMHI |
| 149335 | *Grammatophora* | AFBI; EA; MBA; MSS; PML |
| 149338 | *Grammatophora marina* | AFBI; CEFAS; EA; SEPA; SMHI |
| 149339 | *Grammatophora oceanica* | Unallocated |
| 149340 | *Grammatophora serpentina* | Unallocated |
| 149111 | *Guinardia* | AFBI; CEFAS; EA; MSS |
| 163241 | *Guinardia cylindrus* | AFBI; CEFAS; EA; MBA; MSS; SEPA |
| 149112 | *Guinardia delicatula* | AFBI; CEFAS; EA; MBA; MSS; PML; SEPA; SMHI |
| 149132 | *Guinardia flaccida* | AFBI; CEFAS; EA; MBA; MSS; PML; SEPA; SMHI |
| 149113 | *Guinardia striata* | AFBI; CEFAS; EA; MBA; PML; SEPA; SMHI |
| 109392 | *Gymnodiniales* | AFBI; CEFAS; EA; MSS; SMHI |
| 109475 | *Gymnodinium* | AFBI; CEFAS; EA; MBA; MSS; PML; SEPA; SMHI |
| 109475 | *Gymnodinium* | AFBI; CEFAS; EA; MBA; MSS; PML; SEPA; SMHI |
| 109475 | *Gymnodinium* | AFBI; CEFAS; EA; MBA; MSS; PML; SEPA; SMHI |
| 232716 | *Gymnodinium aureolum* | EA; SMHI |
| 109784 | *Gymnodinium catenatum* | AFBI |
| 109785 | *Gymnodinium chlorophorum* | SMHI |
| 109791 | *Gymnodinium elongatum* | SMHI |
| 232765 | *Gymnodinium galeatum* | MSS; SMHI |
| 109802 | *Gymnodinium halophilum* | SMHI |
| 232778 | *Gymnodinium heterostriatum* | SMHI |
| 109819 | *Gymnodinium ostenfeldii* | SMHI |
| 109825 | *Gymnodinium pygmaeum* | PML |
| 109831 | *Gymnodinium simplex* | AFBI; CEFAS; SMHI |
| 232880 | *Gymnodinium verruculosum* | SMHI |



| AphiaID | Taxon | Institutes |
|---|---|---|
| 109837 | *Gymnodinium vestificii* | AFBI; CEFAS; SMHI |
| 164 | *Gymnosomata* | MSS; SEPA |
| 109476 | *Gyrodinium* | AFBI; CEFAS; EA; MBA; MSS; PML; SEPA; SMHI |
| 109844 | *Gyrodinium calyptoglyphe* | AFBI |
| 109851 | *Gyrodinium dominans* | Unallocated |
| 109852 | *Gyrodinium estuariale* | AFBI; SMHI |
| 109854 | *Gyrodinium flagellare* | SMHI |
| 109856 | *Gyrodinium fusiforme* | AFBI; CEFAS; MSS; SMHI |
| 232943 | *Gyrodinium fusus* | Unallocated |
| 109859 | *Gyrodinium lachryma* | AFBI; MSS |
| 109871 | *Gyrodinium pepo* | MSS |
| 109876 | *Gyrodinium spirale* | AFBI; CEFAS; EA; MSS; PML; SMHI |
| 109878 | *Gyrodinium undulans* | MSS |
| 149033 | *Gyrosigma* | AFBI; CEFAS; EA; MBA; SEPA; SMHI |
| 149035 | *Gyrosigma acuminatum* | Unallocated |
| 149034 | *Gyrosigma arcticum* | Unallocated |
| 661267 | *Gyrosigma attenuatum* | AFBI; CEFAS |
| 149494 | *Gyrosigma fasciola* | AFBI; CEFAS; SMHI |
| 1484 | *Halacaridae* | SEPA |
| 127482 | *Halocyprididae* | MSS; SEPA |
| 104422 | *Haloptilus acutifrons* | MBA |
| 104431 | *Haloptilus longicornis* | MBA; SEPA |
| 104437 | *Haloptilus spiniceps* | MBA |
| 134528 | *Halosphaera* | SMHI |
| 134546 | *Halosphaera viridis* | SMHI |
| 100145 | *Halosphaeria* | PML |
| 699623 | *Haptolina hirta* | Unallocated |
| 163057 | *Haptorida* | SEPA |
| 1102 | *Harpacticoida* | MBA; MSS; PML; SEPA; SMHI |
| 248063 | *Haslea wawrikae* | AFBI; PML |
| 172434 | *Helicostomella* | SMHI |
| 417184 | *Helicostomella fusiformis* | SMHI |
| 240437 | *Helicostomella subulata* | SMHI |
| 157438 | *Helicotheca* | AFBI; EA |
| 157440 | *Helicotheca tamesis* | AFBI; CEFAS; EA; MBA; PML; SEPA |
| 163248 | *Hemiaulus* | AFBI; CEFAS; MBA; SEPA |
| 1818 | *Hemichordata larva* | MSS; PML |
| 413311 | *Hemicyclops aberdonensis* | MBA |
| 180367 | *Hemidiscus cuneiformis* | AFBI; MBA |
| 106287 | *Hemiselmis* | CEFAS; SMHI |
| 106310 | *Hemiselmis virescens* | CEFAS; SMHI |
| 109540 | *Heterocapsa* | AFBI; CEFAS; EA; MSS; PML; SEPA; SMHI |
| 233619 | *Heterocapsa minima* | AFBI; CEFAS; EA; MSS; SEPA; SMHI |
| 233620 | *Heterocapsa niei* | AFBI; CEFAS; EA; MSS; PML |
| 233625 | *Heterocapsa pygmaea* | Unallocated |
| 110152 | *Heterocapsa rotundata* | AFBI; CEFAS; EA; MSS; SMHI |
| 110153 | *Heterocapsa triquetra* | AFBI; CEFAS; EA; MSS; PML; SEPA; SMHI |
| 260629 | *Heterophryxus appendiculatus* | MBA |
| 104576 | *Heterorhabdus abyssalis* | MBA |
| 104577 | *Heterorhabdus clausi* | MBA |
| 104579 | *Heterorhabdus norvegicus* | MBA; SEPA |
| 104580 | *Heterorhabdus papilliger* | MBA; SEPA |
| 104582 | *Heterorhabdus spinifer* | MBA |
| 104178 | *Heterorhabdus* spp. | MBA; SEPA |
| 160584 | *Heterosigma* | CEFAS; MSS; SMHI |





| AphiaID | Taxon | Institutes |
|---|---|---|
| 160585 | *Heterosigma akashiwo* | AFBI; SEPA; SMHI |
| 172815 | *Hexasterias problematica* | MBA |
| 447739 | *Hippdonta capitata var. capitata* | SMHI |
| 109463 | *Histioneis* | AFBI; MBA |
| 248178 | *Holococcolithophorid* | PML |
| 123083 | *Holothuroidea* | SEPA |
| 106903 | *Hyas* | SEPA |
| 117988 | *Hybocodon prolifer* | MSS; SEPA |
| 117117 | *Hydractinia* spp. | MSS; SEPA |
| 101796 | *Hyperia* spp. | MSS; SEPA |
| 101417 | *Hyperiidae* | SEPA; SMHI |
| 1205 | *Hyperiidea* | MBA; PML |
| 163096 | *Imantonia rotunda* | Unallocated |
| 104501 | *Isias clavipes* | MBA; MSS; PML; SEPA |
| 573884 | *Isochrysis galbana* | Unallocated |
| 1131 | *Isopoda* | MBA; MSS; PML; SEPA; SMHI |
| 163257 | *Isthmia* | AFBI |
| 107737 | *Jaxea nocturna* | SEPA |
| 707679 | *Karenia aureola* | Unallocated |
| 233015 | *Karenia brevis* | AFBI |
| 246593 | *Karenia cf. papilionacea* | MSS |
| 233024 | *Karenia mikimotoi* | AFBI; CEFAS; EA; MSS; PML; SEPA; SMHI |
| 231788 | *Karenia* spp. | AFBI; CEFAS; EA; MSS; SEPA |
| 233037 | *Karlodinium veneficum* | AFBI; CEFAS; PML; SMHI |
| 601744 | *Katablepharis* | SMHI |
| 119081 | *Kathablepharis remigera* | SMHI |
| 109477 | *Katodinium* | AFBI; CEFAS; EA; MSS; PML; SEPA; SMHI |
| 109882 | *Katodinium asymmetricum* | Unallocated |
| 109885 | *Katodinium glaucum* | AFBI; CEFAS; MSS; PML; SMHI |
| 178604 | *Kephyrion spirale* | Unallocated |
| 134992 | *Keratella quadrata* | SMHI |
| 163108 | *Kirchneriella* | Unallocated |
| 233165 | *Kofoidinium lebourae* | AFBI; PML |
| 109499 | *Kofoidinium* spp. | AFBI; CEFAS; MSS; SEPA |
| 495390 | *Kofoidinium velelloides* | SMHI |
| 109920 | *Kofoidinium velleloides* | AFBI |
| 451665 | *Komma caudata* | CEFAS |
| 110154 | *Kryptoperidinium foliaceum* | AFBI; CEFAS; EA; MSS; SEPA |
| 104727 | *Labidocera acutifrons* | MBA |
| 104728 | *Labidocera aestiva* | MBA |
| 104208 | *Labidocera* spp. | MBA; SEPA |
| 104736 | *Labidocera wollastoni* | MBA; PML; SEPA |
| 101190 | *Laboea* | Unallocated |
| 101264 | *Laboea strobila* | CEFAS; SEPA; SMHI |
| 178610 | *Lagerheimia genevensis* | Unallocated |
| 138101 | *Lamellaria* spp. *larva* | MSS; SEPA |
| 117725 | *Laodicea undulata* | MSS; SEPA |
| 149134 | *Lauderia* | AFBI; CEFAS; EA; MSS; SMHI |
| 149135 | *Lauderia annulata* | AFBI; CEFAS; EA; MBA; MSS; PML; SEPA; SMHI |
| 101179 | *Leegaardiella* | SMHI |
| 101206 | *Leegaardiella ovalis* | SMHI |
| 101207 | *Leegaardiella sol* | SMHI |
| 177138 | *Lemmermanniella* | SMHI |
| 549205 | *Lennoxia faveolata* | SMHI |
| 450723 | *Lennoxia* spp. | AFBI; CEFAS; EA; MSS; SEPA |





| AphiaID | Taxon | Institutes |
|---|---|---|
| 106087 | *Lepas nauplii* | MBA |
| 345481 | *Lepidodinium chlorophorum* | AFBI |
| 163401 | *Lepocinclis* | Unallocated |
| 624247 | *Lepocinclis acus* | Unallocated |
| 625430 | *Leprotintinnus pellucidus* | SMHI |
| 149038 | *Leptocylindrus* | AFBI; CEFAS; EA; SMHI |
| 573481 | *Leptocylindrus curvatus* | Unallocated |
| 149106 | *Leptocylindrus danicus* | AFBI; CEFAS; EA; MBA; MSS; PML; SEPA; SMHI |
| 149230 | *Leptocylindrus mediterraneus* | AFBI; CEFAS; EA; MBA; MSS; PML; SEPA; SMHI |
| 149039 | *Leptocylindrus minimus* | AFBI; CEFAS; EA; MSS; PML; SEPA; SMHI |
| 578704 | *Leptohalysis scotti* | Unallocated |
| 13552 | *Leptothecata* | MSS; SEPA |
| 232703 | *Lessardia elongata* | CEFAS; PML; SMHI |
| 117791 | *Leuckartiara octona* | MSS; SEPA |
| 119077 | *Leucocryptos marina* | CEFAS; SMHI |
| 149342 | *Licmophora* | AFBI; CEFAS; EA; MSS; PML; SEPA; SMHI |
| 157062 | *Licmophora abbreviata* | AFBI; CEFAS; SMHI |
| 138122 | *Limacina* | SEPA |
| 140227 | *Limacina retroversa* | MSS; PML; SEPA |
| 177508 | *Limnothrix redekei* | Unallocated |
| 233592 | *Lingulodinium polyedrum* | AFBI; CEFAS; EA; MSS; SEPA; SMHI |
| 292726 | *Lioloma* | AFBI |
| 292728 | *Lioloma delicatulum* | PML |
| 573482 | *Lioloma elongatum* | SMHI |
| 117568 | *Liriope tetraphylla* | SEPA |
| 149321 | *Lithodesmium* spp. | AFBI; EA; MSS |
| 149322 | *Lithodesmium undulatum* | AFBI; CEFAS; EA; MBA; PML; SEPA; SMHI |
| 117345 | *Lizzia blondina* | MSS; SEPA |
| 101180 | *Lohmanniella* | SMHI |
| 101209 | *Lohmanniella oviformis* | SMHI |
| 115403 | *Longipedia* spp. | SEPA; SMHI |
| 104225 | *Lophothrix* spp. | MBA |
| 117736 | *Lovenella clausa* | SEPA |
| 128672 | *Lubbockia* spp. | MBA |
| 104183 | *Lucicutia* spp. | MBA |
| 106827 | *Lucifer* spp. | MBA |
| 146538 | *Lyngbya* | Unallocated |
| 418285 | *Lyrella hennedyi* | AFBI |
| 1071 | *Malacostraca* | MSS |
| 249721 | *Mallomonas* | Unallocated |
| 249722 | *Mallomonas akrokomos* | Unallocated |
| 134562 | *Mamiella gilva* | Unallocated |
| 134563 | *Mantoniella squamata* | SMHI |
| 619174 | *Marvania coccoides* | Unallocated |
| 157052 | *Mastogloia* | AFBI |
| 104616 | *Mecynocera clausi* | MBA; SEPA |
| 345485 | *Mediopyxis helysia* | AFBI; CEFAS; EA; MBA; MSS; SEPA |
| 1337 | *Medusae* | PML; SEPA; SMHI |
| 110690 | *Meganyctiphanes norvegica* | SEPA |
| 117743 | *Melicertum octocostatum* | SEPA |
| 149042 | *Melosira* | AFBI; CEFAS; EA; MSS; PML; SEPA; SMHI |
| 160537 | *Melosira arctica* | SMHI |
| 156636 | *Melosira lineata* | MBA |
| 418547 | *Melosira moniliformis* | AFBI; CEFAS; EA; SMHI |
| 149044 | *Melosira nummuloides* | AFBI; SMHI |





| AphiaID | Taxon | Institutes |
|---|---|---|
| 149043 | *Melosira varians* | Unallocated |
| 341546 | *Membraneis challengeri* | CEFAS |
| 149345 | *Meridion* | AFBI |
| 149346 | *Meridion circulare* | EA |
| 115075 | *Meringosphaera* | EA; PML; SMHI |
| 248129 | *Meringosphaera mediterranea* | CEFAS; SMHI |
| 146545 | *Merismopedia* | AFBI; EA; SMHI |
| 177158 | *Merismopedia elegans* | Unallocated |
| 146578 | *Merismopedia glauca* | Unallocated |
| 146577 | *Merismopedia punctata* | Unallocated |
| 146576 | *Merismopedia tenuissima* | Unallocated |
| 146546 | *Merismopedia warmingiana* | Unallocated |
| 104468 | *Mesocalanus tenuicornis* | MBA; SEPA |
| 179320 | *Mesodinium* | Unallocated |
| 232069 | *Mesodinium rubrum* | AFBI; CEFAS; EA; MSS; SMHI |
| 109564 | *Mesoporos* | AFBI; CEFAS; MSS |
| 232516 | *Mesoporos perforatus* | AFBI; EA; PML |
| 104632 | *Metridia longa* | MBA; SEPA |
| 104633 | *Metridia lucens* | MBA; PML; SEPA; SMHI |
| 850801 | *Metridia lucens lucens* | MSS |
| 104190 | *Metridia* spp. (V-VI) | MBA; SEPA; SMHI |
| 104092 | *Metridinidae* | MSS |
| 149144 | *Meuniera* | AFBI; CEFAS; EA |
| 149145 | *Meuniera membranacea* | AFBI; CEFAS; EA; MBA; MSS; PML; SEPA; SMHI |
| 109510 | *Micracanthodinium* | AFBI; EA; SMHI |
| 109992 | *Micracanthodinium claytonii* | AFBI |
| 163479 | *Micractinium pusillum* | Unallocated |
| 109511 | *Micranthodinium* | AFBI; PML |
| 157675 | *Microcalanus pusillus* | MSS; SEPA |
| 104164 | *Microcalanus* spp. | MBA; PML; SEPA; SMHI |
| 146557 | *Microcystis* | AFBI; EA |
| 146558 | *Microcystis aeruginosa* | Unallocated |
| 177399 | *Microcystis reinboldii* | Unallocated |
| 134533 | *Micromonas* | SMHI |
| 134564 | *Micromonas pusilla* | SMHI |
| 116115 | *Microsetella norvegica* C1-6 | MSS; SEPA |
| 116116 | *Microsetella rosea* C1-6 | MSS; SEPA |
| 115341 | *Microsetella* spp. | PML; SEPA; SMHI |
| 148981 | *Minidiscus* | AFBI |
| 663628 | *Miniscula bipes* | EA; MSS |
| 109585 | *Minuscula bipes* | AFBI; EA |
| 116383 | *Miracia efferata* | MBA |
| 117754 | *Mitrocomella brownei* | SEPA |
| 117755 | *Mitrocomella polydiademata* | MSS; SEPA |
| 117970 | *Modeeria rotunda* | SEPA |
| 51 | *Mollusca* | MBA; MSS; SEPA |
| 621192 | *Monactinus simplex* | Unallocated |
| 447754 | *Monomorphina pyrum* | Unallocated |
| 160590 | *Monoraphidium* | EA |
| 160591 | *Monoraphidium contortum* | Unallocated |
| 160592 | *Monoraphidium convolutum* | SMHI |
| 163100 | *Monoraphidium komarkovae* | Unallocated |
| 163101 | *Monoraphidium minutum* | Unallocated |
| 105535 | *Monosiga* | SMHI |
| 119805 | *Monstrilla longiremis* | MBA; SEPA |





| AphiaID | Taxon | Institutes |
|---|---|---|
| 119777 | *Monstrillidae* C1-6 | MSS; SEPA |
| 119816 | *Mormonilla* | SEPA |
| 135366 | *Muggiaea* | SEPA |
| 135441 | *Muggiaea atlantica* | MSS; SEPA |
| 135444 | *Muggiaea kochii* | SEPA |
| 292896 | *Myrionecta rubra* | SEPA |
| 149668 | *Mysidacea* | MBA; MSS; PML; SEPA |
| 104469 | *Nannocalanus minor* | MBA; SEPA |
| 135496 | *Nanomia cara* | MSS; SEPA |
| 248180 | *Nanoneis hasleae* | PML |
| 149142 | *Navicula* | AFBI; CEFAS; EA; MBA; PML; SEPA; SMHI |
| 149467 | *Navicula directa* | Unallocated |
| 149143 | *Navicula distans* | PML |
| 172797 | *Navicula granii* | EA |
| 172799 | *Navicula gregaria* | SMHI |
| 149320 | *Navicula transitans* | AFBI; CEFAS; SMHI |
| 175320 | *Navicula transitans var. derasa* | CEFAS; SMHI |
| 175321 | *Navicula transitans var. derasa f. delicatula* | CEFAS; SMHI |
| 175319 | *Navicula transitans var. transitans* | SMHI |
| 175335 | *Navicula viridula var. rostellata* | Unallocated |
| 149031 | *Naviculaceae* | AFBI; EA |
| 149015 | *Naviculales* | AFBI; EA; MSS |
| 106927 | *Necora* | SEPA |
| 604302 | *Nematodinium* | AFBI; PML |
| 109907 | *Nematodinium armatum* | SMHI |
| 547527 | *Nematopsides vigilans* | SMHI |
| 152391 | *Nemertea* | SMHI |
| 104471 | *Neocalanus gracilis* | MBA; SEPA |
| 104472 | *Neocalanus robustior* | MBA |
| 104155 | *Neocalanus* spp. | MBA; SEPA |
| 345491 | *Neocalyptrella robusta* | AFBI; CEFAS; EA; MBA; MSS; PML; SEPA; SMHI |
| 495629 | *Neoceratium arcticum* | Unallocated |
| 495630 | *Neoceratium arietinum* | Unallocated |
| 495633 | *Neoceratium azoricum* | Unallocated |
| 495635 | *Neoceratium belone* | Unallocated |
| 495637 | *Neoceratium bigelowii* | Unallocated |
| 495638 | *Neoceratium breve* | Unallocated |
| 495640 | *Neoceratium candelabrum* | Unallocated |
| 495641 | *Neoceratium carnegiei* | Unallocated |
| 495644 | *Neoceratium compressum* | Unallocated |
| 495646 | *Neoceratium contortum* | Unallocated |
| 495648 | *Neoceratium declinatum* | Unallocated |
| 495655 | *Neoceratium extensum* | Unallocated |
| 495656 | *Neoceratium falcatiforme* | Unallocated |
| 495659 | *Neoceratium furca* | PML |
| 495660 | *Neoceratium fusus* | PML |
| 495662 | *Neoceratium gibberum* | Unallocated |
| 495664 | *Neoceratium hexacanthum* | Unallocated |
| 495666 | *Neoceratium horridum* | PML |
| 495669 | *Neoceratium inflatum* | Unallocated |
| 495671 | *Neoceratium kofoidii* | Unallocated |
| 495674 | *Neoceratium lineatum* | PML |
| 495675 | *Neoceratium longipes* | Unallocated |
| 495676 | *Neoceratium longirostrum* | Unallocated |
| 495678 | *Neoceratium macroceros* | PML |



| AphiaID | Taxon | Institutes |
|---|---|---|
| 495679 | *Neoceratium massiliense* | PML |
| 495680 | *Neoceratium minutum* | Unallocated |
| 495685 | *Neoceratium pentagonum* | Unallocated |
| 495687 | *Neoceratium platycorne* | Unallocated |
| 495690 | *Neoceratium pulchellum* | Unallocated |
| 495696 | *Neoceratium setaceum* | Unallocated |
| 495697 | *Neoceratium symmetricum* | Unallocated |
| 495700 | *Neoceratium teres* | Unallocated |
| 495701 | *Neoceratium trichoceros* | Unallocated |
| 495702 | *Neoceratium tripos* | PML |
| 495703 | *Neoceratium uncinus* | Unallocated |
| 196813 | *Neodenticula seminae* | AFBI; MBA |
| 450616 | *Neostreptotheca* | Unallocated |
| 107254 | *Nephrops norvegicus larva* | MSS; SEPA |
| 134524 | *Nephroselmis* | SMHI |
| 134541 | *Nephroselmis pyriformis* | SMHI |
| 149046 | *Nitzschia acicularis* | Unallocated |
| 341566 | *Nitzschia bicapitata* | AFBI; MBA |
| 149150 | *Nitzschia longissima* | AFBI; CEFAS; EA; MBA; MSS; SMHI |
| 609727 | *Nitzschia paleacea* | Unallocated |
| 149213 | *Nitzschia sigma* | AFBI |
| 196817 | *Nitzschia sigma rigida* | MBA |
| 149604 | *Nitzschia sigmoidea* | AFBI; CEFAS; PML; SMHI |
| 149045 | *Nitzschia* spp. | AFBI; CEFAS; EA; MBA; MSS; SMHI |
| 109500 | *Noctiluca* | AFBI; EA; MSS; SEPA; SMHI |
| 109921 | *Noctiluca scintillans* | AFBI; CEFAS; EA; MBA; PML; SEPA; SMHI |
| 109393 | *Noctilucales* | AFBI; EA; SEPA |
| 160566 | *Nodularia spumigena* | SMHI |
| 254316 | *Nyctiphanes couchii* | MSS; SEPA |
| 117034 | *Obelia* spp. | MSS; SEPA; SMHI |
| 109542 | *Oblea* | AFBI; CEFAS; MSS |
| 110155 | *Oblea rotunda* | AFBI; CEFAS; EA; MSS; SMHI |
| 249725 | *Ochromonas* | Unallocated |
| 375970 | *Octactis octonaria* | AFBI; CEFAS; EA; MSS; SEPA |
| 148963 | *Odontella* | AFBI; CEFAS; EA; MSS; SEPA; SMHI |
| 149050 | *Odontella aurita* | AFBI; CEFAS; EA; MBA; MSS; SEPA; SMHI |
| 345492 | *Odontella aurita var. minima* | Unallocated |
| 702200 | *Odontella aurita var. obtusa* | Unallocated |
| 149156 | *Odontella granulata* | AFBI; MBA; SEPA |
| 345493 | *Odontella minima* | Unallocated |
| 164116 | *Odontella mobiliensis* | AFBI; EA; MBA; PML; SEPA |
| 162940 | *Odontella obtusa* | MBA |
| 149094 | *Odontella regia* | AFBI; CEFAS; EA; MBA; SEPA |
| 149157 | *Odontella rhombus* | MBA; SEPA |
| 149051 | *Odontella rostrata* | Unallocated |
| 149095 | *Odontella sinensis* | AFBI; CEFAS; EA; MBA; MSS; PML; SEPA; SMHI |
| 341487 | *Odontella weissflogii* | MBA |
| 103367 | *Oikopleura* | SEPA; SMHI |
| 106485 | *Oithona* spp. | MBA; MSS; PML; SEPA |
| 106422 | *Oithonidae* C1-6 | MSS; SEPA |
| 595083 | *Oligotrichida* | SEPA |
| 162740 | *Ollicola vangoorii* | SMHI |
| 494102 | *Oltmannsiellopsis* | SMHI |
| 526636 | *Oltmannsiellopsis viridis* | Unallocated |
| 128690 | *Oncaea* spp. | MBA; PML; SEPA; SMHI |





| AphiaID | Taxon | Institutes |
|---|---|---|
| 128586 | *Oncaeidae* C1-6 | MSS; SEPA; SMHI |
| 178611 | *Oocystis* | SMHI |
| 178613 | *Oocystis parva* | SMHI |
| 248144 | *Oocystis pelagica* | SMHI |
| 178934 | *Oocystis solitaria* | SMHI |
| 123084 | *Ophiuroidea* | SEPA; SMHI |
| 109464 | *Ornithocercus* | AFBI; MBA |
| 146549 | *Oscillatoria* | AFBI |
| 146554 | *Oscillatoria limosa* | Unallocated |
| 1078 | *Ostracoda* | MBA; SEPA; SMHI |
| 109431 | *Ostreopsidaceae* | AFBI; EA |
| 110284 | *Oxyphysis oxytoxoides* | MSS |
| 109486 | *Oxyrrhis* | AFBI; EA |
| 109902 | *Oxyrrhis marina* | AFBI; CEFAS; EA; MSS; SEPA |
| 109528 | *Oxytoxum* | AFBI; CEFAS; EA; MBA; MSS; PML; SEPA; SMHI |
| 110087 | *Oxytoxum crassum* | Unallocated |
| 233857 | *Oxytoxum criophilum* | SMHI |
| 233870 | *Oxytoxum gracile* | MSS; SMHI |
| 110115 | *Oxytoxum scolopax* | AFBI; EA |
| 160594 | *Pachysphaera* spp. | MBA; SMHI |
| 196768 | *Pacillina arctica incertae sedis* | Unallocated |
| 106738 | *Paguridae* | SEPA |
| 623653 | *Palatinus apiculatus* | Unallocated |
| 178919 | *Pandorina colony* | EA |
| 104685 | *Paracalanus parvus* | MSS; PML; SEPA; SMHI |
| 104196 | *Paracalanus* spp. | MBA; SEPA; SMHI |
| 510916 | *Paraeuchaeta acuta* | SEPA |
| 104560 | *Paraeuchaeta glacialis* | MBA |
| 104561 | *Paraeuchaeta gracilis* | MBA |
| 104563 | *Paraeuchaeta hebes* | MBA; PML; SEPA |
| 104566 | *Paraeuchaeta norvegica* | MBA; MSS; SEPA |
| 196874 | *Paraeuchaeta* spp. | MBA; SEPA |
| 359879 | *Paraeuchaeta tonsa* | MBA |
| 196836 | *Parafavella* | Unallocated |
| 196837 | *Parafavella gigantea* | Unallocated |
| 149054 | *Paralia* | SEPA |
| 149055 | *Paralia sulcata* | AFBI; CEFAS; EA; MBA; MSS; PML; SEPA; SMHI |
| 345498 | *Parapedinella* | EA |
| 104686 | *Parapontella brevicornis* | MBA; MSS; PML; SEPA |
| 105408 | *Parasagitta* | SEPA |
| 105440 | *Parasagitta elegans* | MSS; SEPA |
| 105443 | *Parasagitta setosa* | MSS; SEPA |
| 799 | *Parasitic Nematoda* | MBA; SEPA |
| 724226 | *Parathalestris croni* | Unallocated |
| 116598 | *Parathalestris cronii* | MBA |
| 147082 | *Parathemisto* | SEPA |
| 237968 | *Pareucalanus attenuatus* | SEPA |
| 104175 | *Pareuchaeta* | SMHI |
| 136903 | *Paulinella ovalis* | SMHI |
| 109529 | *Pavillardinium* | Unallocated |
| 249731 | *Pavlova* | SMHI |
| 160561 | *Pediastrum* | CEFAS; EA; MSS; SEPA |
| 164061 | *Pediastrum duplex* | AFBI |
| 248148 | *Pedinella* | Unallocated |
| 135305 | *Pelagia noctiluca* | MSS; SEPA |



| AphiaID | Taxon | Institutes |
|---|---|---|
| 106272 | *Penilia avirostris* | MBA; SMHI |
| 1304629 | *Pennales* | AFBI |
| 109504 | *Pentapharsodinium* | AFBI |
| 109925 | *Pentapharsodinium dalei* | AFBI; MSS; SMHI |
| 138327 | *Peracle* | Unallocated |
| 109394 | *Peridiniales* | AFBI; CEFAS; EA; PML; SMHI |
| 110156 | *Peridiniella catenata* | SMHI |
| 233369 | *Peridiniella danica* | EA; SMHI |
| 109549 | *Peridinium* | AFBI; EA; MSS; SEPA |
| 163858 | *Peridinium achromaticum* | Unallocated |
| 233804 | *Peridinium quinquecorne* | AFBI; EA; MSS; SEPA; SMHI |
| 172321 | *Peritromus* | Unallocated |
| 175560 | *Petrodictyon gemma* | Unallocated |
| 175573 | *Petroneis humerosa* | Unallocated |
| 246598 | *Pfiesteria piscicida* | Unallocated |
| 163375 | *Phacus longicauda* | Unallocated |
| 163354 | *Phacus pleuronectes* | Unallocated |
| 104698 | *Phaenna spinifera* | MBA; SEPA |
| 115088 | *Phaeocystis* | AFBI; CEFAS; EA; MSS; SEPA; SMHI |
| 160538 | *Phaeocystis globosa* | AFBI; CEFAS; EA; SMHI |
| 115106 | *Phaeocystis pouchetii* | AFBI; EA; PML; SMHI |
| 175584 | *Phaeodactylum tricornutum* | AFBI; MSS; SMHI |
| 109466 | *Phalacroma* | AFBI; MBA; SEPA |
| 156505 | *Phalacroma rotundatum* | AFBI; CEFAS; EA; MSS; PML; SMHI |
| 117804 | *Phialella quadrata* | MSS; SEPA |
| 1789 | *Phoronida larva* | MSS; SEPA; SMHI |
| 148378 | *Phoronidae* | SMHI |
| 128545 | *Phoronis* | SMHI |
| 149208 | *Pinnularia* | AFBI |
| 107188 | *Pisidia longicornis* | SEPA |
| 149314 | *Plagiogramma* | AFBI |
| 149056 | *Plagiogrammopsis* | AFBI; CEFAS; EA |
| 149057 | *Plagiogrammopsis vanheurckii* | EA |
| 601957 | *Plagiolemma distortum* | PML |
| 370563 | *Plagioselmis nannoplanctica* | Unallocated |
| 106303 | *Plagioselmis prolonga* | CEFAS; SMHI |
| 106283 | *Plagioselmis* sp. | EA; SMHI |
| 149516 | *Plagiotropis* | AFBI |
| 162716 | *Planctonema lauterbornii* | SMHI |
| 196815 | *Planktoniella sol* | AFBI; CEFAS; EA; MBA; MSS; PML; SEPA; SMHI |
| 146552 | *Planktothrix agardhii* | Unallocated |
| 793 | *Platyhelminthes* | SEPA; SMHI |
| 247981 | *Pleopis polyphemoides* | MSS; SEPA |
| 106346 | *Pleurobrachia* | SEPA |
| 106386 | *Pleurobrachia pileus* | MSS; SEPA |
| 235825 | *Pleurochrysis* | SMHI |
| 235969 | *Pleurochrysis carterae* | SMHI |
| 104637 | *Pleuromamma abdominalis* | MBA; SEPA |
| 104638 | *Pleuromamma borealis* | MBA; SEPA |
| 104639 | *Pleuromamma gracilis* | MBA; SEPA |
| 104640 | *Pleuromamma piseki* | MBA; SEPA |
| 104642 | *Pleuromamma robusta* | MBA; SEPA |
| 104191 | *Pleuromamma* spp. | MBA; SEPA |
| 104643 | *Pleuromamma xiphias* | MBA; SEPA |
| 149181 | *Pleurosigma* | AFBI; CEFAS; EA; PML; SEPA; SMHI |



| AphiaID | Taxon | Institutes |
|---|---|---|
| 577792 | *Pleurosigma acutum* | Unallocated |
| 149183 | *Pleurosigma angulatum* | Unallocated |
| 248088 | *Pleurosigma directum* | Unallocated |
| 149185 | *Pleurosigma naviculaceum* | Unallocated |
| 149182 | *Pleurosigma normanii* | EA |
| 231883 | *Pleurosigma planctonicum* | AFBI; PML |
| 231884 | *Pleurosigma simonsenii* | Unallocated |
| 156586 | *Pleurosigma strigosum* | Unallocated |
| 140825 | *Pneumodermopsis ciliata* | SEPA |
| 140826 | *Pneumodermopsis paucidens* | SEPA |
| 138366 | *Pneumodermopsis* spp. | MBA; SEPA |
| 117653 | *Podocoryna aereolata* | SEPA |
| 117654 | *Podocoryna borealis* | SEPA |
| 109550 | *Podolampas* | AFBI; MBA |
| 106276 | *Podon intermedius* | MSS; SEPA; SMHI |
| 106277 | *Podon leuckartii* | MSS; SEPA; SMHI |
| 106269 | *Podon* spp. | MBA; MSS; PML; SEPA; SMHI |
| 149059 | *Podosira* spp. | AFBI; MSS |
| 149060 | *Podosira stelligera* | AFBI; CEFAS; EA; MBA; PML; SEPA |
| 883 | *Polychaeta* | MBA; MSS; PML; SEPA; SMHI |
| 109485 | *Polykrikos* | AFBI; CEFAS; EA; MSS; SEPA; SMHI |
| 109899 | *Polykrikos kofoidii* | Unallocated |
| 109901 | *Polykrikos schwartzii* | AFBI; EA; MSS; PML; SMHI |
| 104097 | *Pontellidae* | MBA; MSS; SEPA; SMHI |
| 104743 | *Pontellina plumata* | MBA; SEPA |
| 107190 | *Porcellana platycheles* | SEPA |
| 106734 | *Porcellanidae* | SEPA |
| 156689 | *Porosira glacialis* | AFBI; CEFAS; MSS; SMHI |
| 17329 | *Prasinophyceae* | AFBI; SMHI |
| 109505 | *Preperidinium* | PML |
| 614618 | *Preperidinium meunieri* | MSS |
| 109927 | *Preperidinium meunierii* | AFBI |
| 149167 | *Proboscia* | AFBI; CEFAS; EA |
| 149168 | *Proboscia alata* | AFBI; CEFAS; EA; MBA; PML; SEPA; SMHI |
| 613575 | *Proboscia curvirostris* | MBA |
| 345513 | *Proboscia indica* | AFBI; EA; MBA; SMHI |
| 341501 | *Proboscia inermis* | AFBI; MBA |
| 248181 | *Proboscia truncata* | AFBI; CEFAS; PML |
| 117836 | *Proboscidactyla stellata* | MSS; SEPA |
| 109487 | *Pronoctiluca* | AFBI; CEFAS; EA; MSS; SEPA |
| 233180 | *Pronoctiluca acuta* | Unallocated |
| 109903 | *Pronoctiluca pelagica* | AFBI; CEFAS; EA; MBA; MSS; PML; SEPA; SMHI |
| 292924 | *Proplectella* | Unallocated |
| 109396 | *Prorocentrales* | AFBI; EA |
| 109566 | *Prorocentrum* | AFBI; CEFAS; EA; MBA; MSS; SEPA; SMHI |
| 110291 | *Prorocentrum aporum* | SMHI |
| 110293 | *Prorocentrum balticum* | AFBI; CEFAS; EA; PML; SEPA; SMHI |
| 110295 | *Prorocentrum compressum* | AFBI; EA; MSS |
| 232376 | *Prorocentrum cordatum* | AFBI; CEFAS; EA; PML; SEPA; SMHI |
| 110298 | *Prorocentrum dentatum* | AFBI; MSS; PML |
| 110300 | *Prorocentrum gracile* | AFBI; CEFAS; EA; MSS |
| 110301 | *Prorocentrum lima* | AFBI; CEFAS; EA; MSS; SEPA |
| 110303 | *Prorocentrum micans* | AFBI; CEFAS; EA; MSS; PML; SEPA; SMHI |
| 110304 | *Prorocentrum minimum* | EA; MSS |
| 110310 | *Prorocentrum redfieldii* | SMHI |





| AphiaID | Taxon | Institutes |
|---------|-------|------------|
| 110314 | *Prorocentrum scutellum* | AFBI; MSS |
| 110316 | *Prorocentrum triestinum* | AFBI; CEFAS; EA; MSS; PML; SEPA |
| 425488 | *Prorodontida* | Unallocated |
| 110321 | *Protoceratium reticulatum* | AFBI; CEFAS; EA; MSS; SMHI |
| 109553 | *Protoperidinium* | AFBI; CEFAS; EA; MBA; MSS; PML; SEPA; SMHI |
| 110208 | *Protoperidinium bipes* | AFBI; CEFAS; EA; MSS; PML; SEPA; SMHI |
| 110209 | *Protoperidinium breve* | AFBI; CEFAS; SMHI |
| 110210 | *Protoperidinium brevipes* | AFBI; CEFAS; EA; MSS; PML; SMHI |
| 162749 | *Protoperidinium cerasus* | AFBI; EA; MSS; SMHI |
| 163862 | *Protoperidinium claudicans* | AFBI; CEFAS; MSS; SMHI |
| 110212 | *Protoperidinium conicoides* | AFBI; CEFAS; MSS; SMHI |
| 110213 | *Protoperidinium conicum* | AFBI; CEFAS; EA; MSS; SMHI |
| 110214 | *Protoperidinium crassipes* | AFBI; CEFAS; MSS; SMHI |
| 110215 | *Protoperidinium curtipes* | AFBI; CEFAS; EA; MSS; PML; SMHI |
| 163934 | *Protoperidinium curvipes* | AFBI; EA |
| 110216 | *Protoperidinium denticulatum* | AFBI; MSS; SMHI |
| 110217 | *Protoperidinium depressum* | AFBI; CEFAS; EA; MSS; PML; SMHI |
| 172460 | *Protoperidinium diabolum* | AFBI; MSS |
| 110219 | *Protoperidinium divergens* | AFBI; CEFAS; EA; MSS; PML; SMHI |
| 110220 | *Protoperidinium excentricum* | AFBI; MSS |
| 232921 | *Protoperidinium globulus* | AFBI |
| 110223 | *Protoperidinium granii* | AFBI; CEFAS; MSS; SMHI |
| 233257 | *Protoperidinium incognitum* | Unallocated |
| 110228 | *Protoperidinium laticeps* | SMHI |
| 110229 | *Protoperidinium leonis* | AFBI; CEFAS; EA; MSS |
| 110231 | *Protoperidinium mariaelebouriae* | MSS |
| 233176 | *Protoperidinium marielebourae* | Unallocated |
| 614620 | *Protoperidinium marielebouriae* | AFBI; SMHI |
| 110233 | *Protoperidinium minutum* | AFBI; EA; MSS |
| 110234 | *Protoperidinium mite* | AFBI; CEFAS; EA; MSS |
| 110237 | *Protoperidinium nudum* | MSS |
| 110238 | *Protoperidinium oblongum* | AFBI; CEFAS; EA; MSS; SMHI |
| 110239 | *Protoperidinium obtusum* | AFBI; MSS; PML |
| 110240 | *Protoperidinium oceanicum* | AFBI; CEFAS; EA; MSS; PML |
| 110241 | *Protoperidinium ovatum* | AFBI; CEFAS; EA; MSS; PML |
| 110244 | *Protoperidinium pallidum* | AFBI; CEFAS; EA; MSS; SMHI |
| 110245 | *Protoperidinium pellucidum* | AFBI; CEFAS; EA; MSS; SMHI |
| 110247 | *Protoperidinium pentagonum* | AFBI; CEFAS; EA; MSS; SMHI |
| 110248 | *Protoperidinium punctulatum* | AFBI; CEFAS; SMHI |
| 110249 | *Protoperidinium pyriforme* | AFBI; EA; MSS; PML; SMHI |
| 110250 | *Protoperidinium quarense* | MSS |
| 110257 | *Protoperidinium steinii* | AFBI; CEFAS; EA; MSS; PML; SMHI |
| 110258 | *Protoperidinium subicurvipes* | MSS |
| 110259 | *Protoperidinium subinerme* | AFBI; CEFAS; MSS; SMHI |
| 110260 | *Protoperidinium thorianum* | AFBI; MSS; SMHI |
| 232861 | *Protoperidinium thulense* | AFBI; MSS |
| 233390 | *Protoperidnium decipiens* | MSS |
| 115061 | *Prymnesiales* | SMHI |
| 115057 | *Prymnesiophyceae* | AFBI; EA; MSS; PML; SEPA; SMHI |
| 160563 | *Prymnesium* | SMHI |
| 149217 | *Psammodictyon panduriforme* | AFBI; PML |
| 177588 | *Pseudanabaena* | SMHI |
| 577876 | *Pseudanabaena acicularis* | SMHI |
| 177590 | *Pseudanabaena limnetica* | Unallocated |
| 103990 | *Pseudobodo* | SMHI |





| AphiaID | Taxon | Institutes |
|---|---|---|
| 104515 | *Pseudocalanus elongatus* | PML; SEPA; SMHI |
| 104517 | *Pseudocalanus minutus* | SMHI |
| 149711 | *Pseudocalanus minutus elongatus* | Unallocated |
| 104165 | *Pseudocalanus* spp. | MBA; MSS; SEPA; SMHI |
| 531445 | *Pseudochattonella* | SMHI |
| 531467 | *Pseudochattonella farcimen* | SMHI |
| 531446 | *Pseudochattonella verruculosa* | SMHI |
| 104757 | *Pseudocyclopia minor* | MSS; SEPA |
| 157680 | *Pseudodiaptomus* spp. | MBA |
| 573543 | *Pseudoguinardia recta* | AFBI; CEFAS; EA |
| 478556 | *Pseudo-nitzschia americana* | CEFAS |
| 246604 | *Pseudo-nitzschia australis* | AFBI |
| 246605 | *Pseudo-nitzschia calliantha* | SMHI |
| 149153 | *Pseudo-nitzschia delicatissima* | CEFAS; EA; MBA; PML; SMHI |
| 246606 | *Pseudo-nitzschia fraudulenta* | SMHI |
| 411764 | *Pseudo-nitzschia heimii* | Unallocated |
| 175738 | *Pseudo-nitzschia multiseries* | Unallocated |
| 246608 | *Pseudo-nitzschia multistriata* | CEFAS |
| 156548 | *Pseudo-nitzschia pseudodelicatissima* | SMHI |
| 160528 | *Pseudo-nitzschia pungens* | PML; SMHI |
| 149152 | *Pseudo-nitzschia seriata* | CEFAS; EA; MBA; PML |
| 175749 | *Pseudo-nitzschia seriata f. seriata* | SMHI |
| 149151 | *Pseudo-nitzschia* spp | AFBI; CEFAS; EA; MSS; SEPA; SMHI |
| 411767 | *Pseudo-nitzschia subcurvata* | Unallocated |
| 418222 | *Pseudo-nitzschia subpacifica* | Unallocated |
| 621601 | *Pseudopediastrum boryanum* | Unallocated |
| 160599 | *Pseudopedinella* | CEFAS; SMHI |
| 248149 | *Pseudopedinella elastica* | SMHI |
| 160600 | *Pseudopedinella pyriformis* | CEFAS; SMHI |
| 160601 | *Pseudopedinella tricostata* | SMHI |
| 418160 | *Pseudopfiesteria shumwayae* | Unallocated |
| 577639 | *Pseudopodosira westii* | Unallocated |
| 105445 | *Pseudosagitta maxima* | SEPA |
| 134566 | *Pseudoscourfieldia marina* | SMHI |
| 163344 | *Pseudosolenia calcar-avis* | AFBI; CEFAS; EA; MBA; MSS; SEPA; SMHI |
| 178959 | *Pseudostaurastrum hastatum* | Unallocated |
| 160595 | *Pterosperma* | MBA; PML; SMHI |
| 345881 | *Pterosperma vanhoeffenii* | SMHI |
| 109478 | *Ptychodiscus* | AFBI; EA |
| 109888 | *Ptychodiscus noctiluca* | AFBI; MBA |
| 1302 | *Pycnogonida* | MBA |
| 134529 | *Pyramimonas* | AFBI; CEFAS; EA; PML; SEPA; SMHI |
| 134550 | *Pyramimonas (Trichocystis) grossii* | Unallocated |
| 495333 | *Pyramimonas disomata* | SMHI |
| 160513 | *Pyramimonas longicauda* | SMHI |
| 134559 | *Pyramimonas virginica* | CEFAS; SMHI |
| 109571 | *Pyrocystis* | AFBI; CEFAS; EA; MBA; MSS; SEPA |
| 164053 | *Pyrocystis lunula* | AFBI |
| 110332 | *Pyrocystis noctiluca* | Unallocated |
| 109555 | *Pyrophacus* | AFBI; CEFAS; MBA |
| 110266 | *Pyrophacus horologicum* | MSS; PML |
| 232598 | *Pyrophacus horologium* | AFBI; CEFAS; EA; SMHI |
| 623185 | *Quadricoccus ellipticus* | SMHI |
| 613622 | *Quadricoccus euryhalinicus* | SMHI |
| 582421 | *Radiozoa* | Unallocated |





| AphiaID | Taxon | Institutes |
|---------|-------|-----------|
| 160581 | *Raphidophyceae* | AFBI; PML |
| 708245 | *Raphidosphaera tenerrima* | EA |
| 149065 | *Raphoneis* spp. | AFBI; CEFAS; EA; MSS |
| 117848 | *Rathkea octopunctata* | MSS; SEPA |
| 375891 | *Resultor mikron* | Unallocated |
| 292925 | *Rhabdoaskenasia* | Unallocated |
| 626382 | *Rhabdolithes claviger* | PML |
| 157072 | *Rhabdonema* | AFBI; CEFAS; MBA; MSS |
| 149066 | *Rhaphoneis amphiceros* | AFBI; CEFAS; EA; MBA; SMHI |
| 104172 | *Rhincalanus* | SEPA |
| 104542 | *Rhincalanus cornutus* | MBA |
| 104543 | *Rhincalanus nasutus* | MBA; MSS; SEPA |
| 118071 | *Rhizomonas setigera* | SMHI |
| 149069 | *Rhizosolenia* | AFBI; CEFAS; EA; MBA; MSS; SEPA; SMHI |
| 196805 | *Rhizosolenia acuminata* | AFBI; MBA |
| 149223 | *Rhizosolenia alata* | Unallocated |
| 196808 | *Rhizosolenia alata f. curvirostris* | Unallocated |
| 196811 | *Rhizosolenia bergonii* | AFBI; MBA |
| 567120 | *Rhizosolenia borealis* | AFBI; CEFAS; EA; MSS |
| 163346 | *Rhizosolenia calcar-avis* | AFBI |
| 341502 | *Rhizosolenia chunii* | PML |
| 573572 | *Rhizosolenia fallax* | MSS |
| 149070 | *Rhizosolenia hebetata* | AFBI; CEFAS; EA; MBA; SEPA; SMHI |
| 163347 | *Rhizosolenia hebetata f. hebetata* | EA |
| 149071 | *Rhizosolenia hebetata f. semispina* | CEFAS; MSS; PML; SMHI |
| 149116 | *Rhizosolenia imbricata* | AFBI; CEFAS; EA; MBA; PML; SEPA; SMHI |
| 149117 | *Rhizosolenia pungens* | AFBI; EA; MBA; MSS; SEPA; SMHI |
| 149115 | *Rhizosolenia setigera* | AFBI; CEFAS; EA; MBA; MSS; PML; SEPA; SMHI |
| 547544 | *Rhizosolenia setigera f. pungens* | CEFAS; EA |
| 149165 | *Rhizosolenia shrubsolei* | MSS |
| 149224 | *Rhizosolenia stolterfothii* | MSS |
| 149629 | *Rhizosolenia styliformis* | AFBI; CEFAS; EA; MBA; MSS; PML; SEPA; SMHI |
| 106289 | *Rhodomonas* | EA; SMHI |
| 106313 | *Rhodomonas baltica* | SMHI |
| 106314 | *Rhodomonas marina* | SMHI |
| 106316 | *Rhodomonas salina* | SMHI |
| 175794 | *Rhoicosphenia abbreviata* | CEFAS |
| 146182 | *Rhynchobodo* | SMHI |
| 149105 | *Roperia tesselata* | AFBI; EA; PML |
| 14260 | *Rotifera* | SMHI |
| 105410 | *Sagitta* | SMHI |
| 105450 | *Sagitta elegans* | SMHI |
| 154102 | *Sagitta elegans elegans* | SMHI |
| 154107 | *Sagitta setosa* | SMHI |
| 5953 | *Sagittidae juvenile* | MSS; SEPA |
| 137272 | *Salpa fusiformis* | MSS; SEPA |
| 137217 | *Salpidae* | MBA; SEPA |
| 183566 | *Salpingella* | SMHI |
| 417228 | *Salpingella acuminata* | SMHI |
| 128722 | *Sapphirina* spp. | MBA; SEPA |
| 117070 | *Sarsia* spp. | MSS; SEPA |
| 117491 | *Sarsia tubulosa* | MSS; SEPA |
| 104793 | *Scaphocalanus echinatus* | MBA |
| 104228 | *Scaphocalanus* spp. | MBA |
| 160541 | *Scenedesmaceae* | Unallocated |





| AphiaID | Taxon | Institutes |
|---|---|---|
| 160602 | *Scenedesmus* | CEFAS; EA; SEPA; SMHI |
| 162929 | *Scenedesmus armatus* | Unallocated |
| 596169 | *Scenedesmus quadricauda* | Unallocated |
| 615516 | *Sclerodinium calyptoglyphe* | AFBI; EA |
| 104811 | *Scolecithricella minor* | MSS; SEPA; SMHI |
| 104229 | *Scolecithricella* spp. | MBA; PML; SEPA |
| 196125 | *Scolecithricidae* | SEPA; SMHI |
| 104820 | *Scolecithrix bradyi* | MBA |
| 104821 | *Scolecithrix danae* | MBA |
| 104230 | *Scolecithrix* spp. | MBA; SEPA |
| 104103 | *Scolecitrichidae* C1-5 | MSS |
| 104832 | *Scottocalanus persecans* | MBA |
| 104833 | *Scottocalanus securifrons* | MBA |
| 109545 | *Scrippsiella* | AFBI; CEFAS; EA; MBA; MSS; PML; SEPA; SMHI |
| 233093 | *Scrippsiella hangoei* | AFBI; SMHI |
| 110172 | *Scrippsiella trochoidea* | AFBI; CEFAS; EA; MSS; PML; SMHI |
| 135220 | *Scyphozoa* | MSS; SEPA; SMHI |
| 106731 | *Sergestidae* | MBA |
| 105464 | *Serratosagitta serratodentata* | MSS; SEPA |
| 573446 | *Shionodiscus oestrupii* | Unallocated |
| 610927 | *Siderocelis ornata* | Unallocated |
| 663187 | *Sinophysis ebriolus* | Unallocated |
| 109467 | *Sinophysis species* | AFBI; CEFAS; EA; MSS; SEPA |
| 1371 | *Siphonophora* | MBA; MSS; PML; SEPA; SMHI |
| 1104 | *Siphonostomatoida* | MBA; PML; SEPA |
| 410762 | *Sipunculus* | Unallocated |
| 149073 | *Skeletonema* | AFBI; CEFAS; EA; MSS; SEPA |
| 149074 | *Skeletonema costatum* | AFBI; CEFAS; EA; MBA; PML |
| 376667 | *Skeletonema marinoi* | AFBI; SMHI |
| 163390 | *Skeletonema potamos* | Unallocated |
| 149075 | *Skeletonema subsalsum* | Unallocated |
| 565148 | *Slabberia halterata* | SEPA |
| 146644 | *Snowella* | SMHI |
| 177465 | *Snowella fennica* | SMHI |
| 117947 | *Solmaris corona* | MSS; SEPA |
| 15177 | *Spadellidae* | MSS; SEPA |
| 109502 | *Spatulodinium* | AFBI |
| 109923 | *Spatulodinium pseudonoctiluca* | AFBI; CEFAS; EA; MSS; SEPA; SMHI |
| 660753 | *Spirulina subsalsa* | Unallocated |
| 162728 | *Staurastrum* | EA |
| 565161 | *Stauridiosarsia gemmifera* | SEPA |
| 248146 | *Stauridium tetras* | Unallocated |
| 149078 | *Stauroneis phoenicenteron* | Unallocated |
| 594013 | *Staurostoma mertensii* | MSS; SEPA |
| 149653 | *Stellarima stellaris* | SMHI |
| 178825 | *Stenosemella* | SMHI |
| 149080 | *Stephanodiscus hantzschii* | Unallocated |
| 149081 | *Stephanodiscus medius* | Unallocated |
| 149082 | *Stephanodiscus niagarae* | Unallocated |
| 149083 | *Stephanodiscus parvus* | Unallocated |
| 149630 | *Stephanopyxis* | AFBI; CEFAS; EA; MBA; MSS; SEPA |
| 231888 | *Stephanopyxis palmeriana* | AFBI; PML |
| 149631 | *Stephanopyxis turris* | AFBI; CEFAS; EA; MSS; SEPA; SMHI |
| 14355 | *Stomatopoda* | MBA; SMHI |
| 149177 | *Striatella unipunctata* | AFBI; CEFAS; EA; MSS; SEPA; SMHI |



| AphiaID | Taxon | Institutes |
|---|---|---|
| 101185 | *Strobilidium* | SMHI |
| 578585 | *Strobilidium sphaericum* | SMHI |
| 101198 | *Strombidinopsis* | SMHI |
| 101195 | *Strombidium* | SMHI |
| 595215 | *Strombidium chlorophilum* | SMHI |
| 732820 | *Strombidium emergens* | SMHI |
| 602678 | *Strombus* | Unallocated |
| 110678 | *Stylocheiron* | SEPA |
| 104544 | *Subeucalanus crassus* | MBA; MSS; SEPA |
| 104545 | *Subeucalanus monachus* | MBA |
| 104546 | *Subeucalanus mucronatus* | MBA |
| 104547 | *Subeucalanus pileatus* | MBA |
| 104173 | *Subeucalanus* spp. | PML |
| 345526 | *Subsilicea fragilarioides* | Unallocated |
| 345525 | *Subsiliceae* sp. | MSS |
| 149084 | *Surirella* | AFBI; CEFAS; MBA |
| 149085 | *Surirella crumena* | Unallocated |
| 149087 | *Surirella ovalis* | Unallocated |
| 134958 | *Synchaeta* | SMHI |
| 160572 | *Synechococcus* | SMHI |
| 610181 | *Synechococcus elongatus* | Unallocated |
| 177482 | *Synechocystis* | Unallocated |
| 149186 | *Synedra* | AFBI; CEFAS; EA |
| 163712 | *Synedra fulgens* | Unallocated |
| 149187 | *Synedra ulna* | Unallocated |
| 341536 | *Synedropsis hyperboreoides* | Unallocated |
| 178594 | *Synura* | EA |
| 236039 | *Syracosphaera molischii* | PML |
| 235979 | *Syracosphaera pulchra* | PML |
| 149333 | *Tabellaria* | AFBI; CEFAS; MSS |
| 106285 | *Teleaulax* | CEFAS; SMHI |
| 106305 | *Teleaulax acuta* | CEFAS; SMHI |
| 106306 | *Teleaulax amphioxeia* | CEFAS; SMHI |
| 118028 | *Telonema* | SMHI |
| 118075 | *Telonema subtile* | SMHI |
| 104241 | *Temora* | SMHI |
| 104878 | *Temora longicornis* | MBA; MSS; PML; SEPA; SMHI |
| 104879 | *Temora stylifera* | MBA; PML; SEPA |
| 104880 | *Temora turbinata* | MBA |
| 1249 | *Tentaculata* | SMHI |
| 178949 | *Tetraedron* | CEFAS; EA; MSS; SEPA |
| 178956 | *Tetraedron minimum* | Unallocated |
| 134526 | *Tetraselmis* | Unallocated |
| 162935 | *Tetrastrum staurogeniaeforme* | Unallocated |
| 149089 | *Thalassiocyclus lucens* | Unallocated |
| 555052 | *Thalassionema frauenfeldii* | AFBI; CEFAS; SMHI |
| 149093 | *Thalassionema nitzschioides* | AFBI; CEFAS; EA; MBA; PML; SEPA; SMHI |
| 704849 | *Thalassionema nitzschoides* | Unallocated |
| 149092 | *Thalassionema* spp | AFBI; EA; MSS; SEPA |
| 148912 | *Thalassiosira* | AFBI; CEFAS; EA; MBA; MSS; PML; SEPA; SMHI |
| 345529 | *Thalassiosira aestivalis* | AFBI |
| 148913 | *Thalassiosira angulata* | AFBI; CEFAS; SMHI |
| 148914 | *Thalassiosira anguste-lineata* | AFBI; CEFAS; PML; SMHI |
| 149099 | *Thalassiosira antarctica* | AFBI; SMHI |
| 156690 | *Thalassiosira baltica* | SMHI |



| AphiaID | Taxon | Institutes |
|---|---|---|
| 345531 | *Thalassiosira constricta* | AFBI; CEFAS; PML; SMHI |
| 148919 | *Thalassiosira decipiens* | AFBI; CEFAS; SMHI |
| 555264 | *Thalassiosira delicatula* | SMHI |
| 148922 | *Thalassiosira eccentrica* | AFBI; PML; SMHI |
| 149102 | *Thalassiosira gravida* | AFBI; CEFAS; PML; SMHI |
| 148923 | *Thalassiosira hendeyi* | Unallocated |
| 149100 | *Thalassiosira hyalina* | CEFAS; SMHI |
| 495332 | *Thalassiosira kushirensis* | Unallocated |
| 163494 | *Thalassiosira lacustris* | SMHI |
| 149308 | *Thalassiosira levanderi* | AFBI; SMHI |
| 345546 | *Thalassiosira lineata* | Unallocated |
| 148925 | *Thalassiosira minima* | CEFAS; SMHI |
| 555293 | *Thalassiosira minuscula* | PML |
| 149256 | *Thalassiosira nana* | Unallocated |
| 148929 | *Thalassiosira nodulolineata* | Unallocated |
| 148931 | *Thalassiosira nordenskioeldii* | AFBI; CEFAS; SMHI |
| 960571 | *Thalassiosira nordenskioldii* | MSS |
| 148932 | *Thalassiosira pacifica* | Unallocated |
| 148933 | *Thalassiosira proschkinae* | Unallocated |
| 148934 | *Thalassiosira pseudonana* | Unallocated |
| 148936 | *Thalassiosira punctigera* | AFBI; CEFAS; MSS; PML; SMHI |
| 148942 | *Thalassiosira rotula* | AFBI; CEFAS; EA; MSS; PML; SMHI |
| 149101 | *Thalassiosira subtilis* | AFBI; PML |
| 163513 | *Thalassiosira weissflogii* | CEFAS |
| 157081 | *Thalassiothrix* | AFBI; PML |
| 157083 | *Thalassiothrix longissima* | AFBI; CEFAS; MBA; SMHI |
| 22626 | *Thaliacea* | SMHI |
| 109557 | *Thecadinium* | AFBI; MSS |
| 13703 | *Thecosomata* | MBA; SEPA; SMHI |
| 101800 | *Themisto* spp. | MSS |
| 110708 | *Thysanoessa inermis* | MSS; SEPA; SMHI |
| 110709 | *Thysanoessa longicaudata adult* | MSS; SEPA |
| 110711 | *Thysanoessa raschii* | SEPA; SMHI |
| 237874 | *Thysanoessa spinifera* | SMHI |
| 110679 | *Thysanoessa* spp. *furcilia* | MSS; SEPA |
| 247913 | *Tiarina* | Unallocated |
| 247943 | *Tiarina fusus* | SEPA; SMHI |
| 117978 | *Tiaropsis multicirrata* | MSS; SEPA |
| 115351 | *Tigriopus* | SEPA |
| 117527 | *Tima bairdii* | SEPA |
| 425497 | *Tintinnida* | SEPA |
| 247915 | *Tintinnidium* | SMHI |
| 732976 | *Tintinnina* | AFBI |
| 163780 | *Tintinnopsis* | SMHI |
| 163782 | *Tintinnopsis beroidea* | SMHI |
| 334946 | *Tomopteris helgolandica* | PML; SEPA |
| 129715 | *Tomopteris* spp. | MBA; MSS; SEPA; SMHI |
| 101196 | *Tontonia* | SMHI |
| 427744 | *Tontonia ovalis* | Unallocated |
| 109479 | *Torodinium* | AFBI; CEFAS; EA; SEPA; SMHI |
| 109889 | *Torodinium robustum* | AFBI; CEFAS; EA; MSS; PML; SMHI |
| 109890 | *Torodinium teredo* | PML |
| 720592 | *Tortanus (Boreotortanus) discaudatus* | SEPA |
| 157684 | *Tortanus discaudatus* | MBA |
| 248074 | *Toxarium* | Unallocated |





| AphiaID | Taxon | Institutes |
|---|---|---|
| 163412 | *Trachelomonas hispida* | Unallocated |
| 16350 | *Trachymedusae* | SEPA |
| 149146 | *Trachyneis* | Unallocated |
| 576713 | *Treubaria* | CEFAS; EA |
| 149170 | *Triceratium favus* | AFBI; MBA |
| 149154 | *Triceratium* sp | AFBI; CEFAS; EA; MSS; SEPA |
| 177604 | *Trichodesmium* | MBA |
| 836581 | *Trieres* | AFBI; CEFAS |
| 839991 | *Trieres mobiliensis* | AFBI; CEFAS; SMHI |
| 699394 | *Trigonium alternans* | AFBI; CEFAS; EA; MSS; PML; SEPA |
| 254445 | *Trigonium* spp. | MSS |
| 494057 | *Tripos* | AFBI |
| 841182 | *Tripos arietinus* | AFBI; CEFAS |
| 837310 | *Tripos azoricus* | AFBI |
| 841188 | *Tripos bigelowii* | AFBI |
| 841199 | *Tripos compressus* | AFBI |
| 841211 | *Tripos eugrammus* | EA |
| 840627 | *Tripos furca* | AFBI; CEFAS |
| 840626 | *Tripos fusus* | CEFAS |
| 837453 | *Tripos horridus* | CEFAS |
| 837456 | *Tripos kofoidii* | AFBI |
| 837459 | *Tripos lineatus* | AFBI; CEFAS |
| 841259 | *Tripos longipes* | AFBI; CEFAS |
| 841260 | *Tripos macroceros* | AFBI; CEFAS |
| 841263 | *Tripos minutus* | AFBI |
| 495363 | *Tripos muelleri* | CEFAS |
| 841746 | *Tripos pentagonus* | AFBI |
| 841751 | *Tripos pulchellus* | AFBI |
| 837234 | *Tripos ranipes* | AFBI |
| 447746 | *Tryblionella compressa* | PML |
| 176534 | *Tryblionella navicularis* | Unallocated |
| 146420 | *Tunicata* | SMHI |
| 117056 | *Turritopsis* | SEPA |
| 447744 | *Ulnaria ulna* | SMHI |
| 235937 | *Umbellosphaera* | PML |
| 104342 | *Undeuchaeta major* | MBA; SEPA |
| 104343 | *Undeuchaeta plumosa* | MBA; SEPA |
| 104128 | *Undeuchaeta* spp. | MBA; SEPA |
| 367334 | *Undinula vulgaris* | MBA |
| 128639 | *Urocorycaeus* spp. | MBA |
| 143943 | *Uronema* | SMHI |
| 120566 | *Vannella* | SMHI |
| 163573 | *Vorticella* | Unallocated |
| 109491 | *Warnowia* | AFBI; PML |
| 104204 | *Xanthocalanus* spp. | MBA; MSS; SEPA |
| 117998 | *Zanclea costata* | SEPA |
| 196832 | *Zoothamnium pelagicum* | Unallocated |

**Table B1:** Relevant traits in lifeform traits list. The trait list reflects the lifeforms and types of plankton data in the datasets used to date in lifeform-based assessment. The list is a living document, whose status reflects ongoing efforts to refine and improve the lifeform approach as well as a necessary compromise between focusing on traits which inform the lifeforms currently used for assessments and including additional information and traits where these are known and readily available. Trait groups for the plankton type "Protozoa" have recently been added to the trait list but are not shown here as they are not finalised and not used in any of the current lifeforms. The inclusion of '*protozoa*' and '*ciliates*' (which are protozoa) designation under '*phytoplankton type*' is to ensure these key taxa are captured into lifeforms. One important example is the abundant group '*Flagellates*' which has '*Plankton Type = phytoplankton*' to allow assignment of the phytoplankton trait groups, despite including both phototrophic and heterotrophic taxa and thus *Phytoplankton Type = protozoa*'.

In the traits list: spaces between words are omitted, e.g. '*Plankton Type*' is written as '*PlanktonType*', and Zooplankton is often abbreviated as '*Zoo*'. For all trait categories, '*Y*': yes (trait applies); '*N*': no (trait does not apply); The following definitions apply to all columns (trait categories) of the list, with additional details given in the table where relevant:

1. '*Ambiguous*': taxa cannot be reliably assigned to any one category for this trait, mostly because taxa within this group can fall under more than one trait category (e.g.: taxa categories which include individuals of both 'large' and 'small' size classes);

2. '*[blank]*': trait is not use for this Plankton Type (e.g.: "PhytoHabitat" is blank for all zooplankton taxa groups);

3. '*NA*': (Not applicable) trait is used for this Plankton Type but not relevant for this taxa (e.g.: the line "fish larvae" has a "ZooHabitat" of *NA* because they are neither meroplankton nor holoplankton, the line is intended to only contribute to the "fish larvae" lifeform);

4. '*NYA*': (Not Yet Assigned) trait is used for this Plankton Type but is not yet assigned (e.g. phytoplankton which have not been assigned any of "tychopelagic", "pelagic", or "ambiguous" under "PhytoDepth".

| Trait Category | | Trait Assigned | Description/notes (see also main text, Table 2) |
|---|---|---|---|
| Plankton Type | | Phytoplankton | The 'type' defines which of the following groups of traits applies to the taxa group in question. |
| | | Protozoa | |
| | | Zooplankton | |
| *Phytoplankton Traits* | Phytoplankton Type | (various) | Types included are: Cercozoa; Charophyte; Chlorophyte; Chrysophyte; Ciliate*; Cryptophyte; Cyanobacteria; Diatom; Dictyochophyte; Dinoflagellate; Euryarchaeote; Eustigmatophyceae; Haptophyte; Protozoa*; Raphidophyte; Silicoflagellate; Xanthophyceae. *Note that while the trait groups for the plankton type "Protozoa" have recently been added to the master taxa list, "protozoa" and "ciliates" (which are protozoa) remain designated under "phytoplankton type" to ensure these key taxa are captured in the lifeforms currently used in assessments. This reflects the 'living' nature of the Master Taxa List. |
| | Plankton Size | Sm | Small (≤ 20 μm individual cell diameter) |
| | | Lg | Large (> 20 μm individual cell diameter) |
| | Size Class | 1 | Used to differentiate between taxa/groups that are of ambiguous size but have size information recorded in the raw datasets. 1 is large 'Plankton Size'; 2 is small 'Plankton Size' |
| | | 2 | |
| | | [blank] | Additional Size Class not required. |
| | Phytoplankton Depth | Pelagic | Living in the water column. |





| | | | |
|---|---|---|---|
| | | Tychopelagic | Benthic taxa, which can become mixed into the water column. |
| | Phytoplankton Feeding Mechanism ('PhytoFeedingMech') | Auto | Autotrophic: nutrition by photosynthesis. |
| | | Auto/Mixo | Auto- and mixotrophic |
| | | Hetro | Heterotrophic: non-photosynthesising. |
| | | Mixo | Mixotrophic: capable of obtaining nourishment via photo(auto)trophy and phago(hetero)trophy, as well as via osmo(hetero)trophy. |
| | Phytoplankton Habitat | Freshwater | |
| | | Marine | |
| | Potentially Toxic or Nuisance | Ambiguous | Cannot assign trait: Some taxa in the group may be toxic. Taxa in this group cannot be identified to the taxonomic level required to confirm if they are a toxin producing strain or species using routine monitoring techniques. Toxin production in the relevant species may be strain dependent requiring confirmation using molecular methods. The mechanism of harm for this species may not be confirmed (e.g.: fish mortalities caused by either anoxia or ichthyotoxins). |
| | | Nuisance | Taxa produce effects which are detrimental to aquaculture and benthos via physical harm or causing anoxia or produce water discolourations, scums or foams that can be aesthetically, socially, or economically negative |
| | | Non-Toxic | Taxa do not produce toxins which pose a risk to marine biota or human health, and do not produce nuisance effects, and do not produce nuisance effects. |
| *Zooplankton Traits* | Zooplankton Type | (various) | Types included are: Bryozoan; Cephalochordate; Cephalopod; Chaetognath; Cladoceran; Crustacean; Echinoderm; Fish; Gastropod; Gelatinous; Hemichordate; Mollusc; Nematode; Nemertea; Phoronid; Polychaete; Rotifer; Sipuncula; Tunicate. |
| | Habitat | Holoplankton | Zooplankton taxa which spend their entire lifecycle in the plankton. |
| | | Meroplankton | Taxa which spend part of their lifecycle as zooplankton. |
| | Diet | Carnivore | Taxa which prey mainly on other zooplankton. |
| | | Herbivore | Taxa which are predominately suspension or filter feeders. |
| | | Omnivore | Can use both carnivorous and herbivorous feeding. |
| | | Ambiguous | Cannot assign trait: feeding mechanism variable or none of carnivore, herbivore or omnivore. |
| | | Parasite | Feeds attached to food source either internally or externally. |
| | Crustacean | Y | Taxa of the Subphylum Crustacea |
| | | N | |
| | Copepod | Y | Taxa of the Subclass Copepoda. |
| | | N | |
| | Gelatinous | Y | Taxa of the phyla Cnidaria and Ctenophora only. |
| | | N | |
| | Zooplankton Size | Sm | < 2 mm adult total body length. |
| | | Lg | ≥ 2 mm adult total body length. |
| | | Ambiguous | Cannot assign trait: taxa in group includes those both under and over 2 mm. |



**Table C1:** Definitions of lifeforms by trait as defined in the traits list. See Table 2 for descriptions of traits used.

| Lifeform | Definition (Trait(s)) |
|---|---|
| (micro)Phytoplankton | **PlanktonType** = *Phytoplankton* |
| Large (micro)phytoplankton (>= 20 µm) | **PlanktonType** = *Phytoplankton* AND **PhytoplanktonSize** = *Lg* |
| Small (micro)phytoplankton (< 20 µm) | **PlanktonType** = *Phytoplankton* AND **PhytoplanktonSize** = *Sm* |
| Diatoms | **PhytoplanktonType** = *Diatom* |
| Pelagic diatoms | **PhytoplanktonType** = *Diatom* AND **PhytoDepth** = *Pelagic* |
| Tychopelagic diatoms | **PhytoplanktonType** = *Diatom* AND **PhytoDepth** = *Tychopelagic* |
| Potentially toxic or nuisance diatoms | **PhytoplanktonType** = *Diatom* AND **Toxic_Nuisance** = (*Toxic* OR *Nuisance*) |
| Dinoflagellates | **PhytoplanktonType** = *Dinoflagellate* |
| Autotrophic and mixotrophic dinoflagellates | **PhytoplanktonType** = *Dinoflagellate* AND **PhytoFeedingMech** = (*Auto* OR *Auto/Mixo*) |
| Potentially toxic or nuisance dinoflagellates | **PhytoplanktonType** = *Dinoflagellate* AND **Toxic_Nuisance** = (*Toxic* OR *Nuisance*) |
| Ciliates | **PhytoplanktonType** = *Ciliate* |
| Holoplankton | **ZooHabitat** = *Holoplankton* |
| Meroplankton | **ZooHabitat** = *Meroplankton* |
| Gelatinous zooplankton | **PlanktonType** = *Zooplankton* AND **Gelatinous** = *Y* |
| Carnivorous zooplankton | **PlanktonType** = *Zooplankton* AND **ZooDiet** = *Carnivore* |
| Non carnivorous zooplankton | **PlanktonType** = *Zooplankton* AND **ZooDiet** = (*Herbivore* OR *Omnivore* OR *Ambiguous*) |
| Crustaceans | **Crustacean** = *Y* |
| Large copepod species(>=2 mm) | **Copepod** = *Y* AND **ZooSize** = *Lg* |
| Small copepod species (< 2 mm) | **Copepod** = *Y* AND **ZooSize** = *Sm* |
| Fish larvae | **ZooType** = *Fish* |