# Peer review of "The Plankton Lifeform Extraction Tool: A digital tool to increase the discoverability and usability of plankton time-series data"

_Earth System Science Data, 2021_

## Author Comment (AC1)

**Authors reply to:** *What an incredible solution! (Comment on essd-2021-171)*
*Todd OBrien (Referee)*
*Referee comment on "The Plankton Lifeform Extraction Tool: A digital tool to increase the discoverability and usability of plankton time-series data" by Clare Ostle et al., Earth Syst. Sci. Data Discuss., https://doi.org/10.5194/essd-2021-171-RC1, 2021*

*Dear Authors,*
***Wow! What an excellent tool/dataset and a super clever solution to the challenge of comparing disparate time series programs and their data!*** *By aggregating species into a cluster of "lifeform" groupings, you solve methodlogical intercomparison challenges, while also creating a product that is more readly understandable to ecosystem- and policy- level users. You have been very careful and thorough in your design, and I especially appreciate the confidence ratings that you applied to the different lifeform categories. The PLET trait lookup table by itself is super valuable and useful, and being able to apply it directly to time series via the BASSH/PLET tool is even better. Overall, everything about your manuscript was excellent, and it was a pleasure to read. You provided a really well written paper and methodology, and I had only a tiny few questions after reading through it.*

Dear Todd O'Brien, thank you for taking the time to assess our paper and provide your extremely helpful and positive review, it is greatly appreciated. Below we address each of your points, including your text in black font, and our response in blue font.

***First Question:*** *You mention that each time series data set is preserved via a DOI. Is this doi/data the "original raw unaggregated data" (e.g., species counts by month by year **before being translated into lifeform categories**) or is it the data after it has been aggregated into lifeform categories (and individual species data is likely removed from that doi)? I ask because the reason IGMETS (and ICES WGZE/WGPME) only worked with totals (total copepods, total diatoms) was/is because some time series holders were hesitant to share their full raw species data. If you are sharing only the aggregated data, that would sooth most contributors (by not releasing the full raw data itself) and greatly increase/encourage more participation. That is an excellent solution to this ongoing challenge, and I think you should talk to ICES WGZE and ICES WGPME about getting more data sets into your tool.*

This is a very valid point, and one that we have discussed as a group many times. We ended up making it the data provider's choice, and we ask them to select an option to provide raw data or not on the '*Permission Agreement Form',* we ask this form to be completed upon submitting data to the lifeform tool. If the data providers have selected not to make the raw data available then the DOI links to their preferred citation which should be used to reference the monthly lifeform output, however if they have made the raw data available then the DOI

landing page will also provide a link to download that raw data. The lead-author is an admirer of your work, and we would very much like to collaborate with the working groups that you suggest, we see the PLET as something that will grow and expand, with the hope of becoming a global tool.

**Second Question:** *I really like Figures 5, 6, 7, and 8. Is it possible to get the BASSH/PLET tool to automatically generate those? (If it already does, I could not get figure it out.) Or perhaps you can pre-generate them, for the fixed site time series at least? This is very useful summary information about the time series, with or without the interactive tool component.*

This is a great suggestion and one that we hope to implement in a future version of PLET. We are hoping that we can provide a link to this manuscript when it is published, to act as a resource for the use of PLET and the current datasets. We have spent a lot of time discussing the potential issues of automating outputted comparisons, one of the issues being if coverage were not sufficient in some regions (depending on the dataset). If users want to extract lifeforms from PLET in a particular region and compare the outputs from different organisations they can apply their own method of normalisation to do so (e.g. The z-score figures within the manuscript), but we are not then accountable for any misinterpretation/misuse of the data.

**Third Question:** *The PLET Trait Look-Up Table is probably most important part of the entire database/tool "ecosystem". With what frequency do you hope to maintain and expand that table? On a related note, while you say you are marine-focused, adding Baltic Sea species would greatly expand your area of coverage in Europe. Surely HELCOM has most of the Trait info you need to make this expansion in the Look-up Table?*

Yes, we are very keen to keep maintaining the PLET trait look-up table, and very much hope to incorporate Baltic Sea species with the help of experts in that area, as well as other coastal species that exist in brackish waters. Currently this process is fairly ad-hoc and this is linked to the availability of funding, essentially when we have a funded project that is linked to the use of PLET, then we are able to work on the developments of the tool and the look-up table.

**Fourth Question:** *While you say (in the manuscript) you can't really compare different time series, you actually did .. in Figures 5,6,7,8, by using the Z-score. If you add these graphics to PLET somewhere (Question 2 above), multi-site comparisons or overviews should also be possible. Maybe not "live" (via the tool), but perhaps as pre-generated products elsewhere in the BASSH/PLET web page?*

What we meant by this statement is that caution should be taken in combining outputs from different datasets. By outputting monthly lifeforms the datasets become comparable to a degree, but there are always nuances or caveats

associated with different sampling techniques that need to be acknowledged. By keeping each data providers data separate but outputted in a unified way for comparison, we respect the differences and acknowledge the data provider, but can also infer changes or trends that may be a function of the sampling technique/regime instead of the actual changes in plankton. We like your suggestion of 'overviews' and this is something we would hope to implement in a future version of PLET. Please also see our response to your second question.

*I do not have anything negative to say, but two suggestions:*
**Suggestion #1: For me, t**he BASSH/PLET tool will usually "timeout" on the CPR data unless I subset the geographic region and/or time period. (This is not a problem with the single site time series, as they are much much smaller.) Are you using raw, full- geographic-resolution CPR data (i.e., the original silk locations)? For performance, you may want to subset those into geographic average boxes, perhaps 0.5 x 0.5 or 1.0 x 1.0 degree boxes. It would greatly reduce the number of data points the tool would have to process "on the fly".*

Thank you for your suggestion. We have developed a caching system whereby if a user has already requested an area/period than that data is cached and outputted to speed up processing, but there are still queries that have not been cached and it can hang on the larger data requests. We have not aggregated the CPR data into degree boxes yet simply because of wanting to keep the spatial resolution for some of the finer-scale regional boundaries used in assessments, but this could be a longer-term solution we consider. We have also written some scripts using curl/wget to run through iterations of data queries to pull out lifeform outputs in bulk, and for the CPR data these outputs can be grouped by degree boxes. We hope to publish these scripts on the PLET site once they have been tested and finalized.

**Suggestion #2:** *Table A1 is super long ... as in 29 pages in the review PDF. Since I am guessing that listing will change fairly regularly, why not make it an online file and only give a one page example of its content in the manuscript? I find the 29 pages distracting as I am trying to get down to Table A2 ...*
*I am really excited to see where this will go! Please reach out to ICES WGZE/WGPME to expand the coverage of this tool!*
*Todd O'Brien*

We agree with this suggestion and have therefore deleted Table A1 and referred to the Plankton Lifeform Traits Master List instead as the information that was within Table A1 is also included in the look-up table.

Thank you again for your thoughtful and thorough review, we hope you find our responses satisfactory, and we will certainly aim to reach out to the working groups suggested in the near future.
Clare Ostle and co-authors.

---

## Author Comment (AC2)

**Authors reply to:** *Aleksandra Lewandowska (Referee): This is an important and much needed tool with a great potential for further development.*

Dear Professor Lewandowska,

Thank you for the positive thoughts and suggestions you have provided to improve our manuscript, we greatly appreciate you taking the time to review it. Below we address each of your points, including your text in black font, and our response in blue font.

*Referee comment on "The Plankton Lifeform Extraction Tool: A digital tool to increase the discoverability and usability of plankton time-series data" by Clare Ostle et al., Earth Syst. Sci. Data Discuss., https://doi.org/10.5194/essd-2021-171-RC2*

*The manuscript is very well written, all functions of the database are clearly described and justified. I can only hope that this tool finds its way to the stakeholders and the policy makers in Europe.*

*I especially admire the functional groups **look-up table and the confidence rating**. It is easy to expand and continuously improve when more data are added. The spatial coverage of the database can be rapidly expanded, especially if SMHI extends access to their time-series from the Baltic Sea. If this happens, it would make sense to expand the lifeforms table by filamentous cyanobacteria to track their blooms in the Baltic Sea. Such information would be highly relevant to policy makers in the Baltic Sea region and some other coastal areas in Europe.*

This is a great suggestion for future versions of PLET and one that we would hope to implement if the datasets were expanded to this area. Thank you for this insight.

*I also appreciate that data from different sources are not aggregated. This gives a lot of **freedom for the users,** who can apply and develop their own statistical techniques to make generalisations.*

*Figures 5-8 are **wonderful examples how to use the PLET** and what kind of information can be extracted. I do not expect that the database developers will offer such visualisation tool, but this content of the manuscript is a great source of inspiration for the users. **Figures 2-3 on the sampling effort are extremely important** from the point of view of statistical diagnostics. It would be great if such figures could be included in the metafile description on the website, so that the user can easily see where are the potential gaps in each dataset and what are the limitations.*

We really like this suggestion, however some of the dataset's gaps will change depending on the regions selected for outputting lifeforms. We are hoping that we can provide a link to this manuscript and figures within when it is published, to act as a resource for the use of PLET and the current source datasets.

*Although the manuscript and the database are impressive, below are my suggestions for some improvement.*

*Regarding the manuscript:*

*It might be a good idea to highlight the **advantage of PLET over satellite derived information in the introduction**. There is a short sentence in the discussion about the limitation of bulk indices, such as total chlorophyll a concentration, but I think it would be good to have it earlier in text.*

Yes, we agree that this is an important point that we have overlooked. We have accordingly inserted a short paragraph at line 75 (i.e. making it the new third introductory paragraph) to add these Earth Observation data strengths and weaknesses.

"*To map changes in ocean colour, Earth Observation (EO) satellite tools provide unparalleled spatial coverage, and now offer the prospect of 20 years of ocean colour data, with increasingly resolved information, for example on trends of specific size-fractions of Chlorophyll a (Schmidt et al. 2020). However, the EO techniques are still not yet sufficiently developed to obtain information on changes in abundance of the key component planktonic functional groups, particularly for the zooplankton. Additionally, some taxonomic datasets now have up to 90 years of data which provide a critical perspective in assessing long-term change and which is unparalleled by satellites. We therefore need to maintain direct monitoring approaches for a holistic view of the plankton, and northwest European waters are particularly well-blessed with these time-series.*"

*There is no mention of **current development of plankton trait databases**, such as nutrient utilisation traits database (Edwards et al. 2015 - Ecological Archives), Baltic Sea phytoplankton traits database (Klais et al. 2017 - Functional Ecology), French phytoplankton traits database (Treyture et al. 2020 - Scientific Data). It would be good to place the PLET in their context. Maybe it would be worth adding the links to such trait databases in the future, if they exist for individual datasets, e.g. in the metafile description. This would be especially valued in those cases where taxonomic lists are made available.*

We agree these are important contributions that should be mentioned, we have added the following text has been added at line 129 to place PLET in the context of these datasets:

*There are a number of plankton trait datasets and plankton compilation efforts that are complementary to the PLET with the potential to feed into future versions of the tool, such as the nutrient utilisation traits dataset* (Edwards et al., 2015)*, the Baltic Sea phytoplankton traits dataset* (Klais et al., 2017)*, and the French lakes phytoplankton traits database* (Laplace-Treyture et al., 2021)*. While these are highly valuable resources, the authors are not aware of a platform to bring such information together and disseminate it in a consistent format. The design of PLET allows for this lifeform extraction and dissemination, with the aim to incorporate further plankton trait datasets in future versions.*

References added to manuscript:

*Edwards, K. F., Klausmeier, C. A. and Litchman, E.: Nutrient utilization traits of*

*phytoplankton, Ecology, 96(8), 2311–2311, doi:10.1890/14-2252.1, 2015.*

*Klais, R., Norros, V., Lehtinen, S., Tamminen, T. and Olli, K.: Community assembly and drivers of phytoplankton functional structure, Funct. Ecol., 31(3), 760–767, doi:10.1111/1365-2435.12784, 2017.*

*Laplace-Treyture, C., Derot, J., Prévost, E., Le Mat, A. and Jamoneau, A.: Phytoplankton morpho-functional trait dataset from French water-bodies, Sci. Data, 8(1), 1–9, doi:10.1038/s41597-021-00814-0, 2021.*

*Please add a short information how the PLET is **dealing with synonyms and updates in plankton taxonomy**. Is there an automatic check applied (e.g. with WORMS or AlgaeBase) or does it need to be made manually by data providers? In general, is there a systematic data quality check performed upon submission of the time-series? How often such quality check should be performed? I wonder how to ensure the consistent data quality among PLET database, if the data quality check is the responsibility of data providers. I am sure this is not a problem at the moment, but how to guarantee it in the future when the tool expands?*

Thank you for this point, it is something that we have worked hard to ensure consistent data quality, and as you can imagine it is an evolving process, we have added the following text has been added at line 160 to describe current procedure:

*When new datasets are submitted to the PLET the data providers supply aphia IDs of all of the taxa within their dataset. Following the pre-processing of the data by the data providers, the data manager of PLET and the manager of the Plankton Lifeform Traits Master List does a check of the submitted aphia IDs to highlight any missing taxa. Any taxa that are not included within the Plankton Lifeform Traits Master List are checked for compatibility with the lifeforms and their traits are added in discussion with an expert group and the data providers.*

*Please add a **link to the SMHI portal** in the chapter 3.1.8 similarly as you did for the other time series (https://www.smhi.se/en/services/open-data/national-archive-for-oceanographic-data/download-data-1.153150).*

Thank you, the link has been added to the manuscript at line 351.

*Regarding the PLET:*

*The **website performance** needs significant optimisation. I believe the problem is not in PLET, but rather in the host server, but this should be fixed before the tool expands. I tired different browsers and different computers, but the problem persists and the service website jams easily, even when I'm trying to limit my search and download data in small pieces. If this causes problems now, it will grow in the future.*

Thank you for your suggestion, we agree that we need to develop PLET further as the database increases in size in future versions of PLET. We have developed a caching system whereby if a user has already requested an area/period than that data is cached and outputted to speed up processing, but there are still queries that have not been cached and it can hang on the larger data requests. We are looking at

aggregating some of the largest datasets (such as CPR data) into subsets such as geographic average boxes of 1 x 1 degree. We have also written some scripts using curl/wget to run through iterations of data queries to pull out lifeform outputs in bulk, and for the CPR data these outputs can be grouped by degree boxes. We hope to publish these scripts on the PLET site once they have been tested and finalized.

*The short **description of sampling methodologies** (chapter 3.1) is excellent and could be added to the metafile together with the information on sampling effort (see my comment to Fig 2-3). This would make the service more user friendly.*

We really like this suggestion, and have added each method paragraph to the individual data providers metadata page within the tool.

*As the tool is meant for biodiversity assessment, it might be good to add some basic **information on changes in taxonomic resolution**. For example, species accumulation curves for each time series could give a clue on significant change in resolution, which can affect interpretation of the outcomes. As many diversity indices, including the most popular species richness, are sensitive to changes in sample size, this is an important information on data quality. Depending on the visualisation, such curves could have annotations with information on changes in methodology or instrumentation, which correspond to observed inconsistency.*

Thank you for this suggestion. The reason we have used lifeforms as indicators is because this is a robust grouping that many different datasets can contribute to, despite differences or changes in taxonomic resolution. For example, even if the taxonomic resolution during the times series improves, such as Decapoda larvae becoming identified to infraorder level (Brachyura), you would still have the same abundance values in the meroplankton and crustacea lifeform outputs from PLET. We stress in the first paragraph of section 3.1 that all the source time series data were screened such that they were internally consistent (i.e. taxa were recorded at the highest taxonomic level in which they were consistently recorded in, through each time series, lumping where necessary). For this reason, the issue of changes in taxonomic resolution within each time series is not an issue. However, each time series was analysed differently, with different levels of taxonomic resolution, so the comparison of diversity indices across multiple time series is not recommended. We have added text on these limitations on lines 570 with an added sentence (italicised below):

"….. coverage does not bias the combined results. *For example, due to the differing taxonomic resolution and sampling methods across the various component time series, we do not recommend simple comparisons of indices of species richness or diversity.* There is also the flexibility to……."

Thank you again for your insightful and thorough review, we hope you find our responses satisfactory.

Dr. Clare Ostle and co-authors.